# REASON FOR FUTURE, ACT FOR NOW: A PRINCIPLED ARCHITECTURE FOR AUTONOMOUS LLM AGENTS

## ABSTRACT

Large language models (LLMs) demonstrate impressive reasoning abilities, but translating reasoning into actions in the real world remains challenging. In particular, it is unclear how to complete a given task provably within a minimum number of interactions with the external environment, e.g., through an internal mechanism of reasoning. To this end, we propose the first framework with provable regret guarantees to orchestrate reasoning and acting, which we call "reason for future, act for now" (RAFA). Specifically, we design a prompt template for reasoning that learns from the memory buffer and plans a future trajectory over a long horizon ("reason for future"). At each step, the LLM agent takes the initial action of the planned trajectory ("act for now"), stores the collected feedback in the memory buffer, and reinvokes the reasoning routine to replan the future trajectory from the new state. The key idea is to cast reasoning in LLMs as learning and planning in Bayesian adaptive Markov decision processes (MDPs). Correspondingly, we prompt LLMs to form an updated posterior of the unknown environment from the memory buffer (learning) and generate an optimal trajectory for multiple future steps that maximizes a value function (planning). The learning and planning subroutines are performed in an "in-context" manner to emulate the actor-critic update for MDPs. Our theoretical analysis establishes a $\sqrt{T}$ regret, while our experimental validation demonstrates superior empirical performance.

## 1 INTRODUCTION

Large language models (LLMs) exhibit remarkable reasoning abilities, which open a new avenue for agents to interact with the real world autonomously. However, turning reasoning into actions remains challenging. Specifically, although LLMs are equipped with the prior knowledge obtained through pretraining, it is stateless in nature and ungrounded in the real world, which makes the resulting action suboptimal. To bridge the reasoning-acting gap, we aim to design an internal mechanism of reasoning on top of LLMs, which optimizes actions iteratively by incorporating feedbacks from the external environment. In particular, we focus on the sample efficiency of autonomous LLM agents in interactive decision-making tasks, which plays a key role in their practical adoption, especially when interactions are costly and risky. Our primary goal is to enable agents to complete a given task in a guaranteed manner through reasoning within a minimum number of interactions with the external environment.

Reinforcement learning (RL) is a well-studied paradigm for improving actions by collecting feedbacks. However, to tailor existing RL techniques for autonomous LLM agents, we lack a rigorous mapping between RL and LLMs, which leads to various conceptual discrepancies. For example, RL operates in a numerical system, where rewards and transitions are defined by scalars and probabilities. In comparison, the inputs and outputs of LLMs are described by tokens in a linguistic system. As another example, LLMs are trained on a general-purpose corpus and remain fixed throughout the interactive process. In contrast, RL trains actors and critics on the collected feedback iteratively. Thus, it appears inappropriate to treat LLMs as actors or critics under the RL framework, although all of them are parameterized by deep neural networks. Moreover, it remains unclear what reasoning with LLMs means under the RL framework, e.g., what are the inputs and outputs of a reasoning routine and how reasoning should be coordinated with acting. Such conceptual discrepancies prevent

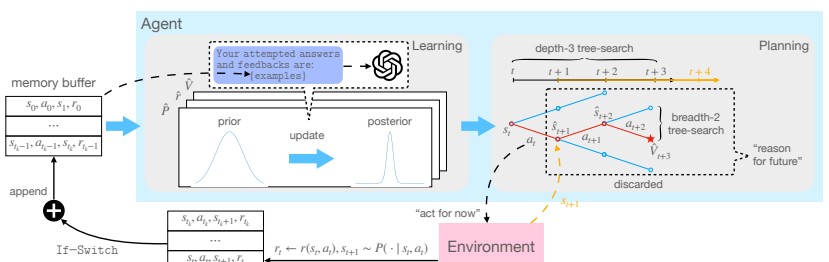

Figure 1: Illustration of the RAFA ("reason for future, act for now") framework.

us from establishing a principled framework beyond borrowing the "trial and error" concept from RL straightforwardly and make it difficult to achieve provable sample efficiency guarantees. For instance, it is known in RL that an improper design of agents may induce an exponential dependency on horizons in the sample complexity. Without the RL-LLM correspondence, it is hard to avoid the same flaw in autonomous LLM agents.

To address such conceptual discrepancies, we formalize reasoning and acting with LLMs under a Bayesian adaptive Markov decision process (MDP) framework, where the latent variable of interest is the unknown environment. The starting point is to cast the full history of states (of the external environment), actions, rewards, and their linguistic summaries in the memory buffer as the information state of Bayesian adaptive MDPs. Throughout the interactive process, the information state accumulates a growing collection of feedbacks from the external environment, which is mapped to an optimized action at each step by an internal mechanism of reasoning. As detailed below, we construct the reasoning routine through two key subroutines, namely learning and planning, which are instantiated by LLMs with specially designed prompts. **(a)** The learning subroutine forms an updated posterior of the unknown environment from the memory buffer. Depending on whether we emulate the model-based or model-free approach of RL, the learning subroutine infers the transition and reward models (model) or/and the value function (critic). **(b)** The planning subroutine generates an optimal policy (actor) or trajectory for multiple future steps, which maximizes the value function (up to a certain error). Depending on the specific configuration of the state and action spaces (continuous versus discrete) and the transition and reward models (stochastic versus deterministic), the planning subroutine emulates the value iteration algorithm, the random shooting algorithm, or the Monte-Carlo tree-search algorithm.

Although LLMs remain fixed throughout the interactive process, they are prompted to utilize the growing collection of feedbacks from the external environment as contexts. Through the learning subroutine, the collected feedback reduces the posterior uncertainty in models or values, which allows the planning subroutine to obtain an improved policy at each step. In other words, we emulate the actor-model or actor-critic update for Bayesian adaptive MDPs in an in-context manner, where LLMs function as an internal mechanism that improves models, values, and policies iteratively. Specifically, existing RL methods use deep neural networks to parameterize models, values, and policies, which map states (of the external environment) and actions to scalars and probabilities. In comparison, we use LLMs to represent the learning and planning algorithms in RL, which are composed to map data in the memory buffer to actions. Here, data and actions are allowed to be tokens in a linguistic system.

We conclude our contributions in this paper from two perspectives. **(a)** Our theoretical analysis proves that RAFA achieves a $\sqrt{T}$ regret. In particular, the regret bound highlights an intriguing interplay between the prior knowledge obtained through pretraining and the uncertainty reduction achieved by reasoning and acting. **(b)** Our empirical validation shows that RAFA outperforms various existing frameworks in interactive decision-making tasks, including ALFWorld, BlocksWorld, Game of 24, and a new benchmark based on Tic-Tac-Toe.

## 1.1 LITERATURE

Due to the page limit, we defer the detailed discussion on large language model (LLM), in-context learning (ICL), and reinforcement learning (RL) under a Bayesian framework to Appendix A.

**Reasoning with LLM.** We build on a recent line of work that develops various prompting schemes to improve the reasoning performance of LLMs. "Chain of thoughts" ("CoT") [67] decomposes

a challenging problem into several reasoning stages and guides LLMs to solve them one by one. As generalizations, "tree of thoughts" [73], "graph of thoughts" [74], "algorithm of thoughts" [50], and "cumulative reasoning" [76] provide different graph-search schemes to guide LLMs. See also [63, 16, 15]. Also, "reasoning via planning" ("RAP") [23] emulates the Monte-Carlo tree-search (MCTS) algorithm to reduce the search complexity. For embodied LLM agents, [25] propose to decompose a complex task into multiple executable steps. Most of them focus on general reasoning tasks, e.g., solving a mathematical or logic puzzle, where LLMs generate a detailed trace (trajectory) of arguments through an internal mechanism to reach a final answer. Here, LLMs play the same role as the planning subroutine in RAFA. In contrast, we focus on interactive decision-making tasks, where autonomous LLM agents collect feedbacks from the external environment to optimize actions iteratively. In particular, we aim to complete a given task within a minimum number of interactions with the external environment. To this end, it is essential to operate three interleaved modules, namely learning, planning, and acting, in a closed loop. While it is feasible to incorporate existing graph-search or MCTS schemes as the planning subroutine for generating trajectories, our core contribution is a principled framework that executes a selected subset of the planned trajectory to collect feedbacks ("act for now") and replans an improved trajectory from the new state by learning from feedbacks ("reason for future"). From an RL perspective, existing graph-search or MCTS schemes are analogous to an open-loop method, e.g., motion planning or trajectory optimization [8], which does not involve interactions with the external environment. To integrate them into a closed-loop approach, e.g., model predictive control [43], one has to specify how to act given the planned trajectory and when to reinvoke the reasoning (learning and planning) routine, which is the key technique of RAFA. Another recent line of work tackles more complex tasks by allowing LLMs to access various additional modules, e.g., tools, programs, and other learning algorithms [4, 51, 35, 34, 11], or by finetuning LLMs on the collected feedback [75, 31, 41]. Integrating them with RAFA is left as a future direction of research.

**Acting (and Reasoning) with LLM.** We build on a recent line of work that develops various closed-loop frameworks for interacting with the external environment. "Inner monologue" [26] and "Re-Act" [72] combine reasoning and acting to refine each other for the first time. In comparison, RAFA provides a specific schedule for orchestrating reasoning and acting (as discussed above). As generalizations, "Reflexion" [53] enables autonomous LLM agents to revise the current action of a pregenerated trajectory by learning from feedbacks, especially when they make mistakes. See also [28]. However, making a local revision to the pregenerated trajectory is myopic because it fails to consider the long-term consequence of actions. Consequently, the obtained policy may get trapped by a local optimum. From an RL perspective, "Reflexion" [53] is an oversimplified version of RAFA, where the planning subroutine revises the current action to maximize the reward function ("reason for now") instead of planning multiple future steps to maximize the value function ("reason for future"), which measures the expected cumulative future reward. To remedy this issue, "AdaPlanner" [58] regenerates the whole trajectory at each step, which yields a global improvement. See also [64]. However, the reasoning routine of "AdaPlanner" requires a handcrafted set of programs to reject suboptimal candidate trajectories. Without the domain knowledge of a specific task, the regenerated trajectory is not necessarily optimal, i.e., maximizing the value function (up to a certain error). In contrast, the reasoning routine of RAFA is designed following the principled approach in RL. In particular, the learning subroutine infers the transition and reward models (model) or/and the value function (critic), while the planning subroutine emulates the value iteration algorithm, the random shooting algorithm, or the MCTS algorithm, none of which use any domain knowledge. As a result, RAFA achieves provable sample efficiency guarantees for the first time and outperforms those existing frameworks empirically.

## 2 BRIDGING LLM AND RL

**Interaction Protocol.** We use Markov decision processes (MDPs) to model how autonomous LLM agents interact with the external environment. We consider an infinite-horizon MDP $M = (\mathcal{S}, \mathcal{A}, P, r, \rho, \gamma)$, where $\mathcal{S}$ is the state space, $\mathcal{A}$ is the action space, $P : \mathcal{S} \times \mathcal{A} \mapsto \Delta(\mathcal{S})$ is the transition kernel, $r : \mathcal{S} \times \mathcal{A} \mapsto [0, 1]$ is the reward function, $\rho$ is the initial distribution of states,

and $\gamma \in (0,1)$ is the discount factor. Here, $P$ gives the probability distribution of the next state given the current state and action, while $r$ is assumed to be deterministic without loss of generality. For notational simplicity, we parameterize $P$ and $r$ by a shared parameter $\theta^\star \in \Theta$ and denote them as $P_{\theta^\star}$ and $r_{\theta^\star}$. At the $t$-th step, the LLM agent receives a state $s_t \in \mathcal{S}$, takes an action $a_t \in \mathcal{A}$ following the current policy $\pi_t : \mathcal{S} \mapsto \mathcal{A}$, and receives a reward $r_t = r_{\theta^\star}(s_t, a_t)$. Subsequently, the external environment transits to the next state $s_{t+1} \sim P_{\theta^\star}(\cdot|s_t, a_t)$, while the LLM agent computes the updated policy $\pi_{t+1}$ through an internal mechanism of reasoning (as discussed below). Note that $\mathcal{S}$ and $\mathcal{A}$ are represented by tokens in a linguistic system. Here, $\pi \in \Pi$ is assumed to be deterministic without loss of generality, where $\Pi$ is the feasible set of policies.

**Value Function.** For a policy $\pi$ and a parameter $\theta$ of the transition and reward models, we define the state-value and action-value functions

$$V_\theta^\pi(s) = \mathbb{E}\Big[\sum_{t=0}^{\infty} \gamma^t r_\theta(s_t, a_t)\Big| s_0 = s\Big], \quad Q_\theta^\pi(s, a) = \mathbb{E}\Big[\sum_{t=0}^{\infty} \gamma^t r_\theta(s_t, a_t)\Big| s_0 = s, a_0 = a\Big], \quad (2.1)$$

where $\mathbb{E}$ is taken with respect to $a_t = \pi(s_t)$ and $s_{t+1} \sim P_\theta(\cdot|s_t, a_t)$ for all $t \geq 0$. In other words, $V_\theta^\pi$ (and $Q_\theta^\pi$) gives the expected cumulative future reward from the current state $s$ (and action $a$). To define the optimal policy $\pi_\theta^\star$ with respect to a given parameter $\theta$, we define the Bellman optimality equation as

$$Q_\theta^\star(s, a) = r_\theta(s, a) + \gamma(P_\theta V_\theta^\star)(s, a), \quad V_\theta^\star(s) = \max_{a \in \mathcal{A}} Q_\theta^\star(s, a), \quad (2.2)$$

where $Q_\theta^\star$ and $V_\theta^\star$ are the fixed-point solutions Here, we define $(P_\theta V_\theta^\star)(s, a) = \mathbb{E}[V_\theta^\star(s')]$, where $\mathbb{E}$ is taken with respect to $s' \sim P_\theta(\cdot|s, a)$. Let $\pi_\theta^\star(s) = \arg\max_{a \in \mathcal{A}} Q_\theta^\star(s, a)$. We define $\texttt{PL}^\star : \Theta \mapsto \Pi$ as the planning oracle that maps $\theta$ to $\pi_\theta^\star$. See [59] for the existence and uniqueness guarantees for $Q_\theta^\star$, $V_\theta^\star$, and $\pi_\theta^\star$.

**Sample Efficiency.** Let $\theta^\star$ be the underlying parameter that generates states and rewards. As the performance metric, we define the Bayesian regret

$$\mathfrak{R}(T) = \mathbb{E}\Big[\sum_{t=0}^{T-1} V_{\theta^\star}^{\pi^\star}(s_t) - V_{\theta^\star}^{\pi_t}(s_t)\Big], \quad \text{where } \pi^\star = \texttt{PL}^\star(\theta^\star). \quad (2.3)$$

Here, $\mathbb{E}$ is taken with respect to the prior distribution $p_0$ of $\theta^\star$, the stochastic outcome of $s_t$, and the iterative update of $\pi_t$, which involves states, actions, and rewards until the $t$-th step, i.e., the full history $\mathcal{D}_t = \{(s_i, a_i, s_{i+1}, r_i)\}_{i=0}^{t-1}$. We aim to design a sample-efficient agent that satisfies $\mathfrak{R}(T) = o(T)$, i.e., the Bayesian regret is sublinear in the total number of interactions $T$.

**What Reasoning Means and Role of LLM.** We formalize reasoning and acting with LLMs under a Bayesian adaptive MDP framework [19], where the underlying parameter $\theta^\star$ is the latent variable of interest and the full history $\mathcal{D}_t$ (and its linguistic summary) is the information state. In particular, we aim to design an internal mechanism on top of LLMs that maps $\mathcal{D}_t$ to an optimized action $a_t$ or the corresponding policy $\pi_t$ (reasoning), which is executed in the external environment (acting). To this end, we construct the reasoning routine through two key subroutines, which emulate the learning and planning algorithms in RL. Specifically, the learning subroutine maps $\mathcal{D}_t$ to the posterior distribution $p_t$ of $\theta^\star$, while the planning subroutine maps $p_t$ or a sampled parameter $\theta \sim p_t$ to $\pi_t$. In other words, the learning subroutine forms an updated posterior of the unknown environment from the memory buffer, while the planning subroutine approximates the planning oracle $\texttt{PL}^\star$. As shown in Section 3, we invoke the ICL ability of LLMs to achieve the former goal (implicitly), while we design a prompt template for LLMs to achieve the latter goal (explicitly). Following the principled approach in RL, we develop a specific schedule for orchestrating reasoning (learning and planning) and acting, which is proven as sample-efficient in Section 4.

## 3 ALGORITHM

**Architecture of `RAFA`.** By leveraging the LLM-RL correspondence in Section 2, we provide a principled framework for orchestrating reasoning and acting, namely "reason for future, act for now" (`RAFA`), in Algorithms 1 and 2. In Section 4, we present the RL counterpart of `RAFA` in Algorithm 3 to illustrate the design rationale and establish the theoretical foundation. At the $t$-th step of Algorithm 1, the LLM agent invokes the reasoning routine, which learns from the memory buffer and plans a future trajectory over a long horizon ("reason for future" in Line 6), takes the

---

**Algorithm 1** Reason for future, act for now (`RAFA`): The LLM version.

---

1: **input**: An LLM learner-planner `LLM-LR-PL`, which aims at generating an optimal trajectory given an initial state and returns the initial action (e.g., Algorithm 2), and a switching condition `If-Switch`.
2: **initialization**: Sample the initial state $s_0 \sim \rho$, set $t = 0$, and initialize the memory buffer $\mathcal{D}_0 = \varnothing$.
3: **for** $k = 0, 1, \ldots,$ **do**
4:      Set $t_k \leftarrow t$.
5:      **repeat**
6:          Learn and plan given memory $\mathcal{D}_{t_k}$ to get action $a_t \leftarrow$ `LLM-LR-PL`$(\mathcal{D}_{t_k}, s_t)$. ("reason for future")
7:          Execute action $a_t$ to receive reward $r_t$ and state $s_{t+1}$ from environment.      ("act for now")
8:          Update memory $\mathcal{D}_{t+1} \leftarrow \mathcal{D}_t \cup \{(s_t, a_t, s_{t+1}, r_t)\}$.
9:          Set $t \leftarrow t + 1$.
10:      **until** `If-Switch`$(\mathcal{D}_t)$ is `True`.      (the switching condition is satisfied)
11: **end for**

---

initial action of the planned trajectory ("act for now" in Line 7), and stores the collected feedback (state, action, and reward) in the memory buffer (Line 8). Upon the state transition of the external environment, the LLM agent reinvokes the reasoning routine to replan another future trajectory from the new state (Line 6 following Line 9). To ensure the learning and planning stability, we impose the switching condition (Line 10) to decide whether to incorporate the newest chunk of history, i.e., the set difference $\mathcal{D}_t - \mathcal{D}_{t_k}$, into the information state, which is used in the reasoning routine as contexts. In other words, the reasoning routine uses the same history $\mathcal{D}_{t_k}$ for all $t_k \leq t < t_{k+1}$ until the $(k+1)$-th switch at the $(t_{k+1} - 1)$-th step, which guarantees that the posterior distribution and the optimized action or the corresponding policy are updated in a conservative manner. We specify the switching condition in Sections 4 and 5.

**"Reason for Future" (Line 6 in Algorithm 1 and Lines 3-11 in Algorithm 2).** As detailed below, the reasoning routine composes the learning and planning subroutines to map the full history $\mathcal{D}_{t_k}$ (until the $t_k$-th step) to an optimized action $a_t$. Note that the reasoning routine does not interact with the external environment throughout the learning and planning subroutines.

• The learning subroutine (Lines 3-4 in Algorithm 2) maps $\mathcal{D}_{t_k}$ to a transition kernel (`Model`) and a value function (`Critic`), which are used in the planning subroutine. Intuitively, we prompt LLMs to form an updated posterior of the unknown environment from the memory buffer. Here, the updated posterior is instantiated by `Model` and `Critic`, which estimate their ground-truth counterparts in association with the data-generating parameter. From an RL perspective (Sections 2 and 4), the learning subroutine maps $\mathcal{D}_{t_k}$ to the posterior distribution $p_t$ of the underlying parameter $\theta^\star$, which generates states and rewards, and returns the transition kernel $P_\theta$ and the value function $V_\theta^{\pi_t}$, where $\theta \sim p_t$ is the sampled parameter and $\pi_t$ is the current policy. On the other hand, the ICL ability of LLMs allows us to bypass the posterior update of $p_t$, sampling $\theta$ from $p_t$, and the explicit parameterization of $P_\theta$ and $V_\theta^{\pi_t}$ in RL. Instead, we represent $P_\theta$ and $V_\theta^{\pi_t}$ using two LLM instances with specially designed prompts, which instruct them to use $\mathcal{D}_{t_k}$ as contexts to generate the next state and evaluate a given trajectory or the corresponding policy. As $\mathcal{D}_{t_k}$ accumulates a growing collection of feedbacks from the external environment, it reduces the posterior uncertainty about the unknown environment, which yields more accurate versions of `Model` and `Critic`. Consequently, the planning subroutine is able to use them to assess the long-term outcome of actions with a higher accuracy. Depending on whether we emulate the model-based or model-free approach of RL, we may choose to emulate `Model` or `Critic` individually. For illustration, we consider a deterministic setting of transitions and rewards with discrete state and action spaces, where we emulate both of them in a tree-search example.

• The planning subroutine (Lines 5-11 in Algorithm 2) maps `Model` and `Critic` to a future trajectory $(s_0^\dagger, a_0^\dagger, \ldots, s_U^\dagger, a_U^\dagger)$, where $s_0^\dagger$ is the current state $s_t$ and $a_0^\dagger$ is executed in the external environment as the current action $a_t$ during the acting phase. Intuitively, we prompt LLMs to generate an optimal policy (actor) for multiple future steps, which maximizes the value function (`Critic`). From an RL perspective (Sections 2 and 4), the planning subroutine approximates the planning oracle `PL`$^\star$, which maps a given parameter $\theta$ to the optimal policy $\pi_\theta^\star$ or the corresponding action $a_t = \pi_\theta^\star(s_t)$. As two LLM instances from the learning subroutine, `Model` and `Critic` instantiate the transition kernel $P_\theta$ and the value function $V_\theta^{\pi_t}$ in association with the sampled parameter

---

**Algorithm 2** The LLM learner-planner (`LLM-LR-PL`): A tree-search example. (the deterministic case)

---

1: **input**: The memory buffer $\mathcal{D}$, the initial state $s$, the search breadth $B$, and the search depth $U$.
2: **initialization**: Initialize the state array $\mathcal{S}_0 \leftarrow \{s\}$ and the action array $\mathcal{A}_0 \leftarrow \varnothing$.
———————————————————— (the learning subroutine) ————————————————————
3: Set `Model` as an LLM instance prompted to use $\mathcal{D}$ as contexts to generate the next state.
4: Set `Critic` as an LLM instance prompted to use $\mathcal{D}$ as contexts to estimate the value function.
———————————————————— (the planning subroutine) ————————————————————
5: Set `Elite` as an LLM instance prompted to use $\mathcal{D}$ as contexts to generate multiple candidate actions.
6: **for** $u = 0, \ldots, U$ **do**
7:    For each current state in $\mathcal{S}_u$, invoke `Elite` to generate $B$ candidate actions and store them in $\mathcal{A}_u$.
8:    For each candidate action in $\mathcal{A}_u$, invoke `Model` to generate the next state and store it in $\mathcal{S}_{u+1}$.
9: **end for**
10: For all resulting rollouts in $\mathcal{S}_0 \times \mathcal{A}_0 \times \cdots \times \mathcal{S}_U \times \mathcal{A}_U$, invoke `Critic` to evaluate the expected cumulative future reward and select the best one $(s_0^\dagger, a_0^\dagger, \ldots, s_U^\dagger, a_U^\dagger)$, where $s_0^\dagger = s$.
11: **output**: The initial action $a_0^\dagger$ of the selected rollout.

---

$\theta \sim p_t$ (as discussed above). Hence, we are able to simulate a given number of trajectories with `Model`, evaluate them with `Critic`, and obtain an improved policy, which is achieved by specially designed prompts instead of a numerical algorithm. By maximizing the expected cumulative future reward (instead of the immediate reward), the planning subroutine returns an optimized action that improves the long-term outcome. In Section 4, we identify two error sources that affect the planning subroutine, namely the posterior uncertainty, which is inherited from `Model` and `Critic` due to the finite size of $\mathcal{D}_{t_k}$, and the planning suboptimality, which is induced by the limited capacity for computation, e.g., the bounded width and depth of tree-search (Lines 6-9 in Algorithm 2). Depending on the specific configuration of the state and action spaces (continuous versus discrete) and the transition and reward models (stochastic versus deterministic), we may choose to emulate the value iteration algorithm, the random shooting algorithm, or the Monte-Carlo tree-search algorithm. All of them allow `RAFA` to achieve provable sample efficiency guarantees as long as they satisfy a specific requirement of optimality (Definition 4.2). For illustration, we emulate the tree-search algorithm and defer its stochastic variant to Appendix B.

**"Act for Now" (Lines 7-10 in Algorithm 1).** At the current state $s_t$, the LLM agent executes the optimized action $a_t$ in the external environment, which is obtained from the reasoning routine. Specifically, we take the initial action $a_0^\dagger$ of the planned trajectory $(s_0^\dagger, a_0^\dagger, \ldots, s_U^\dagger, a_U^\dagger)$, where $s_0^\dagger = s_t$ and $a_0^\dagger = a_t$, and discard the remaining subset. At the next state $s_{t+1}$, the LLM agent replans another future trajectory $(s_0^\dagger, a_0^\dagger, \ldots, s_U^\dagger, a_U^\dagger)$ with $s_0^\dagger = s_{t+1}$ and $a_0^\dagger = a_{t+1}$. In other words, the acting phase follows a short-term subset of the long-term plan, which is regenerated at every new state. The LLM agent stores the collected feedback $(s_t, a_t, r_t, s_{t+1})$ in the memory buffer $\mathcal{D}_t$ and queries a switching condition `If-Switch` to decide when to update the information state $\mathcal{D}_{t_k} \subseteq \mathcal{D}_t$, which is used in the reasoning routine as contexts for learning and planning. Intuitively, we incorporate the newest chunk of history $\mathcal{D}_t - \mathcal{D}_{t_k}$ to improve the current policy only in the case that it carries significant novel information, e.g., when the LLM agent loses for the first time following a winning streak. In Section 4, we provide a principled implementation of the switching condition, which measures the posterior uncertainty given $\mathcal{D}_t$ with entropy and compares it against that given $\mathcal{D}_{t_k}$. From an RL perspective, the lazy update ensures the learning and planning stability and plays a pivotal role in the regret analysis. In Section 5, we develop several practical variants that achieve superior empirical performance.

## 4 THEORY

We establish provable sample efficiency guarantees for `RAFA` (Algorithms 1 and 2) through its RL counterpart (Algorithm 3 in Appendix B). In Line 6 of Algorithm 3, the reasoning routine forms an updated posterior of the unknown environment (learning) and generates an optimized action from an improved policy (planning), mirroring `RAFA`. Here, we emulate the model-based approach of RL and cast `RAFA` as a Thompson sampling (TS) method. The following assumption and definition formalize the learning and planning subroutines of `RAFA` (Lines 3-4 and 5-11 in Algorithm 2).

**Learning.** Let $\text{LLM}_{\mathcal{D},g}$ be an LLM instance with $\mathcal{D}$ as contexts and $g$ as instructions to perform a specific task. Specifically, $g^{\dagger}$ prompts LLMs to predict the next state $s'$ and the received reward $r$ from the current state $s$ and the current action $a$, i.e., $\text{LLM}_{\mathcal{D},g^{\dagger}} : \mathcal{S} \times \mathcal{A} \to \mathcal{S} \times [0,1]$, where the generated state is stochastic. We denote the Markov kernel in association with $\text{LLM}_{\mathcal{D},g^{\dagger}}$ as $P_{\text{LLM}_{\mathcal{D},g^{\dagger}}}(s', r|s, a)$. Also, we denote the posterior distribution of the transition and reward models as $\mathbb{P}_{\text{model}}(P_{\theta}, r_{\theta}|\mathcal{D})$.

**Assumption 4.1** (LLMs Perform Implicit Bayesian Inference). The Markov kernel $P_{\text{LLM}_{\mathcal{D},g^{\dagger}}}$ follows the posterior distribution $\mathbb{P}_{\text{model}}(\cdot|\mathcal{D})$.

Assumption 4.1 states that LLMs perform implicit Bayesian inference, which is verified both theoretically and empirically as the underlying mechanism of ICL [69, 77, 78, 62, 68, 27, 30]. In particular, [69, 62] validate it in a general setting for generating texts, while [30] prove it in the imitation setting of RL to develop a new framework for pretrained decision transformers. We consider a related setting for predicting states and rewards that are described by texts. Here, the pretraining dataset is a general-purpose corpus covering a wide variety of $\mathcal{D}$ and $g$, whereas $(P_{\theta}, r_{\theta})$ or $\theta$ is the latent concept of interest. In comparison, [30] consider the imitation setting for predicting the optimal action without an explicit planner, where the pretraining dataset contains the numerical trajectory labeled by experts. In Appendix D, we prove that Assumption 4.1 holds for a specific parameterization of $(P_{\theta}, r_{\theta})$ under three regularity conditions, namely (a) LLMs are trained to replicate the pretraining distribution, which is assumed in [48, 66, 69] to simplify the statistical analysis, (b) the pretraining dataset is generated through a Bayesian mechanism with a latent concept, which is a simplified version of the latent variable model in [69] and resembles that in [62], and (c) LLMs are able to parameterize an implicit Bayesian inference mechanism, which is proved in [77, 78] for the attention architecture. Note that, if Assumption 4.1 holds approximately, the regret analysis can be relaxed to accommodate the additional error in the posterior distribution.

**Planning.** Assumption 4.1 allows us to bridge RAFA and TS. In the learning subroutine of RAFA, we emulate $P_{\theta}$ with Model (Line 3 in Algorithm 2) and $V_{\theta}^{\pi}$ with Critic (Line 4 in Algorithm 2), which is determined by $P_{\theta}$, $r_{\theta}$, and $\pi$. At the $t$-th step, $\theta$ is sampled from $p_t$, i.e., the updated posterior given the full history $\mathcal{D}_{t_k}$ (until the $t_k$-th step). To formalize the planning subroutine of RAFA, we define the planning suboptimality. Recall that $\Theta$ is the parameter space, $\Pi$ is the policy space, and $\text{PL}^{\star}$ is the planning oracle, which is defined in Section 2.

**Definition 4.2** ($\epsilon$-Optimality of Planner). A planning algorithm $\text{PL}^{\epsilon} : \Theta \mapsto \Pi$ is an $\epsilon$-optimal planner if $\max_{s \in \mathcal{S}}[V_{\theta}^{\text{PL}^{\star}(\theta)}(s) - V_{\theta}^{\text{PL}^{\epsilon}(\theta)}(s)] \leq \epsilon$ for all $\theta \in \Theta$.

As a special case of Definition 4.2, we present the value iteration algorithm in Appendix F, where we use a truncated horizon $U$, i.e., a finite length of the lookahead window. Here, $\epsilon$ decreases as $U$ increases. See a detailed discussion in Appendix C.1.

**Switching.** We consider an implementation of the switching condition (Line 10 in Algorithms 1 and 3). Let $\mathfrak{H}(p)$ be the differential entropy of $p$. We define the posterior entropy given $\mathcal{D}_t$ as

$$H_t = \mathfrak{H}(p_t) = -\int_{\Theta} p_t(\theta) \cdot \log p_t(\theta) \mathrm{d}\theta. \tag{4.1}$$

As long as $H_{t_k} - H_t > \log 2$, i.e., the memory buffer accumulates one extra bit of information, we incorporate $\mathcal{D}_t - \mathcal{D}_{t_k}$ into the information state and use it to improve the current policy. The switching condition ensures that $\pi_t$ is switched for a logarithmic number of times, which is a key step in establishing the sublinear regret. Intuitively, the lazy update of policies ensures the learning and planning stability. On the other hand, calculating the posterior entropy is challenging in practice. In Section 5, we develop several practical variants that achieve superior empirical performance.

**Regret.** We define the information ratio to characterize the tail behavior of the posterior distribution [1, 40, 46, 45, 47, 36]. Let $\delta \in (0, 1)$ be the confidence level, $\mathcal{D}_T = \{(s_t, a_t, s_{t+1}, r_t)\}_{t=0}^{T-1}$ be an arbitrary dataset collected in the underlying MDP, and $\{V_t\}_{t=0}^{T-1}$ be a value function sequence adapted to $\{\sigma(\mathcal{D}_t)\}_{t=0}^{T-1}$, where $\sigma(\mathcal{D}_t)$ is the sigma-algebra of $\mathcal{D}_t \subseteq \mathcal{D}_T$. We define the information gain as $I(\theta; \xi_{t+1}|\mathcal{D}_t) = H_t - H_{t+1}$. Here, $\xi_{t+1}$ denotes $(s_t, a_t, s_{t+1}, r_t)$ and $H_t$ is defined in (4.1), where $p_t$ is the posterior distribution given $\mathcal{D}_t$.

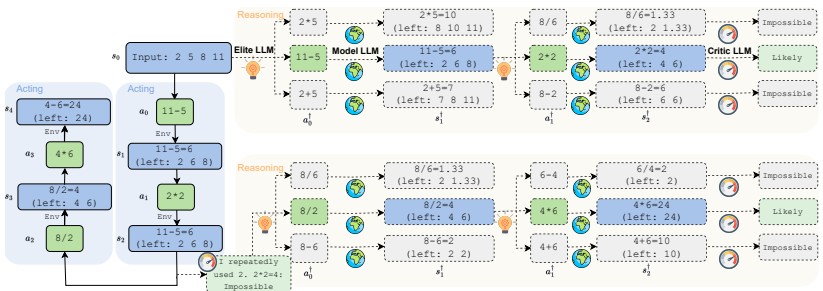

Figure 2: RAFA for Game of 24. Actions are proposed (dotted) and selected (green). Hallucinations that the same number can be reused are mitigated through interactions.

**Definition 4.3** (Information Ratio). The information ratio $\Gamma_{t^\dagger}(\delta)$ is the smallest number for which, if $H_{t^\dagger} - H_t \le \log 2$, then it holds for all $t \in \{t^\dagger, \dots, T-1\}$ with probability at least $1-\delta$ that

$$\left| (r_{\theta^\star} - r_{\theta_{t^\dagger}})(s_t, a_t) + \left((P_{\theta^\star} - P_{\theta_{t^\dagger}})V_t\right)(s_t, a_t) \right| \le \Gamma_{t^\dagger}(\delta) \cdot \sqrt{I(\theta; \xi_{t+1}|\mathcal{D}_t)}, \quad (4.2)$$

where $\theta^\star$ is the data-generating parameter and $\theta_{t^\dagger} \sim p_{t^\dagger}$ is a sampled parameter.

Definition 4.3 quantifies the estimation error of the sampled parameter $\theta_{t^\dagger}$ in terms of approximating the data-generating parameter $\theta^\star$. To achieve this, we use the information gain $I(\theta; \xi_{t+1}|\mathcal{D}_t)$ as a benchmarking quantity. Intuitively, the information ratio $\Gamma_{t^\dagger}(\delta)$ characterizes how exploration reduces uncertainty. See a detailed discussion in Appendix C.2.

We characterize the Bayesian regret of Algorithm 1 by connecting it to Algorithm 3. Recall that the Bayesian regret is defined in (2.3) and $\gamma \in (0, 1)$ is the discount factor.

**Theorem 4.4** (Bayesian Regret). Under Assumption 4.1, the Bayesian regret of RAFA satisfies

$$\mathfrak{R}(T) = \mathcal{O}\left( \frac{\gamma \cdot \sup_{t^\dagger < T} \Gamma_{t^\dagger}(\delta) \cdot \mathbb{E}[\sqrt{H_0 - H_T}]}{1 - \gamma} \cdot \sqrt{T} + \frac{\gamma \delta}{(1-\gamma)^2} \cdot T + \epsilon \cdot T + \frac{\gamma \cdot \mathbb{E}[H_0 - H_T]}{(1-\gamma)^2} \right).$$

We provide the proof in Appendix E. Theorem 4.4 establishes the $\sqrt{T}$ regret of RAFA (Algorithms 1 and 3) for a proper choice of the confidence level $\delta$ and the planning suboptimality $\epsilon$, e.g., $\delta = \mathcal{O}(1/\sqrt{T})$ and $\epsilon = \mathcal{O}(1/\sqrt{T})$. Here, the first term in the upper bound in Theorem 4.4 is the leading term and involves several multiplicative factors, namely the effective horizon $1/(1 - \gamma)$, the information ratio $\Gamma_{t^\dagger}(\delta)$, and the information gain $H_0 - H_T$ throughout the $T$ steps, which are common in the RL literature [1, 40, 46, 45, 47, 36]. In particular, $H_0$ highlights the prior knowledge obtained through pretraining, as $H_0$ quantifies the prior uncertainty of LLMs before incorporating any collected feedback. Hence, $H_0 - H_T$ highlights the uncertainty reduction achieved by reasoning and acting, as $H_T$ quantifies the posterior uncertainty of LLMs after incorporating the collected feedback. In Appendix F, we prove that $H_0 - H_T = \mathcal{O}(d \cdot \log T)$ for linear kernel MDPs, which implies $\mathfrak{R}(T) = \widetilde{\mathcal{O}}(\sqrt{T})$. Here $\widetilde{\mathcal{O}}$ hides the logarithmic factor.

## 5 EXPERIMENT

We evaluate RAFA in several text-based benchmarks, e.g., Game of 24, ALFWorld, BlocksWorld, and Tic-Tac-Toe. The detailed setups, results, and ablations are provided in Appendix G, while the detailed prompts are found in Appendix H.

### 5.1 GAME OF 24

Game of 24 [73] is a mathematical puzzle to obtain 24 from four natural numbers through basic arithmetic operations. The state is the (possibly unfinished) current formula and the action is the next formula (or the modified part).

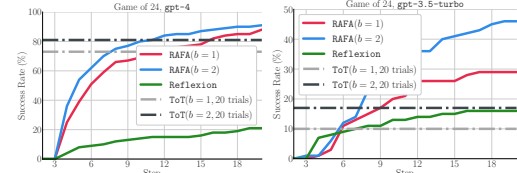

Figure 3: Sample efficiency on Game of 24.

**Setup.** We emulate the tree-search algorithm to plan ($B \in \{1, 2\}$). At the $t$-th step, RAFA learns from the memory buffer and switches to a new policy upon receiving an unexpected reward, which is the switching condition. After the $t$-th step, RAFA digests the collected feedback and generates a linguistic summary, which is saved into the memory buffer to avoid similar previous mistakes.

**Result.** RAFA attains SOTA performances as shown in Table 1. RAFA achieves superior sample efficiency by mitigating hallucinations and avoid careless trials (Figures 2 and 3).

|  | gpt-4 | gpt-3.5 |
|---|---|---|
| RAFA ($B=1$) | 89% | 29% |
| RAFA ($B=2$) | **93%** | **46%** |
| ToT ($B=1$) | 73% | 10% |
| ToT ($B=2$) | 81% | 17% |
| Reflexion | 21% | 16% |

Table 1: Game of 24 results.

|  | Pick | Clean | Heat | Cool | Exam | Pick2 | Total |
|---|---|---|---|---|---|---|---|
| BUTLER | 46.00 | 39.00 | 74.00 | **100.00** | 22.00 | 24.00 | 37.00 |
| ReAct | 66.67 | 41.94 | 91.03 | 80.95 | 55.56 | 35.29 | 61.94 |
| AdaPlanner | **100.00** | **96.77** | 95.65 | **100.00** | **100.00** | 47.06 | 91.79 |
| Reflexion | **100.00** | 90.32 | 82.61 | 90.48 | **100.00** | 94.12 | 92.54 |
| RAFA | **100.00** | **96.77** | **100.00** | **100.00** | **100.00** | **100.00** | **99.25** |

Table 2: ALFWorld results (success rates %).

## 5.2 ALFWORLD

ALFWorld [54] is an interactive environment for embodied agent simulations, which encompasses 134 household tasks in six overall categories (Table 2). We use `gpt-3` (`text-davinci-003`).

**Setup.** We emulate the tree-search algorithm to plan ($B = 2$). RAFA invokes `Critic` to evaluate the completed portion of the desired goal and switches to a new policy after 20 consecutive failures.

**Result.** RAFA outperforms various existing frameworks (right figure). The better performance of `AdaPlanner` at the initial episode is attributed to a handcrafted set of programs for rejecting suboptimal candidate trajectories, which is challenging to construct without the domain knowledge of a specific task. One such example is the PickTwo category.

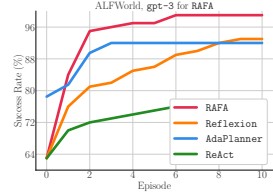

## 5.3 BLOCKSWORLD

BlocksWorld [23] is a rearrangement puzzle. For the RAFA algorithm, we use the `Vicuna` [79] model and emulate the MCTS algorithm to plan (see Figure 16 in Appendix). RAFA achieves superior success rates across multiple `Vicuna` versions (Figure 4). Comparisons with `CoT` and `RAP` demonstrate how the learning subroutine improves the planning optimality.

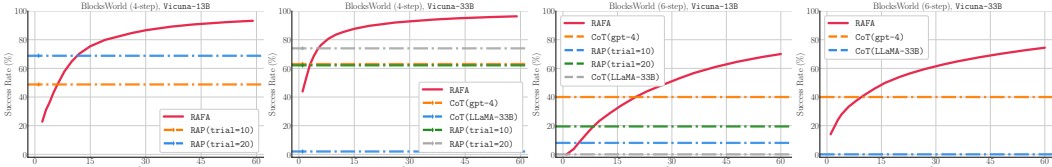

Figure 4: Sample efficiency on BlocksWorld (4 and 6 are the minimum numbers of steps for solving a specific task). `CoT` is prompted by four in-context examples.

## 5.4 TIC-TAC-TOE

Tic-Tac-Toe [7] is a competitive game where the X and O sides take turns to place marks. RAFA invokes `Model` to simulate the transition and opponent dynamics (see Figure 17 in Appendix).

**Setup.** We use `gpt-4` and emulate the tree-search algorithm to plan ($B \in \{3, 4\}$). RAFA switches to a new policy when (a) the predicted state differs from the observed one, (2) the predicted action of opponents differs from the observed one, or (3) `Critic` gives the wrong prediction of the game status. Here, X has an asymmetric advantage (winning surely if played properly).

**Result.** RAFA (playing O) matches and beats `gpt-4` for $T = 5$ and $T = 7$ (Table 3), although O is destined to lose. The ablation study ($B = 3$ versus $B = 4$) illustrates how the planning suboptimality affects the sample efficiency (Figure 5).

| O\X | gpt-4 |
|---|---|
| gpt-4 | 90%, 0%, **10%** |
| RAFA($T=1$) | 90%, 0%, **10%** |
| RAFA($T=5$) | 50%, 0%, **50%** |
| RAFA($T=7$) | 0%, 0%, **100%** |

Table 3: Tic-Tac-Toe Results. We set $B = 4$ and report the winning rate of X, the tie rate, and the winning rate of O.

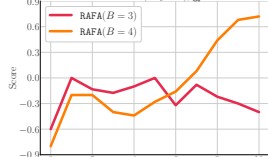

Figure 5: Sample efficiency on Tic-Tac-Toe (0 means tie).

## 6 CONCLUSIONS

In this paper, we establish the LLM-RL correspondence and propose a principled framework RAFA for orchestrating reasoning and acting, which achieves provable sample efficiency guarantees in autonomous LLM agents for the first time. RAFA's outstanding empirical performance underscores its potential for autonomous and adaptive decision-making in various complex environments, which we remain for future work.

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

CONTENTS

## APPENDIX A    MORE LITERATURE

**Large Language Model (LLM) and In-Context Learning (ICL).** LLMs [42, 9, 24, 13, 39, 60] display notable reasoning abilities. A pivotal aspect of reasoning is the ICL ability [33, 44, 52, 38, 5, 29, 18, 61, 32, 3], which allows LLMs to solve a broad range of tasks with only a few in-context examples instead of finetuning parameters on a specific dataset. We focus on harnessing the ICL ability of LLMs to optimize actions in the real world, which is crucial to autonomous LLM agents. In particular, we build on a recent line of work [69, 77, 78, 62, 68, 27, 30, 57] that attributes the ICL ability to implicit Bayesian inference, i.e., an implicit mechanism that enables LLMs to infer a latent concept from those in-context examples, which is verified both theoretically and empirically. In RAFA, the latent concept is the transition and reward models (model) of the unknown environment or/and the value function (critic), which is inferred from the memory buffer in the learning subroutine.

**Reinforcement Learning (RL) under a Bayesian Framework.** We build on a recent line of work on the infinite-horizon [1, 17, 65, 80, 81, 12, 14, 22, 49] and Bayesian [56, 40, 46, 45, 47, 36] settings of RL. The infinite-horizon setting allows RAFA to interact with the external environment continuously without resetting to an initial state, while the Bayesian setting allows RAFA to invoke the ICL ability of LLMs to form an updated posterior of the unknown environment. RL operates in a numerical system, where rewards and transitions are defined by scalars and probabilities, and trains actors and critics on the collected feedback iteratively. We focus on emulating the actor-model or actor-critic update in RL through an internal mechanism of reasoning on top of LLMs, which allows data and actions to be tokens in a linguistic system while bypassing the explicit update of parameters in model-based RL [14, 22, 49]. In particular, the learning and planning subroutines of RAFA emulate the posterior update and various planning algorithms in RL. Moreover, RAFA orchestrates reasoning (learning and planning) and acting following the principled approach in RL, i.e., (re)planning a future trajectory over a long horizon ("reason for future") at the new state and taking the initial action of the planned trajectory ("act for now"). As a result, RAFA inherits provable sample efficiency guarantees from RL.

APPENDIX B   MORE ALGORITHMS

We present the RL counterpart of RAFA in Section 4 in the following algorithm.

---

**Algorithm 3** Reason for future, act for now (RAFA): The RL counterpart.

---

1: **input**: An $\epsilon$-optimal planner $\mathtt{PL}^\epsilon$, which returns an $\epsilon$-optimal policy that maximizes the value function up to an $\epsilon$ accuracy (Definition 4.2), and the prior distribution $p_0$ of the parameter $\theta$.
2: **initialization**: Sample the initial state $s_0 \sim \rho$, set $t = 0$, and initialize the memory buffer $\mathcal{D}_0 = \varnothing$.
3: **for** $k = 0, 1, \ldots,$ **do**
4:     Set $t_k \leftarrow t$.
5:     **repeat**
6:         Update posterior $p_{t_k}$ given memory $\mathcal{D}_{t_k}$, sample parameter $\theta_t \sim p_{t_k}$,
        obtain policy $\pi_t \leftarrow \mathtt{PL}^\epsilon(\theta_t)$, and sample action $a_t \leftarrow \pi_t(s_t)$.       ("reason for future")
7:         Execute action $a_t$ to receive reward $r_t$ and state $s_{t+1}$ from environment.       ("act for now")
8:         Update memory $\mathcal{D}_{t+1} \leftarrow \mathcal{D}_t \cup \{(s_t, a_t, s_{t+1}, r_t)\}$.
9:         Set $t \leftarrow t + 1$.
10:    **until** $H_{t_k} - H_t > \log 2$.  (the switching condition is satisfied, where $H_{t_k}$ and $H_t$ are defined in (4.1))
11: **end for**

---

Depending on the specific configuration of the state and action spaces (continuous versus discrete) and the transition and reward models (stochastic versus deterministic), we may choose to emulate the tree-search algorithm, the value iteration algorithm, the random shooting algorithm, or the MCTS algorithm. All of them allow RAFA to achieve provable sample efficiency guarantees as long as they satisfy a specific requirement of optimality (Definition 4.2). For illustration, we emulate the beam-search algorithm (an advanced version of the tree-search algorithm) in Algorithm 4 and the MCTS algorithm in Algorithm 5. For the theoretical discussion, we present the value iteration algorithm in Algorithm 6.

---

**Algorithm 4** The LLM learner-planner (LLM-LR-PL): A beam-search example (for the deterministic case).

---

1: **input**: The memory buffer $\mathcal{D}$, the initial state $s$, the proposal width $L$, the search breadth $B$, and the search depth $U$.
2: **initialization**: Initialize the state array $\mathcal{S}_0 \leftarrow \{s\}$ and the action array $\mathcal{A}_0 \leftarrow \varnothing$.
                 ——————————————— (the learning subroutine) ———————————————
3: Set `Model` as an LLM instance prompted to use $\mathcal{D}$ as contexts to generate the next state.
4: Set `Critic` as an LLM instance prompted to use $\mathcal{D}$ as contexts to estimate the value function.
                 ——————————————— (the planning subroutine) ———————————————
5: Set `Elite` as an LLM instance prompted to use $\mathcal{D}$ as contexts to generate multiple candidate actions.
6: **for** $u = 0, \ldots, U$ **do**
7:     For each current state $s_u$ in $\mathcal{S}_u$, invoke `Elite` to generate $L$ candidate actions.
8:     For each candidate action $a_u^{(\ell)}$, invoke `Model` to generate the next state $s_{u+1}^{(\ell)}$ and the received reward $r_u^{(\ell)}$.
9:     For each resulting tuple $(s_u, a_u^{(\ell)}, s_{u+1}^{(\ell)}, r_u^{(\ell)})$, invoke `Critic` to evaluate the expected cumulative future reward $\widehat{Q}(s_u, a_u^{(\ell)}) \leftarrow r_u^{(\ell)} + \gamma \widehat{V}(s_{u+1}^{(\ell)})$, where $\widehat{V}$ is given by `Critic`.
10:    Select $B$ best tuples $(s_u, a_u^{(\ell)}, s_{u+1}^{(\ell)})$ with the highest value $\widehat{Q}(s_u, a_u^{(\ell)})$ and write them to $\mathcal{S}_u \times \mathcal{A}_u \times \mathcal{S}_{u+1}$.
11: **end for**
12: For $B$ preserved rollouts in $\mathcal{S}_0 \times \mathcal{A}_0 \times \cdots \times \mathcal{S}_U \times \mathcal{A}_U \times \mathcal{S}_{U+1}$, invoke `Critic` to evaluate the expected cumulative future reward $\sum_{u=0}^{U} \gamma^u r_u^{(b)} + \gamma^{U+1} \widehat{V}(s_{U+1}^{(b)})$ and select the best one $(s_0^\dagger, a_0^\dagger, \ldots, s_U^\dagger, a_U^\dagger, s_{U+1}^\dagger)$, where $\widehat{V}$ is given by `Critic` and $s_0^\dagger = s$.
13: **output**: The initial action $a_0^\dagger$ of the selected rollout.

---

---

**Algorithm 5** LLM learner-planner (`LLM-PL`) for `RAFA`: A Monte-Carlo tree-search example (for the stochastic case).

---

1: **input**: The memory buffer $\mathcal{D}$, the initial state $s$, the proposal width $L$, $L'$, and the expansion budget $E$.
2: **initialization**: Initialize the root node $n \leftarrow s$ and the child function $c(\cdot) \leftarrow \varnothing$.
———————————————— (the learning subroutine) ————————————————
3: Set `Model` as an LLM instance prompted to use $\mathcal{D}$ as contexts to generate the next state.
4: Set `Critic` as an LLM instance prompted to use $\mathcal{D}$ as contexts to estimate the value function.
———————————————— (the planning subroutine) ————————————————
5: Set `Elite` as an LLM instance prompted to use $\mathcal{D}$ as contexts to generate multiple candidate actions.
6: **for** $e = 0, \ldots, E$ **do**
7:     Set $s_e \leftarrow n$.
8:     **while** $s_e$ is not a leaf node, i.e., $c(s_e) \neq \varnothing$, **do**
9:         Invoke `Critic` to evaluate the expected cumulative future reward and select the child node $a_e$ in $c(s_e)$ with the highest value $\widehat{Q}(s_e, a_e)$.
10:         Set $s_e$ as a child node in $c(a_e)$.
11:     **end while**
12:     For the current state $s_e$, invoke `Elite` to generate $L$ candidate actions.
13:     Write each candidate action $a_e^{(\ell)}$ to $c(s_e)$, i.e., $c(s_e) \leftarrow \{a_e^{(\ell)}\}_{\ell=1}^{L}$.
14:     For each candidate action $a_e^{(\ell)}$, invoke `Model` to sample $L'$ next states.
15:     Write each next state $s_e^{(\ell,\ell')}$ to $c(a_e^{(\ell)})$, i.e., $c(a_e^{(\ell)}) \leftarrow \{s_e^{(\ell,\ell')}\}_{\ell'=1}^{L'}$.
16:     For each generated state $s_e^{(\ell,\ell')}$, invoke `Critic` to evaluate the expected cumulative future reward and update the estimated value $\widehat{V}$ for all ancestor nodes.         (Optional)
17: **end for**
18: Set $s_0^{\dagger} \leftarrow n$ and $i \leftarrow 0$.
19: **while** $s_i^{\dagger}$ is not a leaf node, i.e., $c(s_i^{\dagger}) \neq \varnothing$, **do**
20:     Invoke `Critic` to evaluate the expected cumulative future reward and select the child node $a_{i+1}^{\dagger}$ in $c(s_i^{\dagger})$ with the highest value $\widehat{Q}(s_i^{\dagger}, a_i^{\dagger})$.
21:     Set $s_{i+1}^{\dagger}$ as a child node in $c(a_i^{\dagger})$ and $i \leftarrow i + 1$.
22: **end while**
23: **output**: The initial action $a_0^{\dagger}$ of the selected rollout $(s_0^{\dagger}, a_0^{\dagger}, \ldots, s_i^{\dagger}, a_i^{\dagger})$.

---

We also present the value iteration algorithm (Algorithm 6) with a truncated horizon $U$, i.e., a finite length of the lookahead window as the $\epsilon$-optimal planner in Algorithm 3. The following proposition ensures that Algorithm 6 satisfies Definition 4.2.

**Proposition B.1.** Algorithm 6 is an $\epsilon$-optimal planner as long as we set $U \geq \lceil \log((2L/(\epsilon(1 - \gamma))))/\log(1/\gamma) \rceil + 1$ and any value function is upper bounded by $L \geq 0$.

*Proof.* See Appendix B.1 for a detailed proof. $\square$

---

**Algorithm 6** $\epsilon$-Optimal planner: The value iteration algorithm with a truncated horizon.

---

1: **input**: The model parameter $\theta$ and the truncated horizon $U$.
2: **initialization**: Set the value function $V_\theta^{(U)}(\cdot) \leftarrow 0$.
3: **for** $u = U - 1, \ldots, 1$ **do**
4:     Set the value function $V_\theta^{(u)}(\cdot) \leftarrow \max_{a \in \mathcal{A}} Q_\theta^{(u)}(\cdot, a)$, where $Q_\theta^{(u)}(\cdot, \cdot) \leftarrow r_\theta(\cdot, \cdot) + \gamma (P_\theta V_\theta^{(u+1)})(\cdot, \cdot)$.
5: **end for**
6: **output**: The greedy policy $\pi(\cdot) = \arg\max_{a \in \mathcal{A}} Q_\theta^{(1)}(\cdot, a)$.

---

## B.1 PROOF OF PROPOSITION B.1

*Proof of Proposition B.1.* We prove that Algorithm 6 satisfies Definition 4.2, where $U$ is dependent on $\epsilon$. Let $\pi_\theta(\theta)$ be the output policy of Algorithm 6 for the given $\theta$. For notational simplicity, we

denote $\max_{s \in \mathcal{S}}$ and $\max_{a \in \mathcal{A}}$ as $\max_s$ and $\max_a$. Then we have

$$\max_s \left[ V_\theta^{\mathrm{PL}^\star(\theta)}(s) - V_\theta^{\pi_\theta}(s) \right] \le \underbrace{\max_s \left[ V_\theta^{\mathrm{PL}^\star(\theta)}(s) - \max_a Q_\theta^{(1)}(s,a) \right]}_{\text{term (A)}}$$

$$+ \underbrace{\max_s \left[ \max_a Q_\theta^{(1)}(s,a) - V_\theta^{\pi_\theta}(s) \right]}_{\text{term (B)}}, \qquad (\text{B.1})$$

where $Q_\theta^{(1)}$ is defined in Algorithm 6.

**Analysis of Term (A).** For any $1 \le u < U$ and $s \in \mathcal{S}$, we have

$$V_\theta^{\mathrm{PL}^\star(\theta)}(s) - \max_a Q_\theta^{(u)}(s,a) = \max_a Q_\theta^{\mathrm{PL}^\star(\theta)}(s,a) - \max_a Q_\theta^{(u)}(s,a)$$

$$\le \max_a \left| Q_\theta^{\mathrm{PL}^\star(\theta)}(s) - Q_\theta^{(u)}(s,a) \right|$$

$$\le \gamma \cdot \max_a \left| \mathbb{E}_{s' \sim P_\theta(\cdot \,|\, s,a)} \left[ V_\theta^{\mathrm{PL}^\star(\theta)}(s') - \max_{a'} Q_\theta^{(u+1)}(s',a') \right] \right|$$

$$\le \gamma \cdot \max_{s'} \left| V_\theta^{\mathrm{PL}^\star(\theta)}(s') - \max_{a'} Q_\theta^{(u+1)}(s',a') \right|, \qquad (\text{B.2})$$

where the first equality uses the definition of $\mathrm{PL}^\star$ in Section 2, the first inequality uses the contraction property of the $\max$ operator, and the second equality uses the Bellman equation in (E.7) and Algorithm 6. By induction, we have

$$\text{term (A)} \le \gamma^{U-1} \max_s \cdot \left| V_\theta^{\mathrm{PL}^\star(\theta)}(s) - \max_a Q_\theta^{(U)}(s,a) \right|$$

$$\le \gamma^{U-1} L, \qquad (\text{B.3})$$

where the last equality uses $Q_\theta^{(U)} = 0$ and the fact that any value function is upper bounded by $L$.

**Analysis of Term (B).** Recall that $\pi_\theta(s) = \arg\max_a Q_\theta^{(1)}(s,a)$ in Algorithm 6. We have

$$\text{term (B)} = \max_s \left[ Q_\theta^{(1)}(s, \pi_\theta(s)) - Q_\theta^{\pi_\theta}(s, \pi_\theta(s)) \right]. \qquad (\text{B.4})$$

Let

$$\epsilon^\dagger = \max_s \left[ Q_\theta^{(1)}(s, \pi_\theta(s)) - r_\theta(s, \pi_\theta(s)) - \gamma (P_\theta V_\theta^{(1)})(s, \pi_\theta(s)) \right]. \qquad (\text{B.5})$$

We have

$$Q_\theta^{(1)}(s, \pi_\theta(s)) - Q_\theta^{\pi_\theta}(s, \pi_\theta(s)) \le \epsilon^\dagger + r_\theta(s, \pi_\theta(s)) + \gamma (P_\theta V_\theta^{(1)})(s, \pi_\theta(s))$$

$$- r_\theta(s, \pi_\theta(s)) - \gamma (P_\theta V_\theta^{\pi_\theta})(s, \pi_\theta(s))$$

$$= \epsilon^\dagger + \gamma \cdot \mathbb{E}_{s' \sim P_\theta(\cdot \,|\, s, \pi_\theta(s))} \left[ Q_\theta^{(1)}(s', \pi_\theta(s')) - Q_\theta^{\pi_\theta}(s', \pi_\theta(s')) \right]$$

$$\le \epsilon^\dagger + \gamma \cdot \max_s \left| Q_\theta^{(1)}(s, \pi_\theta(s)) - Q_\theta^{\pi_\theta}(s, \pi_\theta(s)) \right|. \qquad (\text{B.6})$$

Here, the first inequality is based on the definition of $\epsilon^\dagger$ and the Bellman equation in (E.7) in the proof of Theorem 4.4, where the first equality uses fact that $Q_\theta^{(1)}(s, \pi_\theta(s)) = \max_a Q_\theta^{(1)}(s, a) = V_\theta^{(1)}(s)$ for all $s \in \mathcal{S}$. Taking the $\sup_{s \in \mathcal{S}}$ in the left-hand side of (B.6), we obtain

$$\max_s \big| Q_\theta^{(1)}(s, \pi_\theta(s)) - Q_\theta^{\pi_\theta}(s, \pi_\theta(s)) \big| \le \epsilon^\dagger + \gamma \cdot \max_s \big| Q_\theta^{(1)}(s, \pi_\theta(s)) - Q_\theta^{\pi_\theta}(s, \pi_\theta(s)) \big|,$$

which implies

$$\begin{aligned}
\text{term (B)} &\le \max_s \big| Q_\theta^{(1)}(s, \pi_\theta(s)) - Q_\theta^{\pi_\theta}(s, \pi_\theta(s)) \big| \\
&\le \epsilon^\dagger / (1 - \gamma).
\end{aligned} \tag{B.7}$$

Meanwhile, the convergence analysis of the value iteration algorithm in [59] gives

$$\max_{s,a} \big| Q_\theta^{(1)}(s, a) - Q_\theta^{(2)}(s, a) \big| \le \gamma^{U-2} \max_{s,a} \big| Q_\theta^{(U-1)}(s, a) - Q_\theta^{(U)}(s, a) \big|,$$

which implies

$$\max_{s,a} \big| Q_\theta^{(1)}(s, a) - Q_\theta^{(2)}(s, a) \big| \le \gamma^{U-1} L.$$

We have

$$\begin{aligned}
\epsilon^\dagger &= \max_s \big[ Q_\theta^{(1)}(s, \pi_\theta(s)) - r_\theta(s, \pi_\theta(s)) - \gamma (P_\theta V_\theta^{(2)})(s, \pi_\theta(s)) \\
&\qquad\qquad + \gamma \mathbb{E}_{s' \sim P_\theta(\cdot \,|\, s, \pi_\theta(s))} \big[ V_\theta^{(1)}(s') - V_\theta^{(2)}(s') \big] \big] \\
&= \gamma \cdot \mathbb{E}_{s' \sim P_\theta(\cdot \,|\, s, \pi_\theta(s))} \big[ V_\theta^{(1)}(s') - V_\theta^{(2)}(s') \big] \\
&= \gamma \cdot \mathbb{E}_{s' \sim P_\theta(\cdot \,|\, s, \pi_\theta(s))} \big[ \max_a Q_\theta^{(1)}(s', a) - \max_a Q_\theta^{(2)}(s', a) \big] \\
&\le \gamma \cdot \mathbb{E}_{s' \sim P_\theta(\cdot \,|\, s, \pi_\theta(s))} \big[ \max_a \big| Q_\theta^{(1)}(s', a) - Q_\theta^{(2)}(s', a) \big| \big] \\
&\le \gamma^{U-1} L, \tag{B.8}
\end{aligned}$$

where the first and third equalities are based on Algorithm 6, the second last inequality uses the contraction property of the $\max$ operator.

Plugging (B.3), (B.7), and (B.8) into (B.1), we obtain

$$\max_s \big[ V_\theta^{\text{PL}^\star(\theta)}(s) - V_\theta^{\pi_\theta}(s) \big] \le \gamma^{U-1} L + \gamma^{U-1} L / (1 - \gamma) \le \epsilon,$$

for $U \ge \lceil \log((2L/(\epsilon(1-\gamma)))) / \log(1/\gamma) \rceil + 1$. Thus, we prove Proposition B.1. $\qquad\square$

## APPENDIX C    DISCUSSION ON DEFINITIONS

### C.1    DISCUSSION ON DEFINITION 4.2

**Definition 4.2** ($\epsilon$-Optimality of Planner). A planning algorithm $\mathtt{PL}^\epsilon : \Theta \mapsto \Pi$ is an $\epsilon$-optimal planner if $\max_{s \in \mathcal{S}}[V_\theta^{\mathtt{PL}^\star(\theta)}(s) - V_\theta^{\mathtt{PL}^\epsilon(\theta)}(s)] \leq \epsilon$ for all $\theta \in \Theta$.

As a special case of Definition 4.2, we present the value iteration algorithm (Algorithm 6) in Appendix B, where we use a truncated horizon $U$, i.e., a finite length of the lookahead window. Alternatively, we may choose to emulate the tree-search algorithm, the random shooting algorithm, or the Monte-Carlo tree-search algorithm. In the tree-search example (Lines 5-11 in Algorithm 2), $\epsilon$ decreases as the search breadth $B$ and depth $U$ increase. Note that, as long as we emulate an $\epsilon$-optimal planner, we are able to establish provable sample efficiency guarantees.

### C.2    DISCUSSION ON DEFINITION 4.3

**Definition 4.3** (Information Ratio). The information ratio $\Gamma_{t^\dagger}(\delta)$ is the smallest number for which, if we have $H_{t^\dagger} - H_t \leq \log 2$, then it holds for all $t \in \{t^\dagger, \ldots, T\}$ with probability at least $1 - \delta$ that

$$\left| (r_{\theta^\star} - r_{\theta_{t^\dagger}})(s_t, a_t) + \left( (P_{\theta^\star} - P_{\theta_{t^\dagger}}) V_t \right)(s_t, a_t) \right| \leq \Gamma_{t^\dagger}(\delta) \cdot \sqrt{I(\theta; \xi_{t+1} | \mathcal{D}_t)}, \qquad \text{(C.1)}$$

where $\theta^\star$ is the data-generating parameter and $\theta_{t^\dagger} \sim p_{t^\dagger}$ is a sampled parameter.

Definition 4.3 quantifies the estimation error of the sampled parameter $\theta_{t^\dagger}$ in terms of approximating the data-generating parameter $\theta^\star$. To achieve this, we use the information gain $I(\theta; \xi_{t+1} | \mathcal{D}_t)$ as a benchmarking quantity. Intuitively, the information ratio $\Gamma_{t^\dagger}(\delta)$ characterizes how exploration reduces uncertainty. As long as $\Gamma_{t^\dagger}(\delta)$ is finite, collecting a better dataset $\mathcal{D}_{t^\dagger}$ reduces the estimation error of $\theta_{t^\dagger}$ by decreasing $I(\theta; \xi_{t+1} | \mathcal{D}_t)$. To see this, we consider the limiting case that $\mathcal{D}_{t^\dagger} \subseteq \mathcal{D}_t$ is already sufficiently informative. As the collected feedback $\xi_{t+1}$ provides little new information, we know that $I(\theta; \xi_{t+1} | \mathcal{D}_t)$ is small, which implies that the estimation error of $\theta_{t^\dagger}$ is small. Note that $\Gamma_{t^\dagger}(\delta)$ is an intrinsic property of the underlying MDP, which is independent of the agent design. In Appendix F, we study a general class of MDPs, namely linear kernel MDPs [6, 10, 81], and prove that the information ratio is $\mathcal{O}(d \cdot \log(dT/\delta))$, where $d$ is the feature dimension.

## APPENDIX D  LLM SAMPLES POSTERIOR

*Verification of Assumption 4.1.*  We prove that Assumption 4.1 holds for a specific parameterization of $(P_\theta, r_\theta)$ under three regularity conditions, namely (a) LLMs are trained to replicate the pretraining distribution, which is assumed in [48, 66, 69] to simplify the statistical analysis, (b) the pretraining dataset is generated through a Bayesian mechanism with a latent concept, which is a simplified version of the latent variable model in [69] and resembles that in [62], and (c) LLMs are able to parameterize an implicit Bayesian inference mechanism, which is proved in [77, 78] for the attention architecture. Note that, if Assumption 4.1 holds approximately, the regret analysis can be relaxed to accommodate the additional error in the posterior distribution.

**Assumption D.1** (Model Class).  We assume that $(P_\theta, r_\theta)$ is parameterized under the following conditions.

Factorization:  For any $(s', s, a, g) \in \mathcal{S} \times \mathcal{S} \times \mathcal{A} \times \mathcal{G}$, it holds that

$$P_\theta(s' \mid s, a; g) = \phi_g(s', s, a)^\top f_g(\theta),$$

where $\phi_g : \mathcal{S} \times \mathcal{S} \times \mathcal{A} \mapsto \mathbb{R}^{d_g}$ is the feature mapping, $f_g : \Theta \mapsto \mathbb{R}^{d_g}$ is the parameter transformation, and $d_g$ is the feature dimension.

No Redundant Feature:  For any $g \in \mathcal{G}$, there exists a set $\{(s'_{(i)}, s_{(i)}, a_{(i)})\}_{i=1}^{d_g} \subseteq \mathcal{S} \times \mathcal{S} \times \mathcal{A}$ such that the set $\{\phi_g(s'_{(i)}, s_{(i)}, a_{(i)})\}_{i=1}^{d_g}$ forms a basis of $\mathbb{R}^{d_g}$.

Identification:  For two probability distributions $q_1$ and $q_2$ over $\Theta$, if it holds for all $g \in \mathcal{G}$ that

$$\mathbb{E}_{\theta \sim q_1}[f_g(\theta)] = \mathbb{E}_{\theta \sim q_2}[f_g(\theta)],$$

then we have $q_1 = q_2$.

The factorization condition in Assumption D.1 is common in the RL literature [71, 10, 81]. As long as $\phi_g$ has sufficient expressive power, $(P_\theta, r_\theta)$ is able to represent a broad range of models.

The redundancy condition states that $\phi_g$ does not have any redundant dimension. In other words, we are able to construct a basis of $\mathbb{R}^{d_g}$ by choosing $d_g$ different tuples $(s', s, a)$. We provide the following example. Let $\mathcal{G}$ be $\{1, \ldots, |\mathcal{G}|\}$ and $d_g = d$ for all $g \in \mathcal{G}$. We assume that there exists a surjective mapping $k : \mathcal{S} \times \mathcal{S} \times \mathcal{A} \mapsto \mathbb{B}^{|\mathcal{G}|d}$, where $\mathbb{B}^{|\mathcal{G}|d}$ is the unit ball under the Euclidean norm in $\mathbb{R}^{|\mathcal{G}|d}$. Let $p_g : \mathbb{R}^{|\mathcal{G}|d} \mapsto \mathbb{R}^d$ be a projection operator, which outputs the $((g-1)d+1)$-th to $gd$-th entries of the input vector. We define $\phi_g$ as $p_g \circ k$. Since $k$ is surjective, we are able to construct $\{(s'_{(i)}, s_{(i)} a_{(i)})\}_{i=1}^{|\mathcal{G}|d}$ as the tuple set in association with the canonical basis of $\mathbb{R}^{|\mathcal{G}|d}$. As a result, $\{\phi_g(s'_{(i)}, s_{(i)}, a_{(i)})\}_{i=(g-1)d+1}^{gd}$ forms the canonical basis of $\mathbb{R}^{d_g}$.

The identification condition ensures that the parameter distribution has a one-to-one correspondence to its mean embedding [55, 37]. For example, the Gaussian distribution is determined by its first-order and second-order moments.

The following assumption specifies the data-generating process, which is a simplified version of the latent variable model in [69] and resembles that in [62].

**Assumption D.2** (Data Generation). We assume that each individual rollout $\{(s_t, a_t, r_t)\}_{t=0}^{\infty}$ in the pretraining dataset $\mathcal{D}$ is generated through the following process.

Instruct: Choose a specific instruction $g \in \mathcal{G}$ and sample an underlying parameter $\theta^\star$ from the prior distribution $p_0$. Store $g$ in $\mathcal{D}$.

Initialize: Sample the initial state $s_0$ from the initial distribution $\rho$.

Interact: At the $t$-th step, take an action $a_t$ given the history $\{(s_i, a_i, s_{i+1}, r_i)\}_{i=0}^{t}$ and receive the reward $r_t$ and the next state $s_{t+1}$, where $r_t = r_{\theta^\star}^g(s_t, a_t)$ and $s_{t+1} \sim P_{\theta^\star}^g(\cdot \,|\, s_t, a_t)$.

The following assumption states that LLMs parameterize an implicit Bayesian inference mechanism, which is proved in [77, 78] for the attention architecture.

**Assumption D.3** (Parameterization of Implicit Bayesian Inference). We assume that LLMs parameterize the following mechanism, which is summarized by the conditional distribution $P_{\mathrm{LLM}}(s' \,|\, s, a, \mathcal{D}, g)$.

Infer: Sample a latent concept $\theta \sim P_{\mathrm{LLM}}(\cdot \,|\, \mathcal{D})$, where $P_{\mathrm{LLM}}$ is the conditional distribution in association with the given LLM.

Generate: Generate the next state $s'$ and the received reward $r$ via $s' \sim P_\theta(\cdot \,|\, s, a; g)$ and $r = r_\theta(s, a; g)$ given a specific instruction $g$ and the current state $s$ and action $a$.

The following assumption states that LLMs are trained to replicate the pretraining distribution, which covers a wide variety of $\mathcal{D}$ and $g$. The same assumption is employed by [48, 66, 69, 30] to simplify the statistical analysis. Let $\mathbb{P}_{\mathrm{Data}}$ be the pretraining distribution.

**Assumption D.4** (Pretraining Consistency). We assume that

$$P_{\mathrm{LLM}}(s' \,|\, s, a, \mathcal{D}, g) = \mathbb{P}_{\mathrm{Data}}(s' \,|\, s, a, \mathcal{D}, g)$$

for all $\mathcal{D}$ and $(s', s, a, g) \in \mathcal{S} \times \mathcal{S} \times \mathcal{A} \times \mathcal{G}$.

In the following, we prove that Assumption 4.1 holds. By the implicit Bayesian inference mechanism in Assumption D.3, we have

$$P_{\mathrm{LLM}}(s' \,|\, s, a, \mathcal{D}, g) = \int P_\theta(s' \,|\, s, a; g) \cdot P_{\mathrm{LLM}}(\theta \,|\, \mathcal{D}) \mathrm{d}\theta, \tag{D.1}$$

for all $\mathcal{D}$ and $(s', s, a, g) \in \mathcal{S} \times \mathcal{S} \times \mathcal{A} \times \mathcal{G}$. By the data-generating mechanism in Assumption D.2, we have

$$
\begin{aligned}
\mathbb{P}_{\text{Data}}(s' \,|\, s, a, \mathcal{D}, g) &= \int P_{\theta^\star}(s' \,|\, s, a; g) \cdot \mathbb{P}_{\text{Data}}(\theta^\star \,|\, s, a, \mathcal{D}, g) \mathrm{d}\theta^\star \\
&= \int P_{\theta^\star}(s' \,|\, s, a; g) \cdot \mathbb{P}_{\text{Data}}(\theta^\star \,|\, \mathcal{D}) \mathrm{d}\theta^\star
\end{aligned}
\tag{D.2}
$$

for all $\mathcal{D}$ and $(s', s, a, g) \in \mathcal{S} \times \mathcal{S} \times \mathcal{A} \times \mathcal{G}$. Let $\mathbb{P}_{\text{Data}}(\cdot \,|\, \mathcal{D})$ be the posterior distribution of the underlying parameter $\theta^\star$. By the pretraining consistency in Assumption D.4, we have

$$
\int P_\theta(s' \,|\, s, a; g) \cdot P_{\text{LLM}}(\theta \,|\, \mathcal{D}) \mathrm{d}\theta = \int P_\theta(s' \,|\, s, a; g) \cdot \mathbb{P}_{\text{Data}}(\theta \,|\, \mathcal{D}) \mathrm{d}\theta
\tag{D.3}
$$

for all $\mathcal{D}$ and $(s', s, a, g) \in \mathcal{S} \times \mathcal{S} \times \mathcal{A} \times \mathcal{G}$. By the factorization condition in Assumption D.1, we transform (D.3) into

$$
\phi_g(s', s, a)^\top \int f_g(\theta) \cdot P_{\text{LLM}}(\theta \,|\, \mathcal{D}) \mathrm{d}\theta = \phi_g(s', s, a)^\top \int f_g(\theta) \cdot \mathbb{P}_{\text{Data}}(\theta \,|\, \mathcal{D}) \mathrm{d}\theta
\tag{D.4}
$$

for all $\mathcal{D}$ and $(s', s, a, g) \in \mathcal{S} \times \mathcal{S} \times \mathcal{A} \times \mathcal{G}$. By the redundancy condition in Assumption D.1 and the property of a basis, we have

$$
\begin{aligned}
\int f_g(\theta) \cdot P_{\text{LLM}}(\theta \,|\, \mathcal{D}) \mathrm{d}\theta &= \sum_{i=1}^{d_g} \Big( \phi_g(s'_{(i)}, s_{(i)}, a_{(i)})^\top \int f_g(\theta) \cdot P_{\text{LLM}}(\theta \,|\, \mathcal{D}) \mathrm{d}\theta \Big) \cdot \phi_g(s'_{(i)}, s_{(i)}, a_{(i)}) \\
&= \sum_{i=1}^{d_g} \Big( \phi_g(s'_{(i)}, s_{(i)}, a_{(i)})^\top \int f_g(\theta) \cdot \mathbb{P}_{\text{Data}}(\theta \,|\, \mathcal{D}) \mathrm{d}\theta \Big) \cdot \phi_g(s'_{(i)}, s_{(i)}, a_{(i)}) \\
&= \int f_g(\theta) \cdot \mathbb{P}_{\text{Data}}(\theta \,|\, \mathcal{D}) \mathrm{d}\theta,
\end{aligned}
\tag{D.5}
$$

which implies

$$
\mathbb{E}_{\theta \sim P_{\text{LLM}}(\cdot \,|\, \mathcal{D})} \big[ f_g(\theta) \big] = \mathbb{E}_{\theta \sim \mathbb{P}_{\text{Data}}(\cdot \,|\, \mathcal{D})} \big[ f_g(\theta) \big]
$$

for all $g \in \mathcal{G}$ and $\mathcal{D}$. By the identifiability condition in Assumption D.1, we have

$$
P_{\text{LLM}}(\theta \,|\, \mathcal{D}) = \mathbb{P}_{\text{Data}}(\theta \,|\, \mathcal{D})
$$

which shows that the implicit Bayesian inference mechanism in Assumption D.3 matches the data-generating process in Assumption D.2. Thus, we verify Assumption 4.1. □

To numerically verify Assumption 4.1, we conduct three experiments on Bayesian linear regression, the ALFWorld environment, and multi-arm bandit, respectively. Since the predictions given by posterior sampling have the contraction property and the variance heterogeneity, we examine these two properties of LLM predictions in multiple settings to study if the distribution of LLM predictions matches the one given by posterior sampling.

**Bayesian Linear Regression.** We test if GPT-3 can make an implicit Bayesian inference for the linear regression problem from the perspectives of unbiasedness, contraction property, and Variance heterogeneity of Predictions.

- **Unbiasedness and contraction property.** The unbiasedness and contraction property ensure LLMs give a responsible answer given enough prompts and we consider a one-dimensional Bayesian linear regression setting to verify them. We first generate i.i.d. data $\{(x_i, y_i)\}_{i=1}^N$ from $x_i \sim \mathcal{N}(0.8, 1)$, $y_i \sim \mathcal{N}(\beta x_i, 0.04)$, where we take $\beta = 0.5$. Then, we query GPT-3 API to obtain 200 predictions when $x_{\text{pred}} = 1$ (we set temperature as 1.0 and top-p as 0.8, where these parameter configurations can be found in [21]). Specifically, the prompts are "The following data is for linear regression: $(x_1, y_1), (x_2, y_2), \ldots, (x_N, y_N), (1.000000,$". Then, we plot the histogram of the LLM-generated predictions and the density function of the calculated $y_{\text{pred}}$ by posterior sampling, where we choose the prior of the coefficient $\beta$ as $\mathcal{N}(0, 0.5)$. We report the results in Figure 6. We find that the histogram of LLM predictions approximately matches the theoretical distribution of the prediction given by the posterior sampling. Also, with an increasing number of generated data, the variance of LLM predictions decays, which shows the contraction property.

- **Variance heterogeneity of Predictions.** The variance heterogeneity of predictions helps LLMs give more diverse answers when the current prompt contains less relevant information. RAFA relies on the diversity of LLM predictions to guarantee the exploration for those states less explored, hence we wish the variance of LLM predictions for the less explored states is higher. We consider a two-dimensional Bayesian linear regression setting to study the variance heterogeneity of LLM predictions. We first generate i.i.d. data $\mathcal{D} = \{((x_i^{(1)}, x_i^{(2)}), y_i)\}_{i=1}^N$ from $x_i = (x_i^{(1)}, x_i^{(2)})^\top \sim \mathcal{N}((0.8, 0.8)^\top, \text{diag}(4, 0.1))$, $y_i \sim \mathcal{N}(x_i^\top \beta, 0.04)$, where we set $N = 60$ and $\beta = (0.25, 0.25)^\top$. Then, we query GPT-3 API to obtain 200 predictions given the covariate $x_{\text{pred}}$ (we set temperature as 1.0 and top-p as 0.8, where these parameter configurations can be found in [21]). Specifically, the prompts are "The following data is for linear regression: $((x_1^{(1)}, x_1^{(2)}), y_1), (x_2^{(1)}, x_2^{(2)}), y_2), \ldots, ((x_N^{(1)}, x_N^{(2)}), y_N), ((x_{\text{pred}}^{(1)}, x_{\text{pred}}^{(2)})$". Theoretically, the variance of predictions given by posterior sampling is $\sigma^2 + (x_{\text{pred}}^\top \Sigma^{-1} x_{\text{pred}})$, where $\Sigma^{-1} = \sigma^{-2} \Lambda_0 + \sum_{i=1}^N (x_i - \bar{x})(x_i - \bar{x})^\top$ is the covariance matrix of the posterior distribution of $\beta$ and $\bar{x}$ denotes the mean of $\{x_i\}_{i=1}^N$. Here, we choose the prior distribution of $\beta$ is $\mathcal{N}((0, 0)^\top, \Lambda_0)$ with $\Lambda_0 = \text{diag}(0.5, 0.5)$. To study the variance heterogeneity of predictions, we conduct a single-value decomposition on $\Sigma^{-1}$ to obtain its eigenvalues $\{\lambda_1, \lambda_2\}$ and the corresponding unit eigenvectors $\{v_1, v_2\}$, where we let $\lambda_1 < \lambda_2$ (if $\lambda_1 = \lambda_2$, we regenerate the data). Note that

$$
\begin{aligned}
\mathbb{V}[y_{\text{pred}} \,|\, \mathcal{D}, x_{\text{pred}} = \bar{x} + \lambda_1^{-1/2} \cdot v_1] &= \sigma^2 + \|v_1\|_2^2 \\
&> \sigma^2 + (\lambda_2/\lambda_1) \cdot \|v_2\|_2^2 \\
&= \mathbb{V}[y_{\text{pred}} \,|\, \mathcal{D}, x_{\text{pred}} = \bar{x} + \lambda_1^{-1/2} \cdot v_2].
\end{aligned}
$$

  Thus, we examine the sample variance of LLM predictions given different covariates and present the results in Table 4. Results show that LLM predictions reveal variance heterogeneity, which helps RAFA explore the unknown environment.

**ALFWorld.** We also test if the LLM performs an implicit Bayesian inference in the ALFWorld environment.

- **Contraction property.** We begin by examining the contraction property in the ALFWorld environment. Specifically, the model LLM is prompted to predict the observation of a certain action "go to countertop 1". The prediction result is provided in Figure 7. We find that as more action-observation paris are added to the GPT-3 prompt when timestep

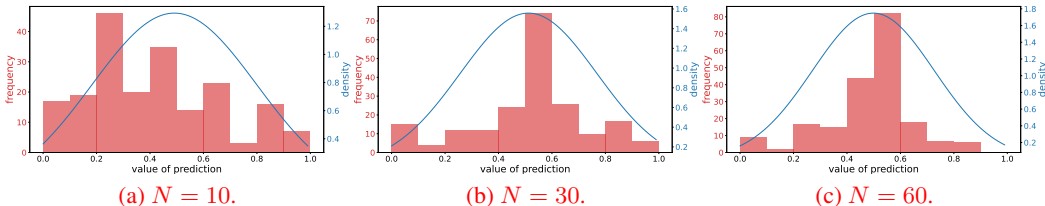

(a) $N = 10$.     (b) $N = 30$.     (c) $N = 60$.

Figure 6: Histograms of LLM predictions and theoretical distribution of the prediction given by the posterior sampling in the Bayesian linear regression problem.

| $x_{\text{pred}}$ | Sample Variance of LLM Predictions |
|---|---|
| $\bar{x}$ | 0.00496 |
| $\bar{x} + \lambda_1^{-1/2} \cdot v_2$ | 0.00752 |
| $\bar{x} + \lambda_1^{-1/2} \cdot v_1$ | 0.00864 |

Table 4: Sample variance of LLM Predictions on different $x_{\text{pred}}$.

increases, the number of distinct prediction responses decreases, i.e., the model LLM has decreasing uncertainty of what objects a certain place has after observing the locations of objects at other places. This corresponds to a decreasing entropy and contraction of the posterior distribution, which supports Assumption 4.1.

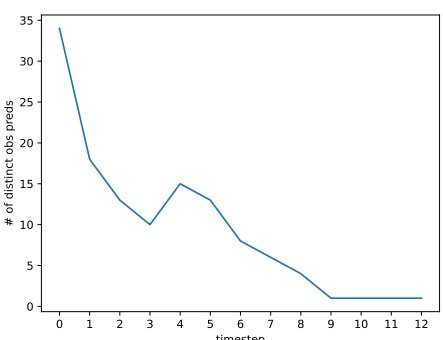

Figure 7: Contraction property in the ALFWorld environment.

Figure 8: Variance heterogeneity of predictions in the ALFWorld environment.

- **Variance heterogeneity of Predictions.** To show the variance heterogeneity of the predictions, we prompt the model LLM to predict the next observations when taking actions that the agent has less relevant information. Specifically, in the trajectory that the agent has never explored any dining table, we prompt the model LLM to predict the observation after taking the action "go to dining table 1" at different timesteps. The result is shown in Figure 8. We can observe that the number of distinct predictions of taking this action is much larger the "go to countertop 1" action where similar locations such as "countertop 2" and "countertop 3" are explored by the agent in this episode. This indicates the variance heterogeneity of the predictions.

**Contextual Bandit.** We also conduct an experiment on a two-arm contextual bandit setting to compare the performance of LLMs, Thompson sampling (posterior sampling), and Bayesian model averaging. At each step, the agent receives a context $x$ sampled from $\mathcal{N}(0.8, 1)$ and pulls the first arm or the second one. If she pulls the first arm, she will receive a reward sampled from $\mathcal{N}(\beta_1 x, 0.04)$ with $\beta_1 = 0.1$. If she pulls the second arm, she will receive a reward sampled from $\mathcal{N}(\beta_2 x, 0.04)$ with $\beta_2 = 0.5$. For the LLM agent, we use the GPT-4 model and prompt it to output the best arm given the history of the context-reward pair when pulling different arms at different timesteps.

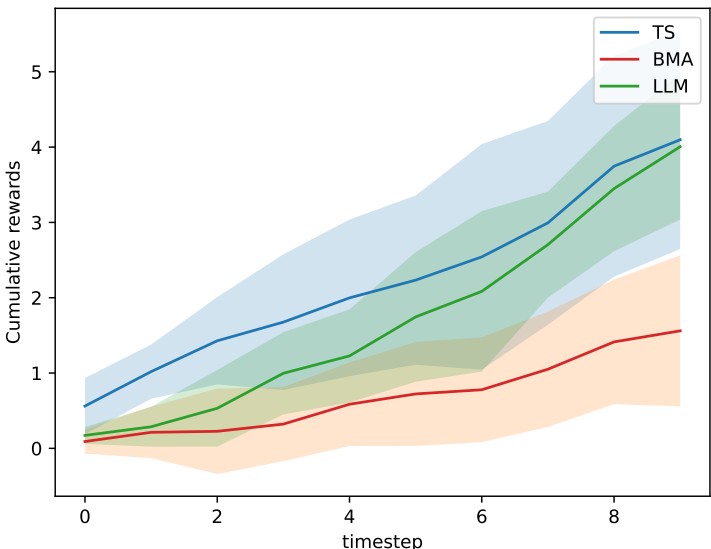

Figure 9: Performance of LLMs, Thompson sampling (posterior sampling), and Bayesian model averaging in a two-arm bandit setting. Results are averaged under five random seeds, and the shaded area represents a confidence band with one standard deviation.

Specifically, the LLM agent is first given the context and asked to determine the next arm to pull. The context state [context], the pulled arm [arm], and the received reward [reward] are all added to the prompt of the LLM. The system prompt we use is "You are a linear bandit expert. Output the best arm based on the observed data. Output format: Best arm: [arm 0 or 1]. DO NOT generate unnecessary text and always end with 0 or 1." and the user prompt we use is "Consider the following contextual linear bandit scenario with two arms. The following data is observed: state: [context], arm: [arm], reward: [reward]; state: [context], arm: [arm], reward: [reward]; …. Will you try arm 0 or arm 1 in the next step? Remember the exploration-exploitation tradeoff when generating the response.". For Thompson sampling, the agent first samples a parameter $\widetilde{\beta}_i$ from the posterior distribution of $\beta_i$ and samples an estimated reward $\widetilde{y}_i$ given $\widetilde{\beta}_i$ for all $i \in \{1, 2\}$. She will only pull the first arm if $\widetilde{y}_1 > \widetilde{y}_2$. For Bayesian averaging, the agent chooses the mean of the posterior of $\beta_i$ as the parameter estimate $\widehat{\beta}_i$ and samples an estimated reward $\widehat{y}_i$ given $\widehat{\beta}_i$ for all $i \in \{1, 2\}$. She will also only pull the first arm if $\widehat{y}_1 > \widehat{y}_2$. The prior distribution of $\beta_i$ in both Thompson sampling and Bayesian model averaging is chosen as $\mathcal{N}(0, 1)$ for all $i \in \{1, 2\}$ We report the cumulative rewards across 10 steps in Figure. 9.

From Figure 9, we observe that the cumulative reward obtained by the LLMs approximately matches the Thompson sampling, both of which perform better than the Bayesian model averaging. This implies that both LLMs and Thompson sampling can make a more effective exploration, while Bayesian model averaging is easier to get trapped in the suboptimality. This close relationship between the Thompson sampling algorithm and LLMs also supports Assumption 4.1.

## APPENDIX E  PROOF OF THEOREM 4.4

*Proof of Theorem 4.4.* We specify the terminating condition for Algorithm 3 (and Algorithm 1). Let $(K - 1)$ be the total number of switches until $t$ reaches $(T - 1)$. Let $t_K = T$. At the $(T - 1)$-th step, Algorithm 3 samples $\theta_{T-1}$ from $p_{t_{K-1}}$ and executes $a_{T-1} = \pi_{T-1}(s_{T-1})$ , where we have $\pi_{T-1} = \mathtt{PL}^\epsilon(\theta_{T-1})$. Upon receiving $r_{T-1}$ and $s_T$ from the external environment, Algorithm 3 updates $\mathcal{D}_T = \{(s_t, a_t, s_{t+1}, r_t)\}_{t=0}^{T-1}$ and terminates. Throughout the following proof, we denote

the upper bound of the value function as $L$, i.e., $\max_{\theta \in \Theta, \pi \in \Pi, s \in \mathcal{S}} |V_\theta^\pi(s)| \leq L$, for a positive constant $L$.

Recall that the Bayesian regret $\mathfrak{R}(T)$ and the optimal planner $\mathtt{PL}^\star(\theta)$ are defined in Section 2. By the tower property of the conditional expectation, we have

$$
\begin{aligned}
\mathfrak{R}(T) &= \mathbb{E}\Big[\sum_{k=0}^{K-1} \sum_{t=t_k}^{t_{k+1}-1} V_{\theta^\star}^{\pi^\star}(s_t) - V_{\theta^\star}^{\pi_t}(s_t)\Big] \\
&= \mathbb{E}\Big[\sum_{k=0}^{K-1} \mathbb{E}\Big[\mathbb{E}\Big[\sum_{t=t_k}^{t_{k+1}-1} V_{\theta^\star}^{\pi^\star}(s_t) - V_{\theta^\star}^{\pi_t}(s_t)\Big|\mathcal{F}_k\Big]\Big]\Big] \\
&= \mathbb{E}\Big[\sum_{k=0}^{K-1} \mathbb{E}\Big[\mathbb{E}\Big[\sum_{t=t_k}^{t_{k+1}-1} V_{\theta^\star}^{\mathtt{PL}^\star(\theta^\star)}(s_t) - V_{\theta^\star}^{\pi_t}(s_t)\Big|\mathcal{F}_k\Big]\Big]\Big] \\
&= \mathbb{E}\Big[\sum_{k=0}^{K-1} \mathbb{E}\Big[\mathbb{E}\Big[\sum_{t=t_k}^{t_{k+1}-1} V_{\theta_t}^{\mathtt{PL}^\star(\theta_t)}(s_t) - V_{\theta^\star}^{\pi_t}(s_t)\Big|\mathcal{F}_k\Big]\Big]\Big] \\
&= \mathbb{E}\Big[\sum_{k=0}^{K-1} \sum_{t=t_k}^{t_{k+1}-1} V_{\theta_t}^{\mathtt{PL}^\star(\theta_t)}(s_t) - V_{\theta^\star}^{\pi_t}(s_t)\Big],
\end{aligned}
\tag{E.1}
$$

where we define

$$
\mathcal{F}_k = \sigma\Big(\big\{(s_t, a_t, s_{t+1}, r_t)\big\}_{t=0}^{t_k-1}\Big)
\tag{E.2}
$$

and the fourth equality uses the fact that $\theta^\star \,|\, \mathcal{F}_k$ and $\theta_t \,|\, \mathcal{F}_k$ follows the same distribution for all $t_k \leq t < t_{k+1}$. Note that the parameter sequence $\{\theta_t\}_{t=t_k}^{t_{k+1}-1}$ is sampled from $p_{t_k}$ independently and identically, while we have $\pi_t = \mathtt{PL}^\epsilon(\theta_t)$, where $\mathtt{PL}^\epsilon$ is an $\epsilon$-optimal planner in Definition 4.2. Let $\theta^k = \theta_{t_k}$ and $\pi^k = \pi_{t_k}$. We are able to rewrite (E.1) as

$$
\mathfrak{R}(T) = \mathbb{E}\Big[\sum_{k=0}^{K-1} \sum_{t=t_k}^{t_{k+1}-1} V_{\theta^k}^{\mathtt{PL}^\star(\theta^k)}(s_t) - V_{\theta^\star}^{\pi^k}(s_t)\Big].
\tag{E.3}
$$

Meanwhile, Definition 4.2 gives

$$
V_{\theta^k}^{\mathtt{PL}^\star(\theta^k)}(s_t) - V_{\theta^k}^{\pi^k}(s_t) \leq \epsilon.
\tag{E.4}
$$

Here, we use $\pi^k = \mathtt{PL}^\epsilon(\theta^k)$.

To connect (E.3) and (E.4), we introduce the regret decomposition. For notational simplicity, we denote by

$$
(B_\theta V)(s, a) = r_\theta(s, a) + \gamma(P_\theta V)(s, a)
\tag{E.5}
$$

for all $(s, a) \in \mathcal{S} \times \mathcal{A}$ and $V$.

**Lemma E.1** (Regret Decomposition). For Algorithm 3, it holds that

$$
\frac{1-\gamma}{\gamma} \cdot \mathbb{E}\Big[\sum_{k=0}^{K-1} \mathbb{E}_{\pi^k}\Big[ \sum_{t=t_k}^{t_{k+1}-1} V_{\theta^k}^{\pi^k}(s_t) - V_{\theta^\star}^{\pi^k}(s_t)\Big]\Big]
$$

$$
= \mathbb{E}\Big[\sum_{k=0}^{K-1} \mathbb{E}_{\pi^k}\Big[ \sum_{t=t_k}^{t_{k+1}-1} (B_{\theta^k} V_{\theta^k}^{\pi^k})(s_t, a_t) - (B_{\theta^\star} V_{\theta^k}^{\pi^k})(s_t, a_t)\Big]\Big]
$$

$$
\underbrace{\phantom{= \mathbb{E}\Big[\sum_{k=0}^{K-1} \mathbb{E}_{\pi^k}\Big[ \sum_{t=t_k}^{t_{k+1}-1} (B_{\theta^k} V_{\theta^k}^{\pi^k})(s_t, a_t) - (B_{\theta^\star} V_{\theta^k}^{\pi^k})(s_t, a_t)\Big]\Big]}}_{\text{term (A): information gain}}
$$

$$
+ \mathbb{E}\Big[\sum_{k=0}^{K-1} \mathbb{E}_{\pi^k}\Big[ \big(V_{\theta^k}^{\pi^k}(s_{t_{k+1}}) - V_{\theta^\star}^{\pi^k}(s_{t_{k+1}})\big) - \big(V_{\theta^k}^{\pi^k}(s_{t_k}) - V_{\theta^\star}^{\pi^k}(s_{t_k})\big)\Big]\Big], \qquad (E.6)
$$

$$
\underbrace{\phantom{+ \mathbb{E}\Big[\sum_{k=0}^{K-1} \mathbb{E}_{\pi^k}\Big[ \big(V_{\theta^k}^{\pi^k}(s_{t_{k+1}}) - V_{\theta^\star}^{\pi^k}(s_{t_{k+1}})\big) - \big(V_{\theta^k}^{\pi^k}(s_{t_k}) - V_{\theta^\star}^{\pi^k}(s_{t_k})\big)\Big]\Big]}}_{\text{term (B): value inconsistency}}
$$

where $\mathbb{E}_{\pi^k}$ is taken with respect to the state-action sequence following $s_{t+1} \sim P_{\theta^\star}(\cdot \,|\, s_t, a_t)$ and $a_t = \pi^k(s_t)$ for all $t_k \le t < t_{k+1}$, while $\mathbb{E}$ is taken with respect to the prior distribution $p_0$ of $\theta^\star$, the posterior distribution $p_{t_k}$ of $\theta^k$, and the iterative update of $\pi^k$. Here, $\pi^k$ is determined by $\theta^k$ and $p_{t_k}$ is conditioned on $\mathcal{D}_{t_k} = \{(s_t, a_t, s_{t+1}, r_t)\}_{t=0}^{t_k-1}$, which is generated by $\theta^\star$ and Algorithm 3.

*Proof of Lemma E.1.* The Bellman equation [59] connects $Q_\theta^\pi(s, a)$ and $V_\theta^\pi(s)$ by

$$
Q_\theta^\pi(s, a) = r_\theta(s, a) + \gamma \left(P_\theta V_\theta^\pi\right)(s, a), \quad V_\theta^\pi(s) = Q_\theta^\pi(s, \pi(s)). \qquad (E.7)
$$

By (E.5), we rewrite (E.7) as $Q_\theta^\pi(s, a) = (B_\theta V_\theta^\pi)(s, a)$. For the left-hand side of (E.6), we have

$$
\mathbb{E}\Big[\sum_{k=0}^{K-1} \mathbb{E}_{\pi^k}\Big[ \sum_{t=t_k}^{t_{k+1}-1} V_{\theta^k}^{\pi^k}(s_t) - V_{\theta^\star}^{\pi^k}(s_t)\Big]\Big]
$$

$$
= \gamma \cdot \mathbb{E}\Big[\sum_{k=0}^{K-1} \mathbb{E}_{\pi^k}\Big[ \sum_{t=t_k}^{t_{k+1}-1} (B_{\theta^k} V_{\theta^k}^{\pi^k})(s_t, a_t) - (B_{\theta^\star} V_{\theta^\star}^{\pi^k})(s_t, a_t)\Big]\Big]
$$

$$
= \gamma \cdot \mathbb{E}\Big[\sum_{k=0}^{K-1} \mathbb{E}_{\pi^k}\Big[ \sum_{t=t_k}^{t_{k+1}-1} (B_{\theta^k} V_{\theta^k}^{\pi^k})(s_t, a_t) - (B_{\theta^\star} V_{\theta^k}^{\pi^k})(s_t, a_t)\Big]\Big]
$$

$$
\underbrace{\phantom{= \gamma \cdot \mathbb{E}\Big[\sum_{k=0}^{K-1} \mathbb{E}_{\pi^k}\Big[ \sum_{t=t_k}^{t_{k+1}-1} (B_{\theta^k} V_{\theta^k}^{\pi^k})(s_t, a_t) - (B_{\theta^\star} V_{\theta^k}^{\pi^k})(s_t, a_t)\Big]\Big]}}_{\text{term (A)}}
$$

$$
+ \gamma \cdot \mathbb{E}\Big[\sum_{k=0}^{K-1} \mathbb{E}_{\pi^k}\Big[ \sum_{t=t_k}^{t_{k+1}-1} (B_{\theta^\star} V_{\theta^k}^{\pi^k})(s_t, a_t) - V_{\theta^k}^{\pi^k}(s_{t+1})\Big]\Big]
$$

$$
\underbrace{\phantom{+ \gamma \cdot \mathbb{E}\Big[\sum_{k=0}^{K-1} \mathbb{E}_{\pi^k}\Big[ \sum_{t=t_k}^{t_{k+1}-1} (B_{\theta^\star} V_{\theta^k}^{\pi^k})(s_t, a_t) - V_{\theta^k}^{\pi^k}(s_{t+1})\Big]\Big]}}_{\text{term (C1)}}
$$

$$
+ \gamma \cdot \mathbb{E}\Big[\sum_{k=0}^{K-1} \mathbb{E}_{\pi^k}\Big[ \sum_{t=t_k}^{t_{k+1}-1} V_{\theta^\star}^{\pi^k}(s_{t+1}) - (B_{\theta^\star} V_{\theta^\star}^{\pi^k})(s_t, a_t)\Big]\Big]
$$

$$
\underbrace{\phantom{+ \gamma \cdot \mathbb{E}\Big[\sum_{k=0}^{K-1} \mathbb{E}_{\pi^k}\Big[ \sum_{t=t_k}^{t_{k+1}-1} V_{\theta^\star}^{\pi^k}(s_{t+1}) - (B_{\theta^\star} V_{\theta^\star}^{\pi^k})(s_t, a_t)\Big]\Big]}}_{\text{term (C2)}}
$$

$$
+ \gamma \cdot \mathbb{E}\Big[\sum_{k=0}^{K-1} \mathbb{E}_{\pi^k}\Big[ \sum_{t=t_k}^{t_{k+1}-1} V_{\theta^k}^{\pi^k}(s_{t+1}) - V_{\theta^\star}^{\pi^k}(s_{t+1})\Big]\Big], \qquad (E.8)
$$

$$
\underbrace{\phantom{+ \gamma \cdot \mathbb{E}\Big[\sum_{k=0}^{K-1} \mathbb{E}_{\pi^k}\Big[ \sum_{t=t_k}^{t_{k+1}-1} V_{\theta^k}^{\pi^k}(s_{t+1}) - V_{\theta^\star}^{\pi^k}(s_{t+1})\Big]\Big]}}_{\text{term (D)}}
$$

where the first equality uses $a_t = \pi^k(s_t)$. Since we have

$$(P_{\theta^\star} V)(s_t, a_t) = \mathbb{E}_{s_{t+1} \sim P_{\theta^\star}(\cdot \mid s_t, a_t)} \big[ V(s_{t+1}) \big],$$

terms (C1) and (C2) in (E.8) are zero. Meanwhile, term (D) in (E.8) satisfies

$$\text{term (D)} = \mathbb{E}\Big[ \sum_{k=0}^{K-1} \mathbb{E}_{\pi^k} \Big[ \sum_{t=t_k}^{t_{k+1}-1} \big( V_{\theta^k}^{\pi^k}(s_t) - V_{\theta^\star}^{\pi^k}(s_t) \big) \Big] \Big]$$

$$+ \mathbb{E}\Big[ \underbrace{\sum_{k=0}^{K-1} \mathbb{E}_{\pi^k} \Big[ \big( V_{\theta^k}^{\pi^k}(s_{t_{k+1}}) - V_{\theta^\star}^{\pi^k}(s_{t_{k+1}}) \big) - \big( V_{\theta^k}^{\pi^k}(s_{t_k}) - V_{\theta^\star}^{\pi^k}(s_{t_k}) \big) \Big]}_{\text{term (B)}} \Big], \quad \text{(E.9)}$$

where term (B) is defined in Lemma E.1. Rearranging (E.8) and (E.9), we prove Lemma E.1. $\qquad \square$

We characterize terms (A) and (B) in (E.6). Let $\mathcal{E}$ be the high-probability event in Definition 4.3. For the information gain in term (A), we have

$$\text{term (A)} \leq \mathbb{E}\Big[ \sum_{k=0}^{K-1} \mathbb{E}_{\pi^k} \Big[ \mathbb{1}_{\mathcal{E}} \cdot \sum_{t=t_k}^{t_{k+1}-1} (B_{\theta^k} V_{\theta^k}^{\pi^k})(s_t, a_t) - (B_{\theta^\star} V_{\theta^k}^{\pi^k})(s_t, a_t) \Big] \Big] + L\delta T$$

$$\leq \mathbb{E}\Big[ \sum_{k=0}^{K-1} \mathbb{E}_{\pi^k} \Big[ \sum_{t=t_k}^{t_{k+1}-1} \Gamma_{t_k}(\delta) \cdot \sqrt{I(\theta; \xi_{t+1} \mid \mathcal{D}_t)} \Big] \Big] + L\delta T$$

$$\leq \sup_{t^\dagger < T} \Gamma_{t^\dagger}(\delta) \cdot \mathbb{E}\Big[ \sum_{k=0}^{K-1} \mathbb{E}_{\pi^k} \Big[ \sum_{t=t_k}^{t_{k+1}-1} \sqrt{I(\theta; \xi_{t+1} \mid \mathcal{D}_t)} \Big] \Big] + L\delta T$$

$$\leq \sup_{t^\dagger < T} \Gamma_{t^\dagger}(\delta) \cdot \mathbb{E}\Big[ \sqrt{T} \cdot \Big( \sum_{t=0}^{T-1} I(\theta; \xi_{t+1} \mid \mathcal{D}_t) \Big)^{1/2} \Big] + L\delta T$$

$$= \sup_{t^\dagger < T} \Gamma_{t^\dagger}(\delta) \cdot \mathbb{E}\big[ \sqrt{T(H_0 - H_T)} \big] + L\delta T. \quad \text{(E.10)}$$

The first inequality uses the fact that the value function is upper bounded by $L$ and $\mathbb{P}(\mathcal{E}) \geq 1 - \delta$ in Definition 4.3. The last inequality invokes the Cauchy-Schwarz inequality. The last equality is derived from $I(\theta; \xi_{t+1} \mid \mathcal{D}_t) = H_t - H_{t+1}$. We explain the second inequality as follows. For any $0 \leq k < K$, we consider $t^\dagger = t_k$ and $\theta^k = \theta_{t_k}$, which is sampled from $p_{t_k}$ given $\mathcal{D}_{t_k}$, and the value function sequence $\{V_t\}_{t=t_k}^{t_{k+1}-1} = \{V_{\theta^k}^{\pi^k}\}_{t=t_k}^{t_{k+1}-1}$, which is adapted to $\{\sigma(\mathcal{D}_t)\}_{t=t_k}^{t_{k+1}-1}$ since $\mathcal{D}_{t_k} \subseteq \mathcal{D}_t$ for all $t_k \leq t < t_{k+1}$. Correspondingly, we have

$$\big| (r_{\theta^\star} - r_{\theta_{t_k}})(s_t, a_t) + \big( (P_{\theta^\star} - P_{\theta_{t_k}}) V_t \big)(s_t, a_t) \big| \leq \Gamma_{t_k}(\delta) \cdot \sqrt{I(\theta; \xi_{t+1} | \mathcal{D}_t)}.$$

Hence, we obtain the second inequality in (E.10). As the switching condition in Algorithm 3 implies $H_{t_k} - H_{t_{k+1}} \geq \log 2$, we have

$$H_0 - H_{t_{K-1}} = \sum_{k=0}^{K-2} H_{t_k} - H_{t_{k+1}} \geq (K-1) \cdot \log 2,$$

which implies

$$K - 1 \leq (H_0 - H_{t_{K-1}})/\log 2 \leq (H_0 - H_T)/\log 2. \tag{E.11}$$

Recall that the value function is upper bounded by $L$. For the value inconsistency in term (B), we have

$$\text{term (B)} \leq (4L/\log 2) \cdot \mathbb{E}[H_0 - H_T] + 4L. \tag{E.12}$$

Plugging (E.10) and (E.12) into Lemma E.1, we have

$$\frac{1-\gamma}{\gamma} \cdot \mathbb{E}\Big[\sum_{k=0}^{K} \mathbb{E}_{\pi^k}\Big[\sum_{t=t_k}^{t_{k+1}-1} V_{\theta^k}^{\pi^k}(s_t) - V_{\theta^\star}^{\pi^k}(s_t)\Big]\Big]$$

$$\leq L\delta T + \sup_{t^\dagger < T} \Gamma_{t^\dagger}(\delta) \cdot \mathbb{E}\big[\sqrt{T(H_0 - H_T)}\big] + (4L/\log 2) \cdot \mathbb{E}[H_0 - H_T] + 4L. \tag{E.13}$$

Combining (E.3), (E.4), and (E.13), we have

$$\mathfrak{R}(T) = \mathcal{O}\Big( \frac{\gamma \cdot \sup_{t^\dagger < T} \Gamma_{t^\dagger}(\delta) \cdot \mathbb{E}[\sqrt{H_0 - H_T}]}{1-\gamma} \cdot \sqrt{T}$$

$$+ \frac{\gamma \delta L}{1-\gamma} \cdot T + \epsilon \cdot T + \frac{\gamma L \cdot \mathbb{E}[H_0 - H_T]}{(1-\gamma)}\Big). \tag{E.14}$$

Since the reward function is bounded in $[0,1]$, we take $L = \sum_{t=0}^{\infty} \gamma^t = 1/(1-\gamma)$ and obtain

$$\mathfrak{R}(T) = \mathcal{O}\Big( \frac{\gamma \cdot \sup_{t^\dagger < T} \Gamma_{t^\dagger}(\delta) \cdot \mathbb{E}[\sqrt{H_0 - H_T}]}{1-\gamma} \cdot \sqrt{T} + \frac{\gamma \delta}{(1-\gamma)^2} \cdot T + \epsilon \cdot T + \frac{\gamma \cdot \mathbb{E}[H_0 - H_T]}{(1-\gamma)^2}\Big).$$

Thus, we prove Theorem 4.4. □

## APPENDIX F   LINEAR SPECIAL CASE

We specialize Algorithm 3 to a linear setting and characterize the Bayesian regret. In particular, we define a Bayesian variant of linear kernel MDPs [71, 70, 10, 81]. Here, $\mathbb{E}_{s' \sim P_\theta(\cdot \mid s,a)} V(s')$ is linear in a feature $\psi_V(s,a) \in \mathbb{R}^d$ for an arbitrary parameter $\theta \in \mathbb{R}^d$, while the prior and posterior distributions of the data-generating parameter $\theta^\star \in \mathbb{R}^d$ are Gaussian. Specifically, $\psi_V(s,a)$ maps the value function $V$ and the state-action pair $(s,a)$ to a $d$-dimensional vector. Recall that $\rho$ is the initial distribution of states, $t$ is the step index, and $T$ is the total number of steps. Also, $p_t$ is the posterior distribution at the $t$-th step.

**Definition F.1** (Bayesian Linear Kernel MDP [19, 71, 70, 10, 81]). A Bayesian linear kernel MDP $M$ satisfies

$$V(s_{t+1}) \mid s_t, a_t \sim \mathcal{N}(\psi_V(s_t, a_t)^\top \theta, 1)$$

for all $t \geq 0$, $(s_t, a_t) \in \mathcal{S} \times \mathcal{A}$, $s_{t+1} \sim P_\theta(\cdot \mid s_t, a_t)$, $\theta \in \mathbb{R}^d$, as well as all value function $V$. Here, $\psi_V(s, a)$ maps $V$ and $(s, a)$ to a $d$-dimensional vector, which satisfies $\|\psi_V(s, a)\|_2 \leq R$ for all $(s, a) \in \mathcal{S} \times \mathcal{A}$ and all $V$. Also, $M$ also satisfies $|\mathbb{E}_{s_0 \sim \rho} V(s_0)| \leq R$ for all $V$. Here, $R$ is a positive constant that is independent of $t$ and $T$. The prior distribution of the data-generating parameter $\theta^\star \in \mathbb{R}^d$ is $\mathcal{N}(0, \lambda I_d)$, where $\lambda$ is a positive constant. Here, $\psi_V$ is known and $\theta^\star$ is unknown. Without loss of generality, we assume that the reward function is deterministic and known.

Let $(K-1)$ be the total number of switches until $t$ reaches $(T-1)$. Let $t_K = T$. At the $(T-1)$-th step, Algorithm 3 samples $\theta_{T-1}$ from $p_{t_{K-1}}$ and executes $a_{T-1} = \pi_{T-1}(s_{T-1})$ , where we have $\pi_{T-1} = \mathrm{PL}^\epsilon(\theta_{T-1})$. Upon receiving $r_{T-1}$ and $s_T$ from the external environment, Algorithm 3 updates $\mathcal{D}_T = \{(s_t, a_t, s_{t+1}, r_t)\}_{t=0}^{T-1}$ and terminates. For notational simplicity, we define $V_t = V_{\theta^k}^{\pi^k}$ for all $t_k \leq t < t_{k+1}$ and obtain the closed form of $p_t$ as follows,

$$\theta \mid \mathcal{D}_t \sim \mathcal{N}(\widehat{\theta}_t; \Sigma_t^{-1}),$$

where

$$\widehat{\theta}_t = \left(\lambda I_d + \sum_{i=0}^{t-1} \psi_{V_i}(s_i, a_i) \psi_{V_i}(s_i, a_i)^\top\right)^{-1} \left(\sum_{i=0}^{t-1} \psi_{V_i}(s_i, a_i) V_i(s_{i+1})\right) \tag{F.1}$$

and

$$\Sigma_t = \lambda I_d + \sum_{i=0}^{t-1} \psi_{V_i}(s_i, a_i) \psi_{V_i}(s_i, a_i)^\top. \tag{F.2}$$

Hence, the posterior entropy is

$$H_t = \mathfrak{H}(p_t) = 1/2 \cdot \log(\det(\Sigma_t)) + d/2 \cdot (1 + \log(2\pi)). \tag{F.3}$$

We specialize the switching condition in Algorithm 3 as follows,

$$1/2 \cdot \log(\det(\Sigma_{t_k})) - 1/2 \cdot \log(\det(\Sigma_t)) > \log 2,$$

which is equivalent to $\det(\Sigma_{t_k}) > 4 \cdot \det(\Sigma_t)$. This switching condition is also similarly adopted in work for RL [81, 1]. As a result, we have

$$\det(\Sigma_{t_k}) \leq 4 \cdot \det(\Sigma_t) \tag{F.4}$$

for all $t_k \leq t < t_{k+1}$ and $k < K$.

To apply Theorem 4.4, we characterize the information gain $H_0 - H_T$ throughout the $T$ steps and the information ratio $\Gamma_{t^\dagger}(\delta)$, which is defined in Definition 4.3.

**Information Gain.** By the definition of $\Sigma_t$ in (F.2), we have $\log \det(\Sigma_0) = d \cdot \log \lambda$ and

$$
\begin{aligned}
\log \det(\Sigma_T) &= \log \det \Big( \lambda I_d + \sum_{t=0}^{T-1} \psi_{V_t}(s_t, a_t) \psi_{V_t}^\top(s_t, a_t) \Big) \\
&\leq d \cdot \log \Big( 1/d \cdot \mathrm{tr} \big( \lambda I_d + \sum_{t=0}^{T-1} \psi_{V_t}(s_t, a_t) \psi_{V_t}^\top(s_t, a_t) \big) \Big) \\
&= d \cdot \log \Big( 1/d \cdot \big( \lambda d + \sum_{t=0}^{T-1} \| \psi_{V_t}(s_t, a_t) \|_2^2 \big) \Big) \\
&\leq d \cdot \log(\lambda + TR^2/d)
\end{aligned}
\tag{F.5}
$$

almost surely. Here, the first inequality uses the relationship between the trace and the determinant of a square matrix, the second equality uses $\mathrm{tr}(a^\top b) = \mathrm{tr}(ba^\top)$ for two arbitrary vectors $a$ and $b$, and the last inequality uses the fact that $\| \psi_V(s, a) \|_2$ is upper bounded by $R$ for all $(s, a) \in \mathcal{S} \times \mathcal{A}$ and $V$. Hence, we have

$$
H_0 - H_T = \mathcal{O}\big( d \cdot \log(1 + TR^2/(d\lambda)) \big)
\tag{F.6}
$$

almost surely. By (E.11) in the proof of Theorem 4.4, we obtain

$$
K - 1 = \mathcal{O}\big( d \cdot \log(1 + TR^2/(d\lambda)) \big)
\tag{F.7}
$$

almost surely.

**Information Ratio.** The following proposition characterizes the information ratio $\Gamma_{t^\dagger}(\delta)$, which is defined in Definition 4.3.

**Proposition F.2.** The information ratio $\Gamma_{t^\dagger}(\delta)$ satisfies $\sup_{t^\dagger < T} \Gamma_{t^\dagger}(\delta) = \mathcal{O}(d \cdot \log(dT/\delta))$.

*Proof of Proposition F.2.* For any $t^\dagger$ and $t \geq t^\dagger$, $H_{t^\dagger} - H_t \leq \log 2$ implies

$$
(1/2) \cdot \log \det(\Sigma_{t^\dagger}) - (1/2) \cdot \log \det(\Sigma_t) \leq \log 2,
$$

which uses (F.3). Hence, we have

$$
\det(\Sigma_{t^\dagger}) \leq 4 \cdot \det(\Sigma_t).
\tag{F.8}
$$

By Definition F.1, we have

$$
\begin{aligned}
\big| ((P_{\theta^\star} - P_{\theta_{t^\dagger}}) V_t)(s_t, a_t) \big| &= (\theta^\star - \theta_{t^\dagger})^\top \psi_{V_t}(s_t, a_t) \\
&\leq \| \theta^\star - \theta_{t^\dagger} \|_{\Sigma_{t^\dagger}} \cdot \| \psi_{V_t}(s_t, a_t) \|_{\Sigma_{t^\dagger}^{-1}} \\
&\leq 4 \cdot \| \theta^\star - \theta_{t^\dagger} \|_{\Sigma_{t^\dagger}} \cdot \| \psi_{V_t}(s_t, a_t) \|_{\Sigma_t^{-1}},
\end{aligned}
\tag{F.9}
$$

where the first inequality invokes the Cauchy-Schwarz inequality and the second inequality is derived from Lemma 12 in [2] and (F.8). Since $\theta_{t^\dagger}$ and $\theta^\star$ share the posterior distribution $\mathcal{N}(\widehat{\theta}_{t^\dagger}, \Sigma_{t^\dagger}^{-1})$ given $\mathcal{D}_{t^\dagger}$, we know that $\| \theta - \widehat{\theta}_{t^\dagger} \|_{\Sigma_{t^\dagger}}^2$ follows $\mathcal{X}_d^2$, where $\theta \in \{\theta^\star, \theta_{t^\dagger}\}$ and $\mathcal{X}_d^2$ is the chi-square

distribution with $d$ degrees of freedom. By the tail behavior of the chi-square distribution [20], we have

$$\mathbb{P}\big(\|\theta^\star - \widehat{\theta}_{t\dagger}\|_{\Sigma_{t\dagger}}^2 > 2d \cdot \log(2d/\delta)\big) \le \delta/2 \tag{F.10}$$

and

$$\mathbb{P}\big(\|\theta_{t\dagger} - \widehat{\theta}_{t\dagger}\|_{\Sigma_{t\dagger}}^2 > 2d \cdot \log(2d/\delta)\big) \le \delta/2 \tag{F.11}$$

for a sufficiently large $d$ and a sufficiently small $\delta$. Hence, it holds with probability at least $1 - \delta$ that

$$\|\theta^\star - \theta_{t\dagger}\|_{\Sigma_{t\dagger}} \le \sqrt{2} \cdot \sqrt{\|\theta_{t\dagger} - \widehat{\theta}_{t\dagger}\|_{\Sigma_{t\dagger}}^2 + \|\theta^\star - \widehat{\theta}_{t\dagger}\|_{\Sigma_{t\dagger}}^2} = \mathcal{O}\big(\sqrt{d} \cdot \log(d/\delta)\big), \tag{F.12}$$

which uses the triangle inequality for $\|\cdot\|_{\Sigma_{t\dagger}}$ and the inequality $(a+b)^2 \le 2 \cdot (a^2 + b^2)$ for all $a, b \ge 0$. Recall that the information gain is defined as

$$I(\theta; \xi_{t+1} \,|\, \mathcal{D}_t) = H_t - H_{t+1}.$$

By (F.3), we have

$$
\begin{aligned}
I(\theta; \xi_{t+1} \,|\, \mathcal{D}_t) &= 1/2 \cdot \log\big(\det(\Sigma_{t+1})/\det(\Sigma_t)\big) \\
&= 1/2 \cdot \log(1 + \psi_{V_t}(s_t, a_t)^\top \Sigma_t^{-1} \psi_{V_t}(s_t, a_t)) \\
&\ge \log(1 + d)/(2d) \cdot \psi_{V_t}(s_t, a_t)^\top \Sigma_t^{-1} \psi_{V_t}(s_t, a_t) \\
&= \log(1 + d)/(2d) \cdot \|\psi_{V_t}(s_t, a_t)\|_{\Sigma_t^{-1}}^2, \tag{F.13}
\end{aligned}
$$

where the second equality uses the matrix determinant lemma and the first inequality uses the fact that $\log(1 + x)/x$ is an increasing function for $x \ge 0$ and

$$
\begin{aligned}
0 &\le \psi_{V_t}(s_t, a_t)^\top \Sigma_t^{-1} \psi_{V_t}(s_t, a_t) \\
&\le \psi_{V_t}(s_t, a_t)^\top \big(\psi_{V_t}(s_t, a_t)\psi_{V_t}(s_t, a_t)^\top\big)^{-1} \psi_{V_t}(s_t, a_t) \\
&= \mathrm{tr}\Big(\psi_{V_t}(s_t, a_t)\psi_{V_t}(s_t, a_t)^\top \big(\psi_{V_t}(s_t, a_t)\psi_{V_t}(s_t, a_t)^\top\big)^{-1}\Big) \\
&= d. \tag{F.14}
\end{aligned}
$$

Here, the first inequality uses the nonnegativity of a quadratic form, the first equality uses $\mathrm{tr}(a^\top b) = \mathrm{tr}(ba^\top)$ for two arbitrary vectors $a$ and $b$, and the second inequality uses (F.2). We obtain

$$
\begin{aligned}
4 \cdot \|\theta^\star - \theta_{t\dagger}\|_{\Sigma_{t\dagger}} \cdot \|\psi_{V_t}(s_t, a_t)\|_{\Sigma_t^{-1}} &\le 4\sqrt{2d}/\sqrt{\log(1 + d)} \cdot \|\theta^\star - \theta_{t\dagger}\|_{\Sigma_{t\dagger}} \cdot \sqrt{I(\theta; \xi_{t+1} \,|\, \mathcal{D}_t)} \\
&\le 4\sqrt{2(d+1)} \cdot \|\theta^\star - \theta_{t\dagger}\|_{\Sigma_{t\dagger}} \cdot \sqrt{I(\theta; \xi_{t+1} \,|\, \mathcal{D}_t)} \\
&= \mathcal{O}\big(\sqrt{d(d+1)} \cdot \log(d/\delta) \cdot \sqrt{I(\theta; \xi_{t+1} \,|\, \mathcal{D}_t)}\big) \tag{F.15}
\end{aligned}
$$

for all $t < T$ such that $H_{t^\dagger} - H_t \le \log 2$. Here, the first inequality uses (F.13), the second inequality uses $\log(1+x) \ge x/(1+x)$ for all $x \ge 0$, and the first equality uses (F.12). Setting $\delta' = \delta/T$ and applying the union bound over $0 \le t < T$, we obtain

$$\sup_{t^\dagger < T} \Gamma_{t^\dagger}(\delta') = \mathcal{O}\big(\sqrt{d(d+1)} \cdot \log(dT/\delta')\big)$$
$$= \mathcal{O}\big(d \cdot \log(dT/\delta')\big)$$

with probability at least $1 - \delta'$ for a sufficiently large enough $d$ and a sufficiently small $\delta'$. Thus, we prove Proposition F.2. $\qquad\square$

**Bayesian Regret.** Proposition F.2 enables us to specialize Theorem 4.4 in Bayesian linear kernel MDPs when any value function is upper bounded by $L$ [40, 45, 47, 36]. To remove the boundedness assumption, we present a refined analysis based on the proof of Theorem 4.4, which characterizes term (A) in Lemma E.1 through a conditional expectation. Recall that Bayesian linear kernel MDPs are defined in Definition F.1.

**Proposition F.3.** For Bayesian linear kernel MDPs, the Bayesian regret of Algorithm 3 satisfies

$$\mathfrak{R}(T) = \mathcal{O}\Big(\frac{\gamma d^{3/2} \cdot \log(TR^2/(d\lambda))}{1-\gamma} \cdot \sqrt{T} + \frac{\gamma \lambda^{1/2} d^{3/2} R}{1-\gamma} \cdot \log(TR^2/(d\lambda)) + \epsilon \cdot T\Big). \quad \text{(F.16)}$$

Proposition F.3 establishes the $\sqrt{T}$ regret of Algorithm 3 for an appropriate choice of the planning suboptimality $\epsilon$, e.g., $\epsilon = \mathcal{O}(1/\sqrt{T})$. Here, the first term on the right-hand side of (F.16) is the leading term.

### F.1 PROOF OF PROPOSITION F.3

*Proof of Proposition F.3.* Following the proof of Theorem 4.4, it suffices to bound

$$\frac{1-\gamma}{\gamma} \cdot \mathbb{E}\Big[\sum_{k=0}^{K-1} \mathbb{E}_{\pi^k}\Big[\sum_{t=t_k}^{t_{k+1}-1} V_{\theta^k}^{\pi^k}(s_t) - V_{\theta^\star}^{\pi^k}(s_t)\Big]\Big]$$

on the left-hand side of (E.6) in Lemma E.1. In particular, Lemma E.1 gives

$$\frac{1-\gamma}{\gamma} \cdot \mathbb{E}\Big[\sum_{k=0}^{K-1} \mathbb{E}_{\pi^k}\Big[\sum_{t=t_k}^{t_{k+1}-1} V_{\theta^k}^{\pi^k}(s_t) - V_{\theta^\star}^{\pi^k}(s_t)\Big]\Big]$$
$$= \underbrace{\mathbb{E}\Big[\sum_{k=0}^{K-1} \mathbb{E}_{\pi^k}\Big[\sum_{t=t_k}^{t_{k+1}-1} (B_{\theta^k} V_{\theta^k}^{\pi^k})(s_t, a_t) - (B_{\theta^\star} V_{\theta^k}^{\pi^k})(s_t, a_t)\Big]\Big]}_{\text{term (A): information gain}}$$
$$+ \underbrace{\mathbb{E}\Big[\sum_{k=0}^{K-1} \mathbb{E}_{\pi^k}\Big[\big(V_{\theta^k}^{\pi^k}(s_{t_{k+1}}) - V_{\theta^\star}^{\pi^k}(s_{t_{k+1}})\big) - \big(V_{\theta^k}^{\pi^k}(s_{t_k}) - V_{\theta^\star}^{\pi^k}(s_{t_k})\big)\Big]\Big]}_{\text{term (B): value inconsistency}}, \quad \text{(F.17)}$$

where $\mathbb{E}_{\pi^k}$ is taken with respect to the state-action sequence following $s_{t+1} \sim P_{\theta^\star}(\cdot \,|\, s_t, a_t)$ and $a_t = \pi^k(s_t)$ for all $t_k \le t < t_{k+1}$, while $\mathbb{E}$ is taken with respect to the prior distribution $p_0$ of $\theta^\star$,

the posterior distribution $p_{t_k}$ of $\theta^k$, and the iterative update of $\pi^k$. Here, $\pi^k$ is determined by $\theta^k$ and $p_{t_k}$ is conditioned on $\mathcal{D}_{t_k} = \{(s_t, a_t, s_{t+1}, r_t)\}_{t=0}^{t_k - 1}$, which is generated by $\theta^\star$ and Algorithm 3.

**Analysis of Term (A).** Recall that we define $V_t = V_{\theta^k}^{\pi^k}$ for all $t_k \le t < t_{k+1}$ and $k < K$. We have

$$
\begin{aligned}
\text{term (A)} &= \mathbb{E}\Big[\sum_{k=0}^{K-1} \mathbb{E}_{\pi^k}\Big[\sum_{t=t_k}^{t_{k+1}-1} \psi_{V_t}(s_t, a_t)^\top (\theta^k - \theta^\star)\Big]\Big] \\
&= \mathbb{E}\Big[\sum_{k=0}^{K-1}\Big[\sum_{t=t_k}^{t_{k+1}-1} \mathbb{E}\big[\psi_{V_t}(s_t, a_t)^\top (\theta^k - \theta^\star) \,\big|\, \mathcal{F}_k\big]\Big]\Big] \\
&\le \mathbb{E}\Big[\sum_{k=0}^{K-1}\sum_{t=t_k}^{t_{k+1}-1} \Big(\mathbb{E}\big[\|\psi_{V_t}(s_t, a_t)\|_{\Sigma_{t_k}^{-1}}^2 \,\big|\, \mathcal{F}_k\big]\Big)^{1/2} \cdot \Big(\mathbb{E}\big[\|\theta^k - \theta^\star\|_{\Sigma_{t_k}}^2 \,\big|\, \mathcal{F}_k\big]\Big)^{1/2}\Big] \\
&\le 4 \cdot \mathbb{E}\Big[\sum_{k=0}^{K-1}\sum_{t=t_k}^{t_{k+1}-1} \Big(\mathbb{E}\big[\|\psi_{V_t}(s_t, a_t)\|_{\Sigma_t^{-1}}^2 \,\big|\, \mathcal{F}_k\big]\Big)^{1/2} \cdot \Big(\mathbb{E}\big[\|\theta^k - \theta^\star\|_{\Sigma_{t_k}}^2 \,\big|\, \mathcal{F}_k\big]\Big)^{1/2}\Big],
\end{aligned}
$$
(F.18)

where the first equality uses Definition F.1 and the second equality uses the tower property of the conditional expectation. Note that $\mathcal{F}_k$ is defined in (E.2) and the reward function is deterministic and known in Definition F.1. Also, the first inequality invokes the Holder's inequality and the second inequality uses Lemma 12 in [2] and (F.4). Since $\theta^k$ and $\theta^\star$ share the posterior distribution $\mathcal{N}(\widehat{\theta}_{t_k}, \Sigma_{t_k}^{-1})$ given $\mathcal{D}_{t_k}$, we have that $\|\theta - \widehat{\theta}_{t_k}\|_{\Sigma_{t_k}}^2$ follows $\mathcal{X}_d^2$. Here, $\theta \in \{\theta^\star, \theta^k\}$ and $\mathcal{X}_d^2$ is the chi-square distribution with $d$ degrees of freedom, whose expectation is $d$. Hence, we have

$$
\mathbb{E}\big[\|\theta^k - \theta^\star\|_{\Sigma_{t_k}}^2 \,\big|\, \mathcal{F}_k\big] \le 2 \cdot \mathbb{E}\big[\|\theta^k - \widehat{\theta}_{t_k}\|_{\Sigma_{t_k}}^2 \,\big|\, \mathcal{F}_k\big] + 2 \cdot \mathbb{E}\big[\|\theta^\star - \widehat{\theta}_{t_k}\|_{\Sigma_{t_k}}^2 \,\big|\, \mathcal{F}_k\big] = 2d, \quad \text{(F.19)}
$$

where the first inequality uses the triangle inequality for $\|\cdot\|_{\Sigma_{t_k}}$ and $(a+b)^2 \le 2(a^2+b^2)$ for all $a, b \ge 0$. Plugging (F.19) into (F.18), we obtain

$$
\begin{aligned}
\text{term (A)} &\le 4\sqrt{2d} \cdot \mathbb{E}\Big[\sum_{k=0}^{K-1}\sum_{t=t_k}^{t_{k+1}-1} \Big(\mathbb{E}\big[\|\psi_{V_t}(s_t, a_t)\|_{\Sigma_t^{-1}}^2 \,\big|\, \mathcal{F}_k\big]\Big)^{1/2}\Big] \\
&\le 8\sqrt{(d+1)d} \cdot \mathbb{E}\Big[\sum_{k=0}^{K-1}\sum_{t=t_k}^{t_{k+1}-1} \Big(\mathbb{E}\big[I(\theta; \xi_{t+1} \,|\, \mathcal{D}_t) \,\big|\, \mathcal{F}_k\big]\Big)^{1/2}\Big] \\
&\le 8\sqrt{(d+1)d} \cdot \sqrt{T} \cdot \mathbb{E}\Big[\Big(\sum_{t=0}^{T-1} \mathbb{E}\big[I(\theta; \xi_{t+1} \,|\, \mathcal{D}_t) \,\big|\, \mathcal{F}_k\big]\Big)^{1/2}\Big] \\
&\le 8\sqrt{(d+1)d} \cdot \sqrt{T} \cdot \Big(\mathbb{E}\Big[\sum_{t=0}^{T-1} \mathbb{E}\big[I(\theta; \xi_{t+1} \,|\, \mathcal{D}_t) \,\big|\, \mathcal{F}_k\big]\Big]\Big)^{1/2} \\
&= 8\sqrt{(d+1)d} \cdot \sqrt{T} \cdot \sqrt{\mathbb{E}\big[H_0 - H_T\big]},
\end{aligned}
$$
(F.20)

where the second inequality follows the same arguments in (F.13) and (F.15), the second last inequality invokes the Cauchy-Schwarz inequality. Also, the last inequality uses the concavity of the square-root function, while the last equality uses $I(\theta; \xi_{t+1} \,|\, \mathcal{D}_t) = H_t - H_{t+1}$ and the tower prop-

erty of the conditional expectation. Recall that $H_0 - H_T = \mathcal{O}(d \cdot \log(1 + TR^2/(d\lambda)))$ almost surely in (F.6). We have

$$\text{term (A)} = \mathcal{O}\big(d^{3/2} \cdot \log(TR^2/(d\lambda)) \cdot \sqrt{T}\big). \tag{F.21}$$

**Analysis of Term (B).** By Definition F.1, we have

$$
\begin{aligned}
\text{term (B)} &= \mathbb{E}\Big[\sum_{k=0}^{K-1} \mathbb{E}_{\pi^k}\Big[\big(\psi_{V_{\theta^k}^{\pi^k}}(s_{t_{k+1}-1}, a_{t_{k+1}-1}) - \psi_{V_{\theta^\star}^{\pi^k}}(s_{t_{k+1}-1}, a_{t_{k+1}-1})\big)^\top \theta^\star\Big]\Big] \\
&\quad - \mathbb{E}\Big[\sum_{k=1}^{K-1} \mathbb{E}_{\pi^k}\Big[\big(\psi_{V_{\theta^k}^{\pi^k}}(s_{t_k-1}, a_{t_k-1}) - \psi_{V_{\theta^\star}^{\pi^k}}(s_{t_k-1}, a_{t_k-1})\big)^\top \theta^\star\Big]\Big] + \mathbb{E}\big[V_{\theta^\star}^{\pi^0}(s_0) - V_{\theta^0}^{\pi^0}(s_0)\big] \\
&\leq \mathbb{E}\Big[\sum_{k=0}^{K-1} \mathbb{E}_{\pi^k}\Big[\big\|\psi_{V_{\theta^k}^{\pi^k}}(s_{t_{k+1}-1}, a_{t_{k+1}-1}) - \psi_{V_{\theta^\star}^{\pi^k}}(s_{t_{k+1}-1}, a_{t_{k+1}-1})\big\|_2 \cdot \|\theta^\star\|_2\Big]\Big] \\
&\quad + \mathbb{E}\Big[\sum_{k=1}^{K-1} \mathbb{E}_{\pi^k}\Big[\big\|\psi_{V_{\theta^\star}^{\pi^k}}(s_{t_k-1}, a_{t_k-1}) - \psi_{V_{\theta^k}^{\pi^k}}(s_{t_k-1}, a_{t_k-1})\big\|_2 \cdot \|\theta^\star\|_2\Big]\Big] + 2R \\
&\leq 4R \cdot \mathbb{E}\big[K \cdot \|\theta^\star\|_2\big] + 2R, \tag{F.22}
\end{aligned}
$$

where the first equality uses $s_{t_k} \sim P_{\theta^\star}(\cdot \mid s_{t_k-1}, a_{t_k-1})$, $a_{t_k-1} = \pi^k(s_{t_k-1})$, and

$$\mathbb{E}_{s_{t_k} \sim P_{\theta^\star}(\cdot \mid s_{t_k-1}, a_{t_k-1})} V(s_{t_k}) = \psi_V(s_{t_k-1}, a_{t_k-1})^\top \theta^\star$$

for all $0 < k < K$ and $V$. In (F.22), the first inequality invokes the Cauchy-Schwarz inequality and $|\mathbb{E}_{s_0 \sim \rho} V(s_0)| \leq R$ for all $V$, while the last inequality uses the triangle inequality for $\|\cdot\|_2$ and $\|\psi_V(s, a)\|_2 \leq R$ for all $(s, a) \in \mathcal{S} \times \mathcal{A}$ and $V$. By (F.7), we have

$$
\begin{aligned}
4R \cdot \mathbb{E}\big[K \cdot \|\theta^\star\|_2\big] + 2R &= \mathcal{O}\Big(R \cdot \big(1 + d\log 2 \cdot \log(1 + TR^2/(d\lambda))\big) \cdot \mathbb{E}\big[\|\theta^\star\|_2\big]\Big) \\
&= \mathcal{O}\Big(\lambda^{1/2} d^{1/2} R \cdot \big(1 + d \cdot \log 2 \cdot \log(1 + TR^2/(d\lambda))\big)\Big),
\end{aligned}
$$

where the last equality uses the fact that the prior distribution is $\mathcal{N}(0, \lambda I_d)$ and $\mathbb{E}[\|\theta^\star\|_2] \leq \sqrt{\mathbb{E}[\|\theta^\star\|_2^2]} \leq \sqrt{\lambda d}$. Here, $\|\theta^\star\|_2^2/\lambda$ follows the chi-square distribution with $d$ degrees of freedom, whose expectation is $d$. For a sufficiently large $d$, we have

$$
\begin{aligned}
\text{term (B)} &= \mathcal{O}\big(\lambda^{1/2} d^{1/2} R \cdot \big(1 + d \cdot \log 2 \cdot \log(1 + TR^2/(d\lambda))\big)\big) \\
&= \mathcal{O}\big(\lambda^{1/2} d^{3/2} R \cdot \log(TR^2/(d\lambda))\big). \tag{F.23}
\end{aligned}
$$

Combining Lemma E.1, (F.21), and (F.23), we obtain

$$\mathbb{E}\Big[\sum_{k=0}^{K-1} \mathbb{E}_{\pi^k}\Big[\sum_{t=t_k}^{t_{k+1}-1} V_{\theta^k}^{\pi^k}(s_t) - V_{\theta^\star}^{\pi^k}(s_t)\Big]\Big] = \mathcal{O}\Big(\frac{\gamma d^{3/2} \cdot \log(TR^2/(d\lambda))}{1 - \gamma} \cdot \sqrt{T} + \frac{\gamma \lambda^{1/2} d^{3/2} R}{1 - \gamma} \cdot \log(TR^2/(d\lambda))\Big).$$

By Definition 4.2 and a similar argument of (E.1) in the proof of Theorem 4.4, we obtain

$$\Re(T) = \mathcal{O}\Big(\frac{\gamma d^{3/2} \cdot \log(TR^2/(d\lambda))}{1-\gamma} \cdot \sqrt{T} + \frac{\gamma \lambda^{1/2} d^{3/2} R}{1-\gamma} \cdot \log(TR^2/(d\lambda)) + \epsilon \cdot T\Big).$$

Thus, we prove Proposition F.3. □

## APPENDIX G    MORE EXPERIMENTS

In what follows, we provide the detailed setups and additional results of our experiments.

### G.1    GAME OF 24

**Task Setup.** Game of 24 [73] is a mathematical puzzle where the player uses basic arithmetic operations (i.e., addition, subtraction, multiplication, division) with four given numbers to get 24. Figure 10 gives an illustrative example for Game of 24.

---

**[Illustrative example for Game of 24]**

- Numbers: [2, 5, 8, 11]
- Arithmetic Operations: $[+, -, \times, /, (, )]$
- **Solution**:
$$(11 - 5) \times 8/4 = 24$$

---

Figure 10: An illustrative example of the Game of 24. The player uses combinations of basic arithmetic operations with four given numbers to get 24.

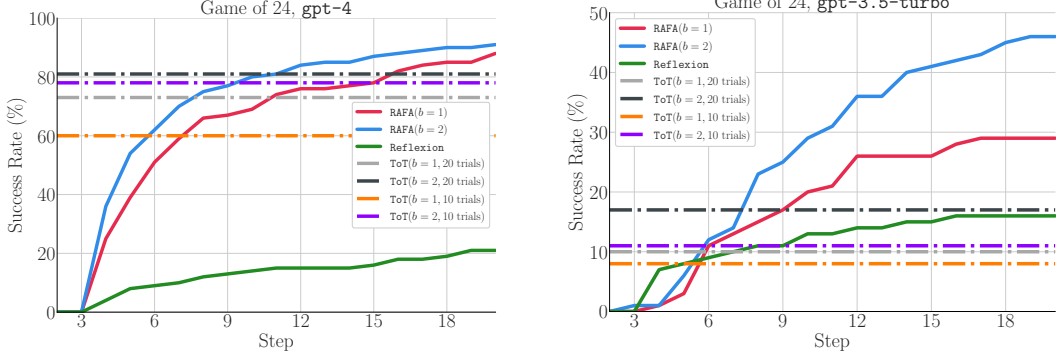

Figure 11: Sample efficiency on Game of 24. RAFA agent achieves strong performance due to an orchestration of reasoning and acting. The success rate at a given step is the number of tasks that is solved within the given step.

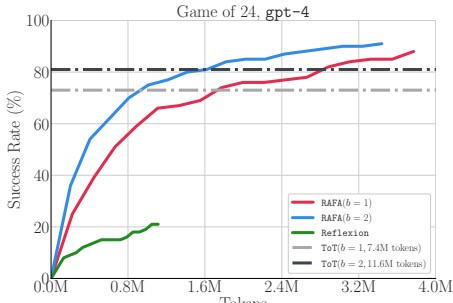 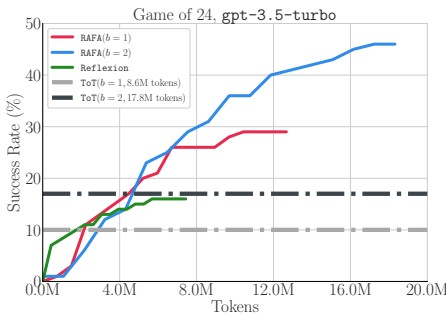

Figure 12: Token efficiency on Game of 24. `RAFA` is superior in terms of token complexity. Methods that lack planning like `Reflexion` has a low token demand, however it is not enough to compensate for the drop in performance. Methods that lacks in-context learning like `ToT` would generate unnecessary repeated trials due to lack of reflection and improvement, which makes the method token inefficient.

To evaluate the performance of `RAFA` in a more profound way, we also consider token complexity for evaluation as shown in Figure 12. This rules out the possibility that some methods can get better performance via just using more queries. We compare the token efficiency of our method on the Game of 24 with various baseline methods and the results and `RAFA` is superior in terms of token complexity. Methods that lack planning like `Reflexion` has a low token demand, however it is not enough to compensate for the drop in performance. Methods that lacks in-context learning like `ToT` would generate unnecessary repeated trials due to lack of reflection and improvement, which makes the method token inefficient.

Following Yao et al. [73], we use the same subset indexed 901-1,000 from a total of 1,362 tasks collected from `4nums.com`. The index is arranged from easy to hard by human solving time so the subset is relatively challenging. The agent receives a reward of 1 if the proposed formula is correct and the proposed formula is accepted and concatenated into the state; if the final result is exactly 24, the agent receives a reward of 10, and the episode terminates. Otherwise, the agent receives a reward of 0, and the proposed formula is not accepted. We limit the maximum trials for each task to 20 to avoid meaningless retries. The task is successful if the agent receives a return larger than 10 [1] (i.e., find a valid solution within 20 steps). We report the final success rate and sample efficiency for each method on the subset of 100 tasks. Notably, a task is considered successful if the `RAFA` agent returns one and only one correct formula, which is more strictly evaluated than Tree of Thoughts (`ToT`, Yao et al. [73]): we allow open-loop agents like `ToT` to retry 20 times and consider them successful if they generate a valid solution in any of the 20 trials. For `CoT` [67] and `Reflexion` [53] agents, we allow them to reflect on the environment's feedback but require them to generate a plan immediately without sophisticated reasoning. only require any one of the best $k$ formulas to be correct for success. Thus, our result is not directly comparable to theirs.

**RAFA Setup.** In the Game of 24, the `RAFA` agent uses ToT as the planner, regenerates a plan when the agent receives a zero reward and continues acting according to the previous plan when the agent receives a positive reward. We set the base ToT planner with beam search width $b = 1, 2$ and use both `gpt-3.5-turbo` and `gpt-4` to test the RAFA's boost-up over LLM agents with different reasoning abilities. We set the temperature $t = 0.2$ by default to favor rigorous reasoning and $t = 0.7$ for majority voting.

**Reduced Hallucination Through Interaction.** A comprehensive review of various method proposals revealed significant hallucination, especially with `gpt-3.5-turbo`. A common hallucination is that the agent believes she can reuse the same number (e.g. using the number 2 twice as

---

[1]For `gpt-3.5-turbo`, we report the success rate when the agent receives a return no less than 3 (i.e., find all sub-steps to get 24 but not necessarily generate a whole correct formula). This is because `ToT` with `gpt-3.5-turbo` is known to suffer from correctly get a whole formula due to limited reasoning ability and non-perfect prompts. See https://github.com/princeton-nlp/tree-of-thought-llm/issues/24 for more details.

illustrated in Figure 2). RAFA efficiently mitigates such hallucination by actively interacting with the environment, displaying exceptional hallucination resistance and improved performance.

**Enhanced Efficiency Through Planning.** Evidenced in Figure 3, the RAFA agent substantially surpasses the Reflexion baseline, reflecting heightened efficiency and minimized regret by negating careless trials. For example, without carefully planning, agent may give negative answers, e.g.,"Impossible to obtain 24 with the given numbers, or unchecked answers, e.g.,"Answer: 6 * 9 / (3 - 2) = 24". This reduction of careless trails is especially achieved when a strong backbone LLMs (e.g., gpt-4) is used, even with a basic planning method, such as BFS with $B = 1$.

**Ablation Study.** The RAFA agent's performance is dissected by individually examining its components: (1) Planning modules or model/elite LLM, (2) Reflection modules or critic LLM, and (3) Different LLMs. Results, displayed in Table 1 and Figure 3, affirm the substantial contribution of each segment to the aggregate performance. Compared to absent or rudimentary zero-shot planning, a basic planner markedly enhances overall performance. However, augmenting planner strength only offers marginal performance enhancements. Both critic LLM and robust LLM usage emerge as pivotal for optimal performance.

## G.2 ALFWORLD

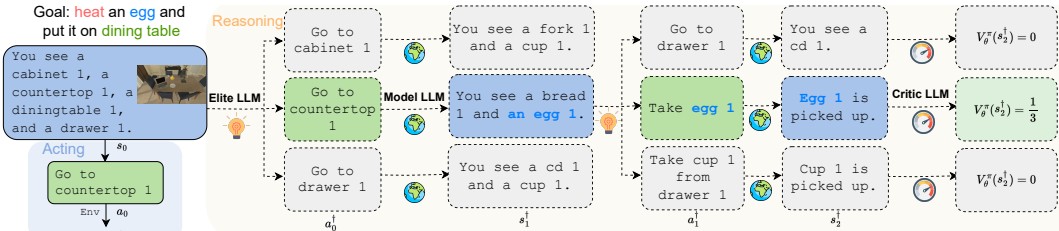

Figure 13: An illustration of RAFA in the ALFWorld environment.

**Task Setup.** The action space of ALFWorld consists of high-level actions such as "heat a potato with a microwave", which is executed in the underlying embodied simulator through low-level action primitives. The egocentric visual observations of the simulator are translated into natural language before being provided to the agent. The state is the history of the observations. If a task goal can be precisely achieved by the agent, it will be counted as a success.

**RAFA Setup.** In the ALFWorld environment, the RAFA planner is instantiated as Breadth First Search (BFS). Specifically, $B$ and $U$ are both set to 2, and we use gpt-3 (text-davinci-003) for the Critic, Model, and Elite modules. Besides, since it is challenging to prompt the LLM with the stored full trajectories in the memory buffer due to the token limit, we make the following modifications: the Model LLM instance uses only the partial trajectory executed so far in the current episode, and the Elite LLM instance uses the same partial executed trajectory with additional model-generated state-action pairs during the planning subroutine. When switching is triggered after 20 failed timesteps (i.e., an episode), a summary from the failure trajectory is generated by gpt-4 and added to the Critic prompt.

**Reduced Hallucination Through Interaction.** The baselines are more likely to hallucinate when the target object is not found after exploring many locations. On the other hand, the critic LLM used in RAFA is able to probe the hallucination by generating the summary "In this environment, my critic assigned a 1/3 value after taking a knife. However, the task is to take and cool a tomato." and avoid it in the next episode. Therefore, RAFA is more sample-efficient due to an orchestration of reasoning and acting and the ability to mitigate hallucination through interaction.

**Ablation Study.** To better understand the role that the planning subroutine plays in the RAFA algorithm, we conduct ablation studies on the search depth $U$ and search breadth $B$. The results are shown in Figure 14 and 15, respectively. We observe that when setting the search depth to $B = U = 2$, the success rate is higher than when setting the search depth to $U = 1$ or setting the search breadth $B = 1$, especially at the initial episode. This indicates that the reasoning ability of

RAFA is enhanced through the planning subroutine. Besides, the algorithm is also more sample-efficient when setting $B = U = 2$, indicating a better capacity for learning and planning through interaction and reasoning.

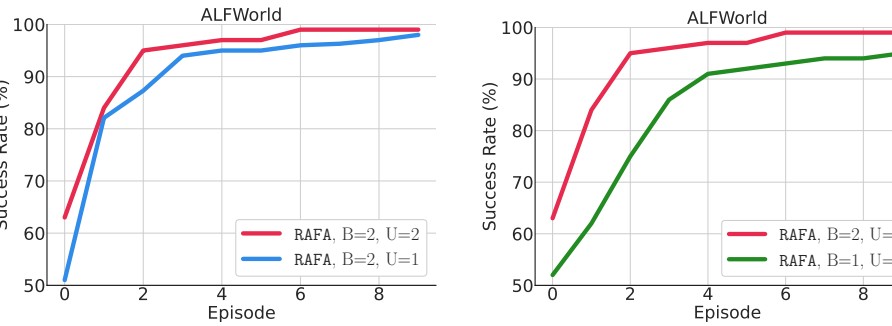

Figure 14: Ablation on the search depth $U$ in the ALFWorld environment.

Figure 15: Ablation on the search breadth $B$ in the ALFWorld environment.

### G.3 BLOCKSWORLD

**Task Setup.** The reported success rates are averaged in tasks that require different minimum steps. Specifically, the evaluation is conducted in 57 4-step tasks and 114 6-step tasks. We set the state as the current arrangement of the blocks and the actions contain Stack, Unstack, Put, and Pickup, coupled with a block being operated.

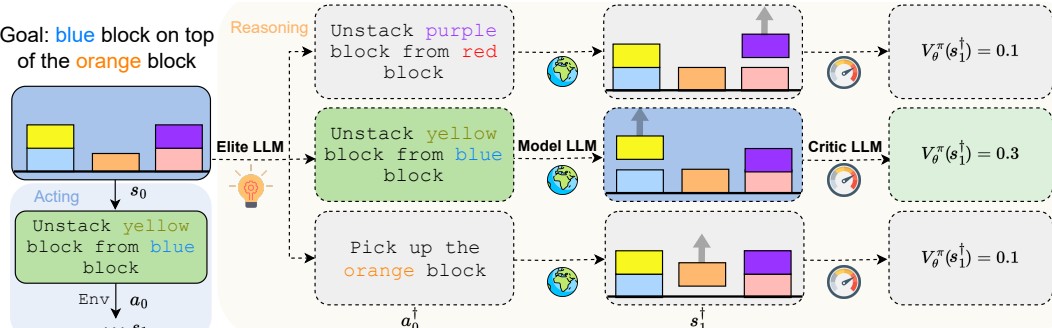

Figure 16: Illustration of RAFA in the BlocksWorld environment.

**RAFA Setup.** The search space is up to $5^4$ for a 4-step task and is up to $5^6$ for a 6-step task. For 4-step tasks, RAFA can achieve over 50% success rate within 8 learning steps with Vicuna-13B(v1.3) and achieve over 80% success rate within 8 learning steps with Vicuna-33B(v1.3). For 6-step tasks, RAFA can achieve over 40% success rate within 20 learning steps with Vicuna-13B(v1.3) and achieve over 50% success rate within 20 learning steps with Vicuna-33B(v1.3). Empirical results show that Vicuna could produce wrong state transition in the planning phase. RAFA can mitigate hallucination with feedback from failure trajectories and active exploration. One can draw such a conclusion by comparing RAFA with RAP as RAP does not receive feedback from the real environment.

### G.4 TIC-TAC-TOE

**Task Setup.** Tic-Tac-Toe [7] is a competitive game in which two players take turns to mark a three-by-three grid with X or O, and a player succeeds when their marks occupy a diagonal, horizontal, or vertical line.

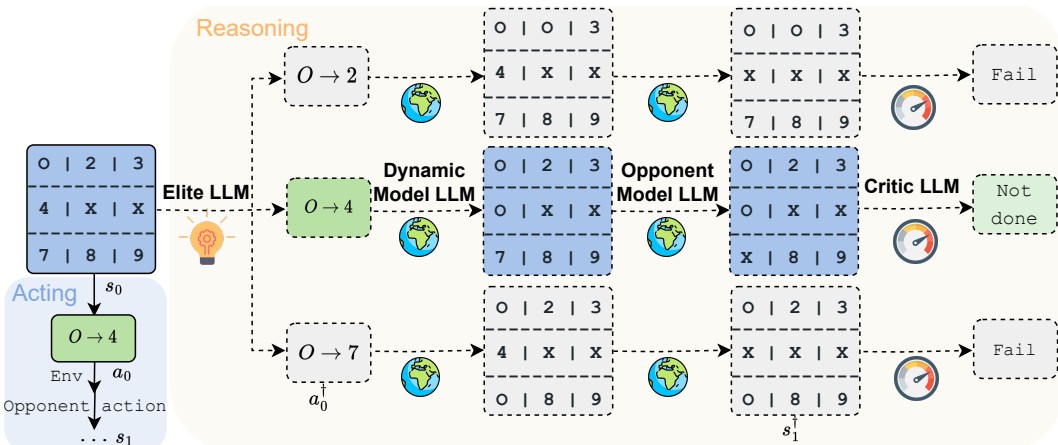

Figure 17: Illustration of RAFA (playing O) in the Tic-Tac-Toe game. States are represented by a numbered $3 \times 3$ grid and actions are represented by a number between 1-9. The opponent is considered part of the environment.

We adopt the convention that X plays first. As illustrated below in Figure ??, we use a numbered $3 \times 3$ grid to represent a state and a number between 1 and 9 to represent an action, which also illustrates the transition and reward function. Although Tic-Tac-Toe is a solved game with a forced draw assuming the best play from both players, it remains a challenge for LLMs to accomplish this task even when prompted to play only the optimal moves. We collected the battle outcomes between different LLM models in Table 5, where we notice that gpt-4 performs worse when playing as "O". Thus, in our experiments, we let RAFA play as "O" and let baseline LLM models play as "X".

| X wins : Tie : O wins | | O | |
| --- | --- | --- | --- |
| | | gpt-3.5 | gpt-4 |
| X | gpt-3.5 | $55\% : 35\% : 10\%$ | $90\% : 0\% : 10\%$ |
| | gpt-4 | $65\% : 15\% : 20\%$ | $90\% : 0\% : 10\%$ |

Table 5: Probability of "X wins," "Tie," and "O wins" in Tic-Tac-Toe. The results are obtained by averaging over 20 simulated games.

**RAFA Setup.** For implementation, we set $B = 3$ and adopt MCTS to evaluate the proposed actions. We set $U = 4$ which is the maximum game depth. We set a prediction-based switching condition triggered when the prediction does not agree with the observation. Specifically, policy switches when one of the following events occurs:

- The RAFA agent takes an action and predicts the next state, which is different from the observed next state.
- Before the opponent takes an action, the RAFA agent tries to predict such an action, which is different from the actual action that the opponent takes.
- After the opponent takes an action, RAFA agent predicts the next state, which is different from the observed next state.
- The RAFA agent predicts the current game status (X wins, O wins, Tie, Not finished), which is different from the environment's feedback.

Besides, we use the ground truth of those predictions to update the agent's belief of the world, which also implicitly affects the agent's policy.

We define a discrete reward function with $r = -1, 0, 1$ corresponding to lose, tie, and win. The agent only gets rewards when the current episode is completed. We define the score of an agent as its expected reward which can be approximated by simulation. The empirical results are shown in

figure 18. We conduct experiments using both `gpt-4` as the backend. The score of RAFA ($B = 4$) increases as it interacts more with the environment. By analyzing the generated trajectories, we also notice that although RAFA agent is not perfect, it exploits the weakness of the baseline model well, which is why it almost never loses after 7 episodes.

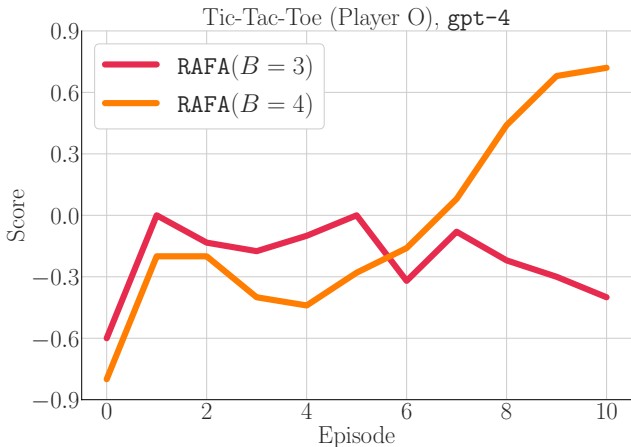

Figure 18: Score curves in the Tic-Tac-Toe game. We use `gpt-4` as backend. Results are averaged across 10 simulations and smoothed with a window size of 5.

## APPENDIX H    PROMPTS

In this section, we give details of the prompts used for each task.

### H.1    GAME OF 24

**Critic LLM.** For the LLM instance of the `Critic`, we prompt it with the current action (formula) with reward and feedback from the environment. The critic is required to determine whether each formula is valid or not and give a "sure" or "impossible" label for the formula. We use critic prompts to generate reflection for formula proposal and evaluation, respectively.

```
Critic prompt (for formula proposal)

Now we would like to play a game of 24. That is, given 4 numbers, try
to use them with arithmetic operations (+ - * /) to get 24. Now we
consider the following puzzle: {input}.
Here is an attempt answer:
{answer}
And we have the following feedback:
{feedback}
Now using the above feedback, give 'sure' or 'impossible' labels for
each formula with left numbers from each step. Give 'sure' if the
formula is correct and can lead to 24 and give 'impossible' if the
formula is incorrect or illegal. First repeat the formula with left
numbers from each step above and then give the label, with the
following form: {{formula}} (left: {{left numbers}}): {{label}}.
```

```
┌─────────── Critic prompt (for formula evaluation) ───────────┐
Now we would like to play a game of 24. That is, given 4 numbers, try
to use them with arithmetic operations (+ - * /) to get 24. Now we
consider the following puzzle: {input}.
Here is an attempt answer:
{answer}
And we have the following feedback:
{feedback}
Now using the above feedback, give 'sure' or 'impossible' labels for
left numbers from each step. Give 'sure' if the formula is correct and
left numbers can lead to 24 and give 'impossible' if the formula is
incorrect or illegal. First repeat the left numbers from each step
above and then give the label, with the following form: {{left
numbers}}: {{label}}.
└──────────────────────────────────────────────────────────────┘
```

**Elite LLM.** We adopt the same prompts used in Tree-of-Thoughts [73] to propose and evaluate formulas, except that we concatenate the reflections from each step to avoid making repeated mistakes.

```
┌─────────── Elite prompt (for formula proposal) ───────────┐
Now we would like to play a game of 24. That is, given 4 numbers, try
to use them with arithmetic operations (+ - * /) to get 24.
Evaluate if given numbers can reach 24 and choose labels from 'sure',
'likely' and 'impossible'.
What you have learned about the puzzle are summarized below.
{reflections}
Now use numbers and basic arithmetic operations (+ - * /) to generate
possible next steps. Make sure use steps that is sure to leads to 24
and avoid steps that are impossible to generate 24. Note that it is
possible that we are considering intermediate steps so the numbers of
the input may be less than 4.
Example:
Input: 2 8 8 14
Possible next steps:
2 + 8 = 10 (left: 8 10 14)
8 / 2 = 4 (left: 4 8 14)
14 + 2 = 16 (left: 8 8 16)
2 * 8 = 16 (left: 8 14 16)
8 - 2 = 6 (left: 6 8 14)
14 - 8 = 6 (left: 2 6 8)
14 /  2 = 7 (left: 7 8 8)
14 - 2 = 12 (left: 8 8 12)
Example:
Input: 2 5 8
5 - 2 = 3 (left: 3 8)
5 * 2 = 10 (left: 10 8)
└───────────────────────────────────────────────────────────┘
```

```
8 / 2 = 4 (left: 4 5)
Now try with the following input:
Input: {input}
Possible next steps:
  {input}
```

```
───────────────── Elite prompt (for formula evaluation) ─────────────────
Now we would like to play a game of 24. That is, given 4 numbers, try
to use them with arithmetic operations (+ - * /) to get 24.
Evaluate if given numbers can reach 24 and choose labels from 'sure',
'likely' and 'impossible'.
What you have learned about the puzzle are summarized below.
{reflections}
If the given numbers are already in the feedback above, just give the
answer. Otherwise enumerate possible steps and try to give an
approximate answer. Give the final answer in a separated line.
  {input}
```

```
───────────── Elite prompt (for last step formula evaluation) ─────────────
Now we would like to play a game of 24. That is, given 4 numbers, try
to use them with arithmetic operations (+ - * /) to get 24.
Evaluate if given numbers can reach 24 and choose labels from 'sure',
'likely' and 'impossible'.
What you have learned about the puzzle are summarized below.
{reflections}
Use numbers and basic arithmetic operations (+ - * /) to obtain 24.
Given an input and an answer, give a judgement (sure/impossible) if
the answer is correct, i.e., it uses each input exactly once and no
other numbers, and reach 24.
Input: 4 4 6 8
Answer: (4 + 8) * (6 - 4) = 24
Judge:
sure
Input: 2 9 10 12
Answer: 2 * 12 * (10 - 9) = 24
Judge:
sure
Input: 4 9 10 13
Answer: (13 - 9) * (10 - 4) = 24
Judge:
sure
Input: 4 4 6 8
Answer: (4 + 8) * (6 - 4) + 1 = 25
Judge:
```

```
impossible
Input: 2 9 10 12
Answer: 2 * (12 - 10) = 24
Judge:
impossible
Input: 4 9 10 13
Answer: (13 - 4) * (10 - 9) = 24
Judge:
impossible
Input: {input}
Answer: {answer}
Judge:
```

For Chain-of-Thought baselines, we adopt the same methodology, and keep the original prompts except for adding reflections as below.

```
┌─ Elite prompt (for chain-of-thought proposals) ─┐
Now we would like to play a game of 24. That is, given 4 numbers, try
to use them with arithmetic operations (+ - * /) to get 24.
Evaluate if given numbers can reach 24 and choose labels from 'sure',
'likely' and 'impossible'.
What you have learned about the puzzle are summarized below.
{reflections}
Now just remember the tips from before (if any) and focus on the new
task. Use numbers and basic arithmetic operations (+ - * /) to obtain
24. Each step, you are only allowed to choose two of the remaining
numbers to obtain a new number.
Input: 4 4 6 8
Steps:
4 + 8 = 12 (left: 4 6 12)
6 - 4 = 2 (left: 2 12)
2 * 12 = 24 (left: 24)
Answer: (6 - 4) * (4 + 8) = 24
Input: 2 9 10 12
Steps:
12 * 2 = 24 (left: 9 10 24)
10 - 9 = 1 (left: 1 24)
24 * 1 = 24 (left: 24)
Answer: (12 * 2) * (10 - 9) = 24
Input: 4 9 10 13
Steps:
13 - 10 = 3 (left: 3 4 9)
9 - 3 = 6 (left: 4 6)
4 * 6 = 24 (left: 24)
Answer: 4 * (9 - (13 - 10)) = 24
```

```
Input: 1 4 8 8
Steps:
8 / 4 = 2 (left: 1 2 8)
1 + 2 = 3 (left: 3 8)
3 * 8 = 24 (left: 24)
Answer: (1 + 8 / 4) * 8 = 24
Input: 5 5 5 9
Steps:
5 + 5 = 10 (left: 5 9 10)
10 + 5 = 15 (left: 9 15)
15 + 9 = 24 (left: 24)
Answer: ((5 + 5) + 5) + 9 = 24
Input: {input}
```

## H.2 ALFWORLD

**Critic LLM.** For the LLM instance of the `Critic`, we prompt it with successful examples for each of the six task types to generate the value within each episode. Specifically, the critic LLM first determines if the value of the current state changes. If it changes, the critic LLM then generates the scalar value with the corresponding reason. We provide the following two examples in the put and cool tasks, respectively.

```
┌──────────── Critic prompt (for value generation), "put" task ────────────┐
You are in the middle of a room. Looking quickly around you, you see a
cabinet 4, a cabinet 3, a cabinet 2, a cabinet 1, a countertop 1, a
garbagecan 1, a handtowelholder 2, a handtowelholder 1, a sinkbasin 2,
a sinkbasin 1, a toilet 1, a toiletpaperhanger 1, and a towelholder
1.Your task is to: put some spraybottle on toilet.
> critic: My task requires two sub-goals in order: take a spraybottle
and put the spraybottle on the toilet. My current state satisfies zero
of the two sub-goals. The value is 0/2=0.
> OK.
On the cabinet 1, you see a cloth 1, a soapbar 1, a soapbottle 1.
> OK.
The cabinet 2 is closed.
> OK.
You open the cabinet 2. The cabinet 2 is open. In it, you see a candle
1, and a spraybottle 2.
> OK.
You pick up the spraybottle 2 from the cabinet 2.
> critic: Now I take a spraybottle. My current state satisfies the
first of the two sub-goals: take a spraybottle. The value is 1/2=0.5.
> OK.
On the toilet 1, you see a soapbottle 2.
> OK.
```

```
You put the spraybottle 2 in/on the toilet 1.
> critic: Now I put the spraybottle on the toilet. My current state
satisfies all the two sub-goals. The value is 2/2=1.
```

```
────── Critic prompt (for value generation), "cool" task ──────
You are in the middle of a room. Looking quickly around you, you see a
cabinet 16, a cabinet 15, a cabinet 14, a cabinet 13, a cabinet 12, a
cabinet 11, a cabinet 10, a cabinet 9, a cabinet 8, a cabinet 7, a
cabinet 6, a cabinet 5, a cabinet 4, a cabinet 3, a cabinet 2, a
cabinet 1, a coffeemachine 1, a countertop 2, a countertop 1, a
diningtable 1, a drawer 5, a drawer 4, a drawer 3, a drawer 2, a
drawer 1, a fridge 1, a garbagecan 1, a microwave 1, a safe 1, a
sinkbasin 1, a stoveburner 4, a stoveburner 3, a stoveburner 2, a
stoveburner 1, and a toaster 1.
Your task is to: cool some pan and put it in stoveburner.
> critic: My task requires three sub-goals in order: take a pan, cool
the pan with fridge, and put the pan in the stoveburner. My current
state satisfies zero of the three sub-goals. The value is 0/3=0.
> OK.
On the stoveburner 1, you see nothing.
> OK.
On the stoveburner 2, you see a pot 1.
> OK.
On the stoveburner 3, you see a pan 1.
> OK.
You pick up the pan 1 from the stoveburner 3.
> critic: Now I take a pan. My current state satisfies the first of
the three sub-goals: take a pan. The value is 1/3=0.333.
> OK.
The fridge 1 is closed.
> OK.
You cool the pan 1 using the fridge 1.
> critic: Now I cool the pan. My current state satisfies the first two
of the three sub-goals: take a pan and cool the pan with fridge. The
value is 2/3=0.667.
> OK.
On the stoveburner 1, you see nothing.
> OK.
You put the pan 1 in/on the stoveburner 1.
> critic: Now I put the pan in the stoveburner. My current state
satisfies all the three sub-goals. The value is 3/3=1.
```

In addition to the prompt for value generation within each episode, we also prompt the `Critic` LLM to generate summary feedback according to the past failure trajectory. We use the same prompt

for all six task types. Specifically, it contains the full failure trajectory (observation only), as well as the failure feedback at the end of each trajectory.

```
┌─ Critic prompt (for feedback generation) ─┐
│ You are in the middle of a room. Looking quickly around you, you see a  │
│ cabinet 6, a cabinet 5, a cabinet 4, a cabinet 3, a cabinet 2, a  │
│ cabinet 1, a coffeemachine 1, a countertop 3, a countertop 2, a  │
│ countertop 1, a drawer 3, a drawer 2, a drawer 1, a fridge 1, a  │
│ garbagecan 1, a microwave 1, a shelf 3, a shelf 2, a shelf 1, a  │
│ sinkbasin 1, a stoveburner 4, a stoveburner 3, a stoveburner 2, a  │
│ stoveburner 1, and a toaster 1.  │
│ Your task is to: put a clean plate in countertop.  │
│ > critic: My task requires three sub-goals in order: take a plate,  │
│ clean the plate with sinkbasin, and put the plate on the countertop.  │
│ My current state satisfies zero of the three sub-goals. The value is  │
│ 0/3=0.  │
│ > OK.  │
│ The drawer 1 is closed.  │
│ > OK.  │
│ You open the drawer 1. The drawer 1 is open. In it, you see a  │
│ creditcard 2, a knife 1, and a spatula 2.  │
│ > OK.  │
│ On the stoveburner 2, you see a pan 1.  │
│ > OK.  │
│ You pick up the pan 1 from the stoveburner 2.  │
│ > critic: Now I take a pan. My current state satisfies the first of  │
│ the three sub-goals: take a plate. The value is 1/3=0.167.  │
│ > OK.  │
│ On the sinkbasin 1, you see nothing.  │
│ > OK.  │
│ You put the pan 1 in/on the sinkbasin 1.  │
│ > OK.  │
│ The microwave 1 is closed.  │
│ > OK.  │
│ You open the microwave 1. The microwave 1 is open. In it, you see a  │
│ potato 1.  │
│ > OK.  │
│ On the garbagecan 1, you see a soapbottle 2.  │
│ > OK.  │
│ The microwave 1 is open. In it, you see a potato 1.  │
│ > OK.  │
│ On the coffeemachine 1, you see nothing.  │
│ > OK.  │
│ On the countertop 2, you see a bread 1, a cellphone 2, a cellphone 1,  │
│ a papertowelroll 1, a plate 2, and a soapbottle 1.  │
```

```
> OK.
The drawer 2 is closed.
> OK.
You open the drawer 2. The drawer 2 is open. In it, you see a spatula
1.
> OK.
On the sinkbasin 1, you see a pan 1.
> OK.
On the cabinet 3, you see a cup 1.
> OK.
On the countertop 1, you see a apple 2, a dishsponge 2, a potato 3,
and a potato 2.
STATUS: FAIL
Failure feedback: In this environment, my critic assigned a 1/3 value
after taking a pan. However, the task is to take and clean a plate. I
noticed that the plate was found on countertop 2. In the next trial, I
will go to countertop 2 to take the plate, then go to a sinkbasin to
clean the plate.

You are in the middle of a room. Looking quickly around you, you see a
cabinet 20, a cabinet 19, a cabinet 18, a cabinet 17, a cabinet 16, a
cabinet 15, a cabinet 14, a cabinet 13, a cabinet 12, a cabinet 11, a
cabinet 10, a cabinet 9, a cabinet 8, a cabinet 7, a cabinet 6, a
cabinet 5, a cabinet 4, a cabinet 3, a cabinet 2, a cabinet 1, a
coffeemachine 1, a countertop 3, a countertop 2, a countertop 1, a
drawer 3, a drawer 2, a drawer 1, a fridge 1, a garbagecan 1, a
microwave 1, a shelf 3, a shelf 2, a shelf 1, a sinkbasin 1, a
stoveburner 4, a stoveburner 3, a stoveburner 2, a stoveburner 1, and
a toaster 1.
Your task is to: put two saltshaker in drawer.
> critic: My task requires four sub-goals in order: take the first
saltshaker, put the first saltshaker in the drawer, take the second
saltshaker, and put the second saltshaker in the drawer. My current
state satisfies zero of the four sub-goals. The value is 0/4=0.
> OK.
On the coffeemachine 1, you see a cup 1, a fork 4, and a potato 2.
> OK.
On the toaster 1, you see a pan 1 and a knife 1.
> OK.
On the countertop 3, you see a bowl 2, a houseplant 1, a knife 2, a
peppershaker 1, a saltshaker 4, a soapbottle 1, a spatula 2, and a
tomato 3.
> OK.
```

```
You pick up the saltshaker 4 from the countertop 3.
> critic: Now I take the first saltshaker. My current state satisfies
the first of the four sub-goals: take the first saltshaker. The value
is 1/4=0.25.
> OK.
The drawer 1 is open. In it, you see nothing.
> OK.
You put the saltshaker 4 in/on the drawer 1.
> critic: Now I put the first creditcard in the dresser. My current
state satisfies the first two of the four sub-goals: take the first
saltshaker and put the first saltshaker in the drawer. The value is
2/4=0.5.
> OK.
On the stoveburner 1, you see a bread 1.
> OK.
On the stoveburner 2, you see a potato 1, a pot 2, and a knife 3.
> OK.
On the stoveburner 3, you see a fork 1, a fork 2, and a cup 2.
> OK.
On the stoveburner 4, you see a peppershaker 1 and a pan 2.
> OK.
You pick up the peppershaker 1 from the stoveburner 4.
> critic: Now I take the second saltshaker. My current state satisfies
the first three of the four sub-goals: take the first saltshaker, put
the first saltshaker in the drawer, and take the second saltshaker.
The value is 3/4=0.75.
> OK.
The drawer 1 is open. In it, you see a saltshaker 4.
> OK.
You put the peppershaker 1 in/on the drawer 1.
> critic: Now I put the second creditcard in the dresser. My current
state satisfies all the four sub-goals. The value is 4/4=1.
STATUS: FAIL
Failure feedback: In this environment, my critic assigned a 1/4 value
after taking a saltshaker and a 3/4 value after taking a peppershaker.
However, the task is to put two saltshaker in the drawer. I noticed
that one saltshaker was put on the countertop 3. I didn't find the
second saltshaker in this trial. In the next trial, I will first go to
countertop 3 to take the first saltshaker and then explore other
locations to find a second saltshaker, including drawer 2-3, shelf
1-3, and cabinet 1-20.
```

```
You are in the middle of a room. Looking quickly around you, you see a
cabinet 5, a cabinet 4, a cabinet 3, a cabinet 2, a cabinet 1, a
coffeemachine 1, a countertop 3, a countertop 2, a countertop 1, a
drawer 3, a drawer 2, a drawer 1, a fridge 1, a garbagecan 1, a
microwave 1, a shelf 3, a shelf 2, a shelf 1, a sinkbasin 1, a
stoveburner 4, a stoveburner 3, a stoveburner 2, a stoveburner 1, and
a toaster 1.
Your task is to: put a cool tomato in microwave.
> critic: My task requires three sub-goals in order: take a tomato,
cool the tomato with fridge, and put the tomato in the microwave. My
current state satisfies zero of the three sub-goals. The value is
0/3=0.
> OK.
On the countertop 1, you see a mug 1, and a saltshaker 1.
> OK.
On the sinkbasin 1, you see a fork 1, a glassbottle 2, a spatula 1,
and a spoon 3.
> OK.
You pick up the fork 1 from the sinkbasin 1.
> critic: Now I take a fork. My current state satisfies the first of
the six sub-goals: take a fork. The value is 1/3=0.333.
> OK.
On the countertop 2, you see a butterknife 2, a butterknife 1, a
cellphone 1, a glassbottle 1, a knife 1, a lettuce 3, a peppershaker
1, a statue 2, a statue 1, and a vase 1.
> OK.
On the countertop 3, you see a bread 1, a butterknife 3, a creditcard
2, a houseplant 1, a knife 3, a knife 2, a mug 2, a peppershaker 2,
and a spatula 2.
> OK.
On the stoveburner 4, you see a pan 1.
> OK.
The drawer 3 is closed.
> OK.
You open the drawer 3. The drawer 3 is open. In it, you see a
saltshaker 3.
> OK.
The fridge 1 is closed.
> OK.
On the countertop 3, you see a bread 1, a butterknife 3, a creditcard
2, a houseplant 1, a knife 3, a knife 2, a mug 2, a peppershaker 2,
and a spatula 2.
> OK.
```

```
On the cabinet 1, you see a dishsponge 1.
> OK.
The cabinet 2 is closed.
> OK.
You open the cabinet 2. The cabinet 2 is open. In it, you see nothing.
> OK.
On the cabinet 3, you see a cd 1 and a plate 4.
STATUS: FAIL
Failure feedback: In this environment, my critic assigned a 1/3 value
after taking a fork. However, the task is to take and cool a tomato. I
didn't find the tomato in this trial. In the next trial, I will
explore other locations to find a tomato, including cabinet 4, cabinet
5, coffeemachine 1, microwave 1, shelf 1-3, stoveburner 1-4 and
toaster 1, etc.
```

**Model LLM and Elite LLM.** We use the same prompt for both the `Model` LLM and the `Elite` LLM. Specifically, we simply prompt these two instances with the successful past trajectories (observations and actions) for each task type and expect the `Model` LLM and `Elite` LLM to generate the possible next observation and the potential actions, respectively. Below, we provide two prompt examples in the put and cool tasks, respectively.

```
─────────── Model & Elite prompt, "put" task ───────────
You are in the middle of a room. Looking quickly around you, you see a
cabinet 4, a cabinet 3, a cabinet 2, a cabinet 1, a countertop 1, a
garbagecan 1, a handtowelholder 2, a handtowelholder 1, a sinkbasin 2,
a sinkbasin 1, a toilet 1, a toiletpaperhanger 1, and a towelholder 1.
Your task is to: put some spraybottle on toilet.
> go to cabinet 1
On the cabinet 1, you see a cloth 1, a soapbar 1, a soapbottle 1.
> go to cabinet 2
The cabinet 2 is closed.
> open cabinet 2
You open the cabinet 2. The cabinet 2 is open. In it, you see a candle
1, and a spraybottle 2.
> take spraybottle 2 from cabinet 2
You pick up the spraybottle 2 from the cabinet 2.
> go to toilet 1
On the toilet 1, you see a soapbottle 2.
> put spraybottle 2 in/on toilet 1
You put the spraybottle 2 in/on the toilet 1.
```

```
┌──────────────────── Model & Elite prompt, "cool" task ────────────────────┐
You are in the middle of a room. Looking quickly around you, you see a
cabinet 16, a cabinet 15, a cabinet 14, a cabinet 13, a cabinet 12, a
cabinet 11, a cabinet 10, a cabinet 9, a cabinet 8, a cabinet 7, a
cabinet 6, a cabinet 5, a cabinet 4, a cabinet 3, a cabinet 2, a
cabinet 1, a coffeemachine 1, a countertop 2, a countertop 1, a
diningtable 1, a drawer 5, a drawer 4, a drawer 3, a drawer 2, a
drawer 1, a fridge 1, a garbagecan 1, a microwave 1, a safe 1, a
sinkbasin 1, a stoveburner 4, a stoveburner 3, a stoveburner 2, a
stoveburner 1, and a toaster 1.
Your task is to: cool some pan and put it in stoveburner.
> go to stoveburner 1
On the stoveburner 1, you see nothing.
> go to stoveburner 2
On the stoveburner 2, you see a pot 1.
> go to stoveburner 3
On the stoveburner 3, you see a pan 1.
> take pan 1 from stoveburner 3
You pick up the pan 1 from the stoveburner 3.
> go to fridge 1
The fridge 1 is closed.
> cool pan 1 with fridge 1
You cool the pan 1 using the fridge 1.
> go to stoveburner 1
On the stoveburner 1, you see nothing.
> put pan 1 in/on stoveburner 1
You put the pan 1 in/on the stoveburner 1.
```

## H.3 BLOCKSWORLD

**Critic LLM.** We evaluate RAFA and RAP with the reward scheme proposed by [23]. We prompt the language model with the previous state-action trajectory and calculate the log probabilities of taking each feasible action. Given the action taken in the current state, the `Model` LLM predicts the next state and we calculate the percentage of subgoals completed in the next state. We adopt the prompt examples from [23] to ensure the fairness in comparison.

```
┌──────────── Critic prompt example (for log probability), "step-4" task ────────────┐
I am playing with a set of blocks where I need to arrange the blocks
into stacks. Here are the actions I can do

Pick up a block
Unstack a block from on top of another block
Put down a block
Stack a block on top of another block

I have the following restrictions on my actions:
```

I can only pick up or unstack one block at a time.
I can only pick up or unstack a block if my hand is empty.
I can only pick up a block if the block is on the table and the block
is clear. A block is clear if the block has no other blocks on top of
it and if the block is not picked up.
I can only unstack a block from on top of another block if the block I
am unstacking was really on top of the other block.
I can only unstack a block from on top of another block if the block I
am unstacking is clear.
Once I pick up or unstack a block, I am holding the block.
I can only put down a block that I am holding.
I can only stack a block on top of another block if I am holding the
block being stacked.
I can only stack a block on top of another block if the block onto
which I am stacking the block is clear.
Once I put down or stack a block, my hand becomes empty.

[STATEMENT]
As initial conditions I have that, the red block is clear, the yellow
block is clear, the hand is empty, the red block is on top of the blue
block, the yellow block is on top of the orange block, the blue block
is on the table and the orange block is on the table.
My goal is to have that the orange block is on top of the red block.

My plan is as follows:

[PLAN]
unstack the yellow block from on top of the orange block
put down the yellow block
pick up the orange block
stack the orange block on top of the red block
[PLAN END]

[STATEMENT]
As initial conditions I have that, the orange block is clear, the
yellow block is clear, the hand is empty, the blue block is on top of
the red block, the orange block is on top of the blue block, the red
block is on the table and the yellow block is on the table.
My goal is to have that the blue block is on top of the red block and
the yellow block is on top of the orange block.

My plan is as follows:

```
[PLAN]
pick up the yellow block
stack the yellow block on top of the orange block
[PLAN END]

[STATEMENT]
As initial conditions I have that, the red block is clear, the blue
block is clear, the orange block is clear, the hand is empty, the blue
block is on top of the yellow block, the red block is on the table,
the orange block is on the table and the yellow block is on the table.
My goal is to have that the blue block is on top of the orange block
and the yellow block is on top of the red block.

My plan is as follows:

[PLAN]
unstack the blue block from on top of the yellow block
stack the blue block on top of the orange block
pick up the yellow block
stack the yellow block on top of the red block
[PLAN END]

[STATEMENT]
As initial conditions I have that, the red block is clear, the blue
block is clear, the yellow block is clear, the hand is empty, the
yellow block is on top of the orange block, the red block is on the
table, the blue block is on the table and the orange block is on the
table.
My goal is to have that the orange block is on top of the blue block
and the yellow block is on top of the red block.

My plan is as follows:

[PLAN]
unstack the yellow block from on top of the orange block
stack the yellow block on top of the red block
pick up the orange block
stack the orange block on top of the blue block
[PLAN END]
```

**Model LLM.** we prompt the `Model` LLM with few-shot examples and the current state and action. The `Model` LLM generates the predicted next state description. We adopt the prompt examples from [23] to ensure the fairness in comparison.

```
┌───────── Model prompt template, "Pick up" action ─────────┐
```

I am playing with a set of blocks where I need to arrange the blocks
into stacks. Here are the actions I can do

Pick up a block
Unstack a block from on top of another block
Put down a block
Stack a block on top of another block

I have the following restrictions on my actions:
I can only pick up or unstack one block at a time.
I can only pick up or unstack a block if my hand is empty.
I can only pick up a block if the block is on the table and the block
is clear. A block is clear if the block has no other blocks on top of
it and if the block is not picked up.
I can only unstack a block from on top of another block if the block I
am unstacking was really on top of the other block.
I can only unstack a block from on top of another block if the block I
am unstacking is clear. Once I pick up or unstack a block, I am
holding the block.
I can only put down a block that I am holding.
I can only stack a block on top of another block if I am holding the
block being stacked.
I can only stack a block on top of another block if the block onto
which I am stacking the block is clear. Once I put down or stack a
block, my hand becomes empty.

After being given an initial state and an action, give the new state
after performing the action.

[SCENARIO 1]
[STATE 0] I have that, the white block is clear, the cyan block is
clear, the brown block is clear, the hand is empty, the white block is
on top of the purple block, the purple block is on the table, the cyan
block is on the table and the brown block is on the table.
[ACTION] Pick up the brown block.
[CHANGE] The hand was empty and is now holding the brown block, the
brown block was on the table and is now in the hand, and the brown
block is no longer clear.
[STATE 1] I have that, the white block is clear, the cyan block is
clear, the brown block is in the hand, the hand is holding the brown
block, the white block is on top of the purple block, the purple block
is on the table and the cyan block is on the table.

```
[SCENARIO 2]
[STATE 0] I have that, the purple block is clear, the cyan block is
clear, the white block is clear, the hand is empty, the white block is
on top of the brown block, the purple block is on the table, the cyan
block is on the table and the brown block is on the table.
[ACTION] Pick up the cyan block.
[CHANGE] The hand was empty and is now holding the cyan block, the
cyan block was on the table and is now in the hand, and the cyan block
is no longer clear.
[STATE 1] I have that, the cyan block is in the hand, the white block
is clear, the purple block is clear, the hand is holding the cyan
block, the white block is on top of the brown block, the purple block
is on the table and the brown block is on the table.
```

```
┌──────────── Model prompt template, "Unstack" action ────────────┐
I am playing with a set of blocks where I need to arrange the blocks
into stacks. Here are the actions I can do

Pick up a block
Unstack a block from on top of another block
Put down a block
Stack a block on top of another block

I have the following restrictions on my actions:
I can only pick up or unstack one block at a time.
I can only pick up or unstack a block if my hand is empty.
I can only pick up a block if the block is on the table and the block
is clear. A block is clear if the block has no other blocks on top of
it and if the block is not picked up.
I can only unstack a block from on top of another block if the block I
am unstacking was really on top of the other block.
I can only unstack a block from on top of another block if the block I
am unstacking is clear. Once I pick up or unstack a block, I am
holding the block.
I can only put down a block that I am holding.
I can only stack a block on top of another block if I am holding the
block being stacked.
I can only stack a block on top of another block if the block onto
which I am stacking the block is clear. Once I put down or stack a
block, my hand becomes empty.
```

After being given an initial state and an action, give the new state
after performing the action.

[SCENARIO 1]
[STATE 0] I have that, the white block is clear, the cyan block is
clear, the brown block is clear, the hand is empty, the white block is
on top of the purple block, the purple block is on the table, the cyan
block is on the table and the brown block is on the table.
[ACTION] Unstack the white block from on top of the purple block.
[CHANGE] The hand was empty and is now holding the white block, the
white block was on top of the purple block and is now in the hand, the
white block is no longer clear, and the purple block is now clear.
[STATE 1] I have that, the purple block is clear, the cyan block is
clear, the brown block is clear, the hand is holding the white block,
the white block is in the hand, the purple block is on the table, the
cyan block is on the table and the brown block is on the table.

[SCENARIO 2]
[STATE 0] I have that, the purple block is clear, the cyan block is
clear, the white block is clear, the hand is empty, the cyan block is
on top of the brown block, the purple block is on the table, the white
block is on the table and the brown block is on the table.
[ACTION] Unstack the cyan block from on top of the brown block.
[CHANGE] The hand was empty and is now holding the cyan block, the
cyan block was on top of the brown block and is now in the hand, the
cyan block is no longer clear, and the brown block is now clear.
[STATE 1] I have that, the purple block is clear, the brown block is
clear, the cyan block is in the hand, the white block is clear, the
hand is holding the cyan block, the purple block is on the table, the
white block is on the table and the brown block is on the table.

```
┌──────────── Model prompt template, "Put down" action ────────────┐
I am playing with a set of blocks where I need to arrange the blocks
into stacks. Here are the actions I can do

Pick up a block
Unstack a block from on top of another block
Put down a block
Stack a block on top of another block

I have the following restrictions on my actions:
I can only pick up or unstack one block at a time.
I can only pick up or unstack a block if my hand is empty.
```

I can only pick up a block if the block is on the table and the block
is clear. A block is clear if the block has no other blocks on top of
it and if the block is not picked up.
I can only unstack a block from on top of another block if the block I
am unstacking was really on top of the other block.
I can only unstack a block from on top of another block if the block I
am unstacking is clear. Once I pick up or unstack a block, I am
holding the block.
I can only put down a block that I am holding.
I can only stack a block on top of another block if I am holding the
block being stacked.
I can only stack a block on top of another block if the block onto
which I am stacking the block is clear. Once I put down or stack a
block, my hand becomes empty.

After being given an initial state and an action, give the new state
after performing the action.

[SCENARIO 1]
[STATE 0] I have that, the white block is clear, the purple block is
clear, the cyan block is in the hand, the brown block is clear, the
hand is holding the cyan block, the white block is on the table, the
purple block is on the table, and the brown block is on the table.
[ACTION] Put down the cyan block.
[CHANGE] The hand was holding the cyan block and is now empty, the
cyan block was in the hand and is now on the table, and the cyan block
is now clear.
[STATE 1] I have that, the cyan block is clear, the purple block is
clear, the white block is clear, the brown block is clear, the hand is
empty, the white block is on the table, the purple block is on the
table, the cyan block is on the table, and the brown block is on the
table.

[SCENARIO 2]
[STATE 0] I have that, the purple block is clear, the black block is
in the hand, the white block is clear, the hand is holding the black
block, the white block is on top of the brown block, the purple block
is on the table, and the brown block is on the table.
[ACTION] Put down the black block.

```
[CHANGE] The hand was holding the black block and is now empty, the
black block was in the hand and is now on the table, and the black
block is now clear.
[STATE 1] I have that, the black block is clear, the purple block is
clear, the white block is clear, the hand is empty, the white block is
on top of the brown block, the purple block is on the table, the brown
block is on the table, and the black block is on the table.
```

```
─────────────────── Model prompt template, "Stack" action ───────────────────
I am playing with a set of blocks where I need to arrange the blocks
into stacks. Here are the actions I can do

Pick up a block
Unstack a block from on top of another block
Put down a block
Stack a block on top of another block

I have the following restrictions on my actions:
I can only pick up or unstack one block at a time.
I can only pick up or unstack a block if my hand is empty.
I can only pick up a block if the block is on the table and the block
is clear. A block is clear if the block has no other blocks on top of
it and if the block is not picked up.
I can only unstack a block from on top of another block if the block I
am unstacking was really on top of the other block.
I can only unstack a block from on top of another block if the block I
am unstacking is clear. Once I pick up or unstack a block, I am
holding the block.
I can only put down a block that I am holding.
I can only stack a block on top of another block if I am holding the
block being stacked.
I can only stack a block on top of another block if the block onto
which I am stacking the block is clear. Once I put down or stack a
block, my hand becomes empty.

After being given an initial state and an action, give the new state
after performing the action.

[SCENARIO 1]
[STATE 0] I have that, the white block is clear, the purple block is
clear, the cyan block is in the hand, the brown block is clear, the
hand is holding the cyan block, the white block is on the table, the
purple block is on the table, and the brown block is on the table.
```

```
[ACTION] Stack the cyan block on top of the brown block.
[CHANGE] The hand was holding the cyan block and is now empty, the
cyan block was in the hand and is now on top of the brown block, the
brown block is no longer clear, and the cyan block is now clear.
[STATE 1] I have that, the cyan block is clear, the purple block is
clear, the white block is clear, the hand is empty, the cyan block is
on top of the brown block, the brown block is on the table, the purple
block is on the table, and the white block is on the table.

[SCENARIO 2]
[STATE 0] I have that, the purple block is clear, the black block is
in the hand, the white block is clear, the hand is holding the black
block, the white block is on top of the brown block, the purple block
is on the table, and the brown block is on the table.
[ACTION] Stack the black block on top of the purple block.
[CHANGE] The hand was holding the black block and is now empty, the
black block was in the hand and is now on top of the purple block, the
purple block is no longer clear, and the black block is now clear.
[STATE 1] I have that, the black block is clear, the white block is
clear, the hand is empty, the black block is on top of the purple
block, the white block is on top of the brown block, the brown block
is on the table, and the purple block is on the table.
```

## H.4 TIC-TAC-TOE

**Elite LLM**

```
┌─ Elite prompt, propose n actions ─────────────────────┐
In the game of Tic-Tac-Toe, two players, "X" and "O," alternate
placing their symbols on a 3x3 grid. The objective is to be the first
to get three of their symbols in a row, either horizontally,
vertically, or diagonally. We use numbers to indicate empty positions,
and then replace them with "X" or "O" as moves are made. For example,
an empty board is denoted by

1 | 2 | 3
---------
4 | 5 | 6
---------
7 | 8 | 9

Your task is to identify the optimal position for the next move based
on the current board state. Assume that it's your turn and you're
playing as "{role}". Please make sure the optimal position is EMPTY.
For example, in the following Tic-Tac-Toe Board:
```

```
1 | 2 | 3
---------
4 | X | 6
---------
7 | 8 | 9

Position 5 is occupied by "X". Thus, position 5 is not an optimal
position. Provide only the optimal position in the first line. In the
second line, give a brief explanation for this choice.

Current Tic-Tac-Toe Board:

{state}

Role: {role}

Optimal Position:
```

**Model LLM**

```
┌─── Model prompt, predict next state ───────────┐

Predict the Next State of the Tic-Tac-Toe Board

In a game of Tic-Tac-Toe, two players, "X" and "O," take turns to
place their symbols on a 3x3 grid. Your task is to predict what the
board will look like after a specified move has been made.

Examples
{examples}
Now, Predict the Next State of the Following Tic-Tac-Toe Board:
Initial Tic-Tac-Toe Board:

{state}

Move: Player puts "{role}" in position {action}.

Updated Board:
```

```
┌─── Model prompt, predict opponent's action ───────────┐

In Tic-Tac-Toe, each player takes turns placing their respective
symbols ("X" or "O") on a 3x3 board. Your task is to predict where the
opponent will place their symbol based on their past moves and the
current board state.
```

```
Example
Tic-Tac-Toe Board:

O | X | O
---------
X | O | X
---------
7 | 8 | X

Opponent's Move: "O" in position 7
{examples}
Here's how the Tic-Tac-Toe board currently looks:
Tic-Tac-Toe Board:

{state}

Given the history and current board state, where do you think the
opponent will place their "{role}" next? Please make sure the output
is an empty position without "X" or "O".

Opponent's Move: "{role}" in position
```

**Critic LLM**

```
                    ┌─ Critic prompt, evaluate winner ─┐
Determine the Winner in a Tic-Tac-Toe Game

In Tic-Tac-Toe, two players, "X" and "O" take turns to place their
respective symbols on a 3x3 board. The first player to get three of
their symbols in a row, either horizontally, vertically, or
diagonally, wins the game. Your task is to evaluate the board state
and determine if there is a winner.

Examples

Example
Tic-Tac-Toe Board:

O | X | O
---------
X | X | X
---------
O | O | X
```

```
Question: Is there a winner?

Answer: Let's think step by step.

First row: O X O, no winner
Second row: X X X, X wins

Therefore, "X" wins

Example
Tic-Tac-Toe Board:

X | 2 | O
---------
4 | O | X
---------
O | X | 9

Question: Is there a winner?

Answer: Let's think step by step.

First row: X 2 O, no winner
Second row: 4 O X, no winner
Third row: O X 9, no winner
First column: X 4 O, no winner
Second column: 2 O X, no winner
Thrid column: O X 9, no winner
Main diagonal: X O 9, no winner
Anti-diagonal: O O O, O wins

Therefore, "O" wins.
{examples}
Now, for the Current Tic-Tac-Toe Board:
Tic-Tac-Toe Board:

{state}

Question: Is there a winner?

Answer: Let's think step by step.
```

```
┌─ Critic prompt, evaluate tie (when there is no winner) ─┐
In the game of Tic-Tac-Toe, two players alternate turns to fill a 3x3
grid with their respective symbols: "X" and "O". A board is considered
"completely filled" when all nine cells of the grid contain either an
'X' or an 'O', with no empty spaces or other characters.

Examples:
{examples}
Now for the Current Tic-Tac-Toe Board:
Tic-Tac-Toe Board:

{state}

Is the board completely filled?

Answer:
```

