# OpenReview forum: "Reason for Future, Act for Now: A Principled Architecture for Autonomous LLM Agents"
_ICLR.cc/2024/Conference — Submitted to ICLR 2024_

### Official Review · Reviewer_3FoC · 2023-10-23

**Soundness:** 3 good
**Presentation:** 4 excellent
**Contribution:** 2 fair
**Rating:** 5
**Confidence:** 4

**Summary:**

This paper proposes a framework called RAFA that combines model-based reasoning and acting for autonomous LLM agents. The key idea is to use the LLM to plan a future trajectory that maximizes long-term rewards, take the first action from the planned trajectory, collect feedback, and replan at each new state. Theoretical analysis shows the regret bound under a series of assumptions. The method is evaluated on text-based environments like game of 24, ALFWorld, BlocksWorld, and TicTacToe.

**Strengths:**

- The idea of combining model-based planning and short-term execution is logical and aligns well with model predictive control techniques in deep RL.
- Theoretical analysis accounts for limitations like approximate planning and partial observability. Derives regret bound under stated assumptions.
- Overall presentation, structure, and writing quality are good. Key ideas and algorithms are clearly explained.

**Weaknesses:**

1. The proposed model-based planning framework is identical to existing techniques like PETS [1], Planet [2], and Plan2Explore [3] from the deep RL literature. These prior works are not mentioned or compared anywhere in the paper, which significantly weakens the novelty claims. PETS, Planet, and Plan2Explore also learn dynamics models to plan future trajectories within a model predictive control framework [4]. The lack of citation and comparison with these highly relevant prior works makes the technical contributions unclear. I would also expect to see the authors compare with these traditional approaches that do not rely on LLM.

2. The assumptions required for the theoretical results are very strong and may not perfectly hold in practice. For instance, Assumption 4.1 requires the LLM to accurately reconstruct the true transition dynamics from the provided prompts. However, black-box LLM models have imperfect mathematical reasoning capabilities, so perfectly modeling the dynamics via prompting is unrealistic for complex environments with high-dimensional and highly stochastic dynamics. The authors provide no empirical evidence or analysis on how well this assumption actually holds. If the core assumptions are violated, then the theoretical results lose significance since similar analyses exist for traditional model-based RL methods.

3. The experimental validation uses simple toy domains like game of 24, ALFWorld, BlocksWorld and TicTacToe. These environments seem trivial for planning and optimization algorithms to solve. The necessity of using a powerful LLM is unclear when traditional planners could potentially succeed. Moreover, the provided prompts contain many examples and solutions, which greatly simplifies the problem. The authors should test the approach on more complex planning tasks where traditional methods fail but the LLM succeeds, such as Minecraft.

4. It is also necessary to study the sensitivity to the number of examples provided in the prompts. Since the prompts contain many examples in a few-shot learning set-up, ablation studies should analyze the scaling of success rate as the number of prompt examples is varied. This would shed light on how much the pre-provided solutions are aiding the LLM versus solving tasks from scratch (or maybe can not solve the problem with very few prompt examples).

The above concerns weaken the contributions and should be addressed.


[1] Chua, Kurtland, et al. "Deep reinforcement learning in a handful of trials using probabilistic dynamics models." Advances in neural information processing systems 31 (2018).

[2] Hafner, Danijar, et al. "Learning latent dynamics for planning from pixels." International conference on machine learning. PMLR, 2019.

[3] Sekar, Ramanan, et al. "Planning to explore via self-supervised world models." International Conference on Machine Learning. PMLR, 2020.

[4] Moerland, Thomas M., et al. "Model-based reinforcement learning: A survey." Foundations and Trends® in Machine Learning 16.1 (2023): 1-118.

**Questions:**

Please see the weaknesses section for the main concerns. I also have some questions as follows:

1. How long does it take to learn for each task? What computing resources are needed (such as memory) and how much would it cost to finish one task? How does the computation consumption compare to traditional planning/optimization methods?

2. Can the authors provide the results on different base LLM models, including both API-based ones and open-source ones?

3. I am also curious how much effort the authors spent on tuning the prompts. For example, how many trials of revising the prompts before it can work for a particular task? What are the observations and what is the strategy to tune the prompts?

---

> ### Author Response · Authors · 2023-11-22
> **Response for Reviewer 3FoC (Part I)**
>
> Thank you for your review. We address your concerns as follows.
>
> **W1** Compare the novelty and performance of RAFA with Traditional RL agents (PETS, Planet, and Plan2Explore); Present tasks are too simple, and traditional RL can handle them.
>
> **A for W1** From the theoretical perspective, our analysis is aimed at discussing the theoretical performance of our proposed LLM mechanisms (RAFA), where we adopt the analysis tools from the Thomspon sampling (PSRL) in RL. Besides, our sample complexity analysis extends the present PSRL to an infinity horizon setting beyond the linear structure assumption, which is still missing in the current literature. Since we want to interpret the theoretical performance of LLM agents, it is important to let the analysis accommodate general function approximation. For the linear setting, we also refine the analysis in the previous literature ([6, 7]), which assumes a bounded reward while considering the reward to be Gaussian distributed. On the other hand, we claim that our major contribution is not on PSRL but on orchestrating reasoning (learning and planning) and acting following the principled approach in RL. Please see the Appendix A for more detailed discussions. Empirically, we do not need to update the parameter nor sample a parameter from the posterior as PETS, Planet, and Plan2Explore. LLM agent makes an implicit Bayesian inference via In-Context learning, which enables LLM agents to achieve good performance with a few interactions. Since our proposed method is a refinement of previously proposed LLM mechanisms, we use the same large language model for a fair comparison. We also add the citations to the work that you mentioned in Appendix A of the updated manuscript.
>
> **W2** Numerical verification of Assumption 4.1 and discussion of the effect when Assumption 4.1 fails in practice.
>
> **A for W2** We remark that we can extend our analysis to accommodate cases where LLM approximates the posterior within an error margin $\iota$, resulting in a bounded additional regret of $\iota \cdot T$. Importantly, our novel RAFA algorithm (Algorithm 1) stands distinctly from prevalent LLM mechanisms ([1], [2], [3], [4]), and our analysis, guided by Algorithm 3 and Assumption 4.1, only elucidates the theoretical performance of RAFA.
>
>
> Besides, we conduct two experiments to numerically verify Assumption 4.1. Since the predictions given by posterior sampling have the contraction property and the variance heterogeneity, we examine these two properties of LLM predictions in multiple settings to study if the distribution of LLM predictions matches the one given by posterior sampling.
>
> **Experiment 1.** We test if GPT-3 can make an implicit Bayesian inference for the linear regression problem. We first generate i.i.d. data $\{(x_i,y_i)\}^N_{i = 1}$ from $x_i \sim \mathcal{N}(0.8,1),\ y_i\sim \mathcal N(0.5x_i,0.04)$. Then, we feed the following prompt to GPT-3 API (temperature=1.0, top_p=0.8, these parameter configurations can be found in [7]) to obtain 200 predictions when $x_{\text{pred}} = 1$. Specifically, the prompts are "The following data is for linear regression: $(x_1, y_1), (x_2, y_2), \cdots, (x_N, y_N), (1.000000, $". Then, we plot the histogram of the LLM-generated predictions and the density function of the calculated $y_{\text{pred}}$ by posterior sampling, where we choose the prior of the coefficient as $\mathcal N(0,0.5)$. We report the results in Figure 8 of the updated manuscript. We find that the histogram of LLM predictions approximately matches the theoretical distribution of the prediction given by the posterior sampling, which supports Assumption 4.1. Also, with an increasing number of generated data, the variance of LLM predictions decays, which shows the contraction property. We also examine variance heterogeneity in the two-dimensional Bayesian linear regression setting. The variance heterogeneity of predictions helps LLMs give more diverse answers when the current prompt contains less relevant information. \texttt{RAFA} relies on the diversity of LLM predictions to guarantee the exploration for those states less explored, hence we wish the variance of LLM predictions for the less explored states is higher. Please check more details and results in Appendix D of the updated manuscript.

---

> > ### Author Response · Authors · 2023-11-22
> > **Response for Reviewer 3FoC (Part II)**
> >
> > **Experiment 2.** We also test if the LLM performs an implicit Bayesian inference in the ALFWorld environment. We begin by examining the contraction property in the ALFWorld environment. Specifically, the model LLM is prompted to predict the observation of a certain action "go to countertop 1". The prediction result is provided in Figure 9 of the updated manuscript. We find that as more action-observation paris are added to the GPT-3 prompt when timestep increases, the number of distinct prediction responses decreases, i.e., the model LLM has decreasing uncertainty of what objects a certain place has after observing the locations of objects at other places. This corresponds to a decreasing entropy and contraction of the posterior distribution, which supports Assumption 4.1. Besides, to show the variance heterogeneity of the predictions, we prompt the model LLM to predict the next observations when taking actions that the agent has less relevant information. Specifically, in the trajectory that the agent has never explored any dining table, we prompt the model LLM to predict the observation after taking the action "go to dining table 1" at different timesteps. The result is shown in Figure 10 of the updated manuscript. We can observe that the number of distinct predictions of taking this action is much larger than the "go to countertop 1" action where similar locations such as "countertop 2" and "countertop 3" are explored by the agent in this episode. This indicates the variance heterogeneity of the predictions.
> >
> > **W3 & W4** Test the approach on more complex planning tasks such as Minecraft and study the sensitivity to the number of examples provided in the prompts.
> >
> > **A for W3 & W4** We thank the reviewer for pointing these out and we will add these experiments to the final version.
> >
> > **Q1** How long does it take to learn for each task? What computing resources are needed? How does the computation consumption compare to traditional planning/optimization methods?
> >
> > **A for Q1** Since we leverage the in-context learning ability of LLMs, there is no explicit learning procedure needed. The local computation/memory cost is negligible and the major cost is the inference procedure with LLMs. To investigate the computation complexity of our proposed method, we compare the token complexity for each method.  We compare the token efficiency of our method on the Game of 24 with various baseline methods and the results are shown in Appendix G.1 of the updated manuscript. RAFA is superior in terms of token complexity. Methods that lack planning like Reflexion have a low token demand, however, it is not enough to compensate for the drop in performance. Methods that lack in-context learning like [1] would generate unnecessary repeated trials due to a lack of reflection and improvement, which makes the method token inefficient. The computation cost of our method is generally lower compared to traditional planning/optimization methods since we require much fewer episodes to finish the task.

---

> > > ### Author Response · Authors · 2023-11-22
> > > **Response for Reviewer 3FoC (Part III)**
> > >
> > > Here are the references.
> > >
> > > [1] Yao, Shunyu, et al. "Tree of thoughts: Deliberate problem solving with large language models." arXiv preprint arXiv:2305.10601 (2023).
> > >
> > > [2] Yao, Shunyu, et al. "React: Synergizing reasoning and acting in language models." arXiv preprint arXiv:2210.03629 (2022).
> > >
> > > [3] Shinn, Noah, Beck Labash, and Ashwin Gopinath. "Reflexion: an autonomous agent with dynamic memory and self-reflection." arXiv preprint arXiv:2303.11366 (2023).
> > >
> > > [4] Sun, Haotian, et al. "AdaPlanner: Adaptive Planning from Feedback with Language Models." arXiv preprint arXiv:2305.16653 (2023).
> > >
> > > [5] Gruver, Nate, et al. "Large language models are zero-shot time series forecasters." arXiv preprint arXiv:2310.07820 (2023).
> > >
> > > [6] Lu, Xiuyuan, and Benjamin Van Roy. "Information-theoretic confidence bounds for reinforcement learning." Advances in Neural Information Processing Systems 32 (2019).
> > >
> > > [7] Russo, Daniel, and Benjamin Van Roy. "Learning to optimize via posterior sampling." Mathematics of Operations Research 39.4 (2014): 1221-1243.

---

> > > > ### Author Response · Authors · 2023-11-23
> > > > **A new experiment is added**
> > > >
> > > > We are writing to kindly remind that we also add a new experiment in a two-arm bandit setting to make a clearer comparison with LLMs, Thompson sampling (posterior sampling), and Bayesian model averaging. Please see the details and plot in Appendix D of the updated manuscript. The result shows that Thompson sampling (posterior sampling) and LLMs perform better than Bayesian model averaging due to a more efficient exploration, which also numerically supports Assumption 4.1. We hope these clarifications solve your concern and are pleased to have more discussions.

---

### Official Review · Reviewer_Aro2 · 2023-10-30

**Soundness:** 2 fair
**Presentation:** 3 good
**Contribution:** 1 poor
**Rating:** 3
**Confidence:** 4

**Summary:**

The paper presents a novel framework, "Reason for Future, Act for Now" (RAFA), aiming to optimize the actions of large language models (LLMs) in real-world scenarios. This framework is designed to achieve a task using the fewest possible interactions with the external environment by melding reasoning with acting. The theoretical analysis establishes a regret bound for the proposed framework, which is corroborated by superior empirical performance.

**Strengths:**

**In-Context Learning Utilization**: RAFA astutely leverages the inherent in-context learning ability of LLMs to bolster reinforcement learning efficiency.

**Comprehensive Experimental Evidence**: The paper furnishes extensive experimental evaluations that underscore RAFA's efficacy and robustness.

**Weaknesses:**

**Irrelevant Theoretical Analysis**: Although the LLM is seemingly exploited in the theoretical analysis, what the theory really draws upon is actually a posterior inference oracle named LLM. Admittedly, a large transformer model pre-trained on carefully curated dataset may become such an oracle, but the assumption that a pre-trained large language model inherently serves this purpose is questionable. Besides, the analysis claims to draw connections between RAFA and Thompson sampling, but it is not clear how the posterior sampling is accomplished, which is absent from both the main text and the actual implementation in appendix.

**Misrepresentation of Existing Methods**: The paper's assertion that existing techniques like graph-search or MCTS are akin to open-loop methodologies seems misleading. Techniques like MCTS can be effortlessly reconfigured for closed-loop control, wherein planning is grounded in the current knowledge base, and the inaugural action in the plan is executed. This approach essentially mirrors the RAFA framework, calling into question its claimed novelty.

**Questions:**

1. How is the posterior sampling accomplished?
2. How does the RAFA framework tackle the challenge of efficient exploration in its operations?

---

> ### Author Response · Authors · 2023-11-22
> **Response for Reviewer Aro2**
>
> Thank you for your review. We address your concerns as follows.
>
> **Q1 and W1** No verification of Assumption 4.1. How is the posterior sampling accomplished?
>
> **A for Q1 and W1**  First, we want to clarify that Assumption 4.1 casts LLM's In-Context Learning as an implicit Bayesian inference, which finds support in various existing literature [1], [2], [3]. To reinforce its theoretical foundation, we have rigorously verified Assumption 4.1 in Appendix D, outlining the requisite regularity conditions and justifications for each. Furthermore, we extend our analysis to accommodate cases where LLM approximates the posterior within an error margin $\iota$, resulting in a bounded additional regret of $\iota \cdot T$. Importantly, our novel RAFA algorithm (Algorithm 1) stands distinct from prevalent LLM mechanisms ([4], [5], [6], [7]), and our analysis, guided by Algorithm 3 and Assumption 4.1, only elucidates the theoretical performance of RAFA. Besides, we conduct two experiments to numerically verify Assumption 4.1. For more details, please see the first argument in the General Response.
>
> **W2** MCTS can also be implemented in a closed-loop manner.
>
> **A for W2** It is true that MCTS can be implemented in a closed-loop manner to converge for a deterministic environment, which is studied in [8]. However, there is no work that studies how to incorporate closed-loop MCTS with LLMs. Besides, it is still unclear if the closed-loop can achieve a $\sqrt T$ Bayesian regret, as the sample complexity bound in [8] is $\tilde{O} ((1/\epsilon)^d)$ but \texttt{RAFA} only requires $\tilde{O}((1/\epsilon)^2)$ samples to achieve the average $\epsilon$ accuracy, where $d$ is the dimension of the state space. The key difference between \texttt{RAFA} and the closed-loop MCTS is the method of model and critic learning. \textt{RAFA} utilizes ICL to learn the unknown world model and generalizes to the unseen states, which leads to a more efficient exploration and a sharpened regret upper bound.
>
> **Q2** How does the RAFA framework tackle the challenge of efficient exploration in its operations?
>
> **A for Q2** Since we cast the ICL of LLMs as an implicit Bayesian inference, RAFA explores the environment via the internal randomness of the LLM responses, which is similar to the posterior sampling methods. We also conduct additional experiments to better understand the connections between the randomness of LLM predictions and the exploration of the environment. Specifically, in the ALFWorld environment, the model LLM is prompted to predict the observation of a certain action "go to countertop 1". In Figure 9 of the updated manuscript, we report the diversity of LLM predictions. We find that as more action-observation paris are added to the GPT-3 prompt when timestep increases, the number of distinct prediction responses decreases, i.e., the model LLM has decreasing uncertainty of what objects a certain place has after observing the locations of objects at other places. This corresponds to a decreasing entropy, indicating that the variance heterogeneity of predictions LLMs gives more diverse answers when the current prompt contains less relevant information. In other words, \texttt{RAFA} relies on the diversity of LLM predictions to guarantee the exploration for those states less explored. For more setup details and the results, please refer to Appendix D in the updated manuscript.
>
>
> [1] Zhang, Yufeng, et al. "What and How does In-Context Learning Learn? Bayesian Model Averaging, Parameterization, and Generalization." arXiv preprint arXiv:2305.19420 (2023).
>
> [2] Lee, Jonathan N., et al. "Supervised Pretraining Can Learn In-Context Reinforcement Learning." arXiv preprint arXiv:2306.14892 (2023).
>
> [3] Hollmann, Noah, et al. "Tabpfn: A transformer that solves small tabular classification problems in a second." arXiv preprint arXiv:2207.01848 (2022).
>
> [4] Yao, Shunyu, et al. "Tree of thoughts: Deliberate problem solving with large language models." arXiv preprint arXiv:2305.10601 (2023).
>
> [5] Yao, Shunyu, et al. "React: Synergizing reasoning and acting in language models." arXiv preprint arXiv:2210.03629 (2022).
>
> [6] Shinn, Noah, Beck Labash, and Ashwin Gopinath. "Reflexion: an autonomous agent with dynamic memory and self-reflection." arXiv preprint arXiv:2303.11366 (2023).
>
> [7] Sun, Haotian, et al. "AdaPlanner: Adaptive Planning from Feedback with Language Models." arXiv preprint arXiv:2305.16653 (2023).
>
> [8] Shah, Devavrat, Qiaomin Xie, and Zhi Xu. "Non-asymptotic analysis of monte carlo tree search." Abstracts of the 2020 SIGMETRICS/Performance Joint International Conference on Measurement and Modeling of Computer Systems. 2020.

---

> > ### Comment · Reviewer_Aro2 · 2023-11-22
> >
> > Thank you for your effort in addressing the concerns raised in my previous comments. While your rebuttal has shed some light on the validity of your assumptions, I feel that my primary concern regarding the posterior sampling in Thompson sampling was not fully addressed. Allow me to clarify further.
> >
> > Thompson sampling necessitates the sampling of a plausible model from the posterior distribution and deriving its optimal policy. By setting a fixed model prompt and planning with it, RAFA is essentially trying to solve for the optimal policy of a posterior averaged model, rather than exploring one potential model. This fundamental discrepancy between theorety and implementation calls into question the relevance of your analysis.
> >
> > From a practical standpoint, RAFA, in its current form, appears to function similarly to a closed-loop MCTS integrated with an LLM. You assert that the critical distinction between RAFA and traditional closed-loop MCTS lies in the application of in-context learning. However, this claim is not entirely persuasive. It appears to me that any closed-loop MCTS implementation utilizing LLM would inherently involve some form of in-context learning.
> >
> > I hope these points provide a clearer picture of the areas where further explanation and evidence are needed. I look forward to your response.

---

> > > ### Author Response · Authors · 2023-11-23
> > > **Updated Response for Reviewer Aro2**
> > >
> > > Thank you for your updated response and review of our revisions. We address your concerns as follows.
> > >
> > > We clarify that the added experiments in Appendix D of the updated manuscript support the claim that LLM is equivalent to posterior sampling beyond Bayesian model averaging. This is because Bayesian model averaging may not have variance heterogeneity, while LLMs reveal this important feature of posterior sampling. The variance heterogeneity of predictions helps LLMs give more diverse answers when the current prompt contains less relevant information. \texttt{RAFA} relies on the diversity of LLM predictions to guarantee the exploration for those states less explored, hence we wish that the variance of LLM predictions for the less explored states is higher. Experiments in Appendix D show that LLMs have variance heterogeneity, which is an important feature of posterior sampling. To make a clearer comparison with LLMs, posterior sampling, and Bayesian model averaging, we conduct a two-arm bandit experiment to test the performance of these three methods. Please see the details and plot in Appendix D of the updated manuscript. The result shows that posterior sampling and LLMs perform better than Bayesian model averaging due to a more efficient exploration, which also numerically supports Assumption 4.1.
> > >
> > > As for the closed-loop MCTS with LLMs, we remark that not an arbitrary combination of LLMs and MCTS in a closed-loop can enjoy both good theoretical properties and empirical performance. RAFA casts MCTS as a specific planner, where LLMs function as transition model, actor, and critics, rather than only one or two separately. These LLMs are fed with historical trajectories to give a better response via ICL, synergize reasoning, and act in a closed-loop manner. Also, there is no (theoretical or empirical) work that studies how to incorporate closed-loop MCTS with LLMs and our work provides the first solution for incorporating MCTS with LLMs in a closed-loop manner.
> > >
> > > Thank you again for your response.

---

### Official Review · Reviewer_n4NZ · 2023-10-31

**Soundness:** 3 good
**Presentation:** 1 poor
**Contribution:** 2 fair
**Rating:** 6
**Confidence:** 3

**Summary:**

This paper proposes to cast the problem of designing language model agents as an RL problem. The proposed algorithm RAFA leverages the in-context learning ability of LLM to do model-based posterior sampling reinforcement learning (PSRL). Specifically, RAFA builds a memory $\mathcal{D}$ of environment transitions and then uses an LLM to infer the environment dynamic in a novel, "non-parametric" and "in-context" fashion.

The policy rollout involves three components that are all in-context learning algorithms that are conditioned on the current $\mathcal{D}$:

- A **model**  predicts the next state conditioned on an input state-action pair
- A **critic** predicts the value of transitions/trajectories
- An **elite** proposes a number of potential actions based on an input state.

During planning, the algorithm uses an MPC-style action selection. At each step of the rollout of length $U$,  the elite proposes $B$ action for $s_u$ and the model predicts the next state for each action. At the end of the rollout, the critic evaluates the value of each trajectory and selects the initial action $a_0$ that leads to the highest cumulative return. Then the agent takes a_0 and adds the new transition to $\mathcal{D}$.

The paper then proceeds to analyze the theoretical properties of the proposed algorithm and the associated assumptions and shows that the algorithm achieves $O(\sqrt{T})$ regret where $T$ is the number of environment steps. The proof methodologies employed are reminiscent of those typically found in standard PSRL literature., although I have not thoroughly checked the correctness of every detail in the appendix due to the time constraint.

The paper concludes with empirical assessments across four benchmarks: Game of 24, AlfWorld, BlockWorld, and Tic-Tac-Toe. In these evaluations, RAFA demonstrates superior performance when juxtaposed with selected baseline algorithms, occasionally by a significant margin.

**Strengths:**

Overall, I think the idea of this paper is quite neat. Formalizing the LLM reasoning through RL could be the first step towards formalizing the emergent but rapidly growing fields and laying a good theoretical and conceptual foundation. The proposed algorithm makes intuitive sense and the empirical performance seems competitive. I can see that many future works could build on this framework of treating LLM reasoning as RL.

**Weaknesses:**

The paper presents an interesting concept; however, there are areas that require further attention and clarification to fully realize its potential. I would like to offer a constructive critique on some aspects that, if addressed, could enhance the submission.

**Theoretical contribution**

Overall, my biggest concern is with the theoretical claims made by the paper. My concern is not with the correctness of the paper but how much utility the theoretical results have for the actual performance of RAFA. While the theoretical foundations of the paper appear sound, the practical implications of these theories on the RAFA algorithm's performance require further exploration. It is essential to clearly articulate the direct benefits of the theoretical results to the algorithm to prevent any misinterpretations about their significance.
The paper is currently written in a way that could lead to unknowledgeable readers thinking that this theory (e.g., regret guarantee) is crucial for the algorithm. My primary research area is not RL theory but I am reasonably familiar with the main developments in RL theory. It seems to me that the theoretical component is not particularly innovative in the context of existing PSRL literature ( for some examples, [1,2,3]). If the authors do not agree with my assessment, then I think it would be good to have a more detailed and explicit discussion about how the theoretical result of this work relates to existing works in PSRL.

Since all of the theory is rooted in generic RL, where does LLM come in? I believe that the answer is entirely in the assumptions made by the theoretical results, specifically, Assumption 4.1 and those in Appendix D. Assumption 4.1 assumes that LLM is capable of doing generic posterior sampling given any $\mathcal{D}$. This assumption seems overly optimistic, considering the fact that, in general, good posterior sampling is computationally challenging in all but the simplest cases. The paper cites [4] and [5] as supporting evidence that LLMs can do posterior sampling on natural text data, but [4] only uses a small-scale synthetic dataset, and [5] updates a continuous embedding to approximate $\theta$, neither of which seems to be close to the kind of posterior sampling ability required by this work. Of course, it is entirely possible that LLMs can indeed do the kind of posterior sampling required here, but in that case, I think more thorough empirical verification is needed. For example, one can construct a simple environment where one can compute posterior sampling exactly and measure how well the results produced by the LLM match the analytical solution.

Moreover, the empirical setting used to demonstrate the algorithm's performance involves a relatively small number of environment steps, which raises questions about the applicability of the regret analysis to the algorithm's real-world performance. Additional empirical evidence in this area could substantiate the theoretical claims.


**Formatting**

The current formatting of the paper detracts from its readability and overall presentation. The text is densely packed, and the spacing between figures and tables is too narrow, often resulting in severe margin violation.  The absence of a conclusion section is also notable. I recommend a revision of the manuscript to address these formatting issues. This might involve condensing the theoretical content in the main body to allocate space for a conclusion and to resolve the formatting challenges. As I mentioned earlier, the regret guarantee, while interesting, may not be as central to the understanding and application of the algorithm as suggested. I am open to further discussion on this point if the authors have a different perspective.



## Reference

[1] Model-based RL with Optimistic Posterior Sampling: Structural Conditions and Sample Complexity. Agarwal et al.

[2] Model-based Reinforcement Learning for Continuous Control with Posterior Sampling. Fan et al.

[3] (More) Efficient Reinforcement Learning via Posterior Sampling. Osband et al.

[4] An Explanation of In-context Learning as Implicit Bayesian Inference. Xie et al.

[5] Large Language Models Are Latent Variable Models: Explaining and Finding Good Demonstrations for In-Context Learning. Wang et al.

**Questions:**

- does this value iteration using LLM as the critic converge?

- I don't fully understand how $\mathcal{D}$ is being incorporated into the prompts for the model, critic, and elite even after looking at the example in the appendix. Could you give me a concrete example?

- Why sometimes do you use MCTS and sometimes you use other tree searches? How should the users choose between them and what are the trade-offs?

- Why do you say you search all sequences in the pseudocode? Wouldn't that create $B^U$ number of trajectories which is not scalable at all?

- How many queries are needed for each run? In traditional RL, the number of environmental steps is important because that is the bottleneck. I am not sure if this is the case here so the number of environment steps seems less useful. For a more fair comparison to the baselines, I think it would be fairer to compare the number of queries vs return to rule out the possibility that other methods can get better performance via more queries.

---

> ### Author Response · Authors · 2023-11-22
> **Response for Reviewer n4NZ (Part I)**
>
> Thank you for your review. We address your concerns as follows.
>
> **W1** This paper is less innovative in RL and lacks discussions with PSRL.
>
> **A for W1** From the theoretical perspective, our analysis is aimed at discussing the theoretical performance of our proposed LLM mechanisms (RAFA), where we adopt the analysis tools from the Thomspon sampling (PSRL) in RL. Besides, our sample complexity analysis extends the present PSRL to an infinity horizon setting beyond the linear structure assumption, which is still missing in the current literature. Since we want to interpret the theoretical performance of LLM agents, it is important to let the analysis accommodate general function approximation. For the linear setting, we also refine the analysis in the previous literature ([1, 2]), which assumes a bounded reward while considering the reward to be Gaussian distributed. On the other hand, we claim that our major contribution is not on PSRL but on orchestrating reasoning (learning and planning) and acting following the principled approach in RL. Please see the Appendix A for more detailed discussions. Empirically, we do not need to update the parameter nor sample a parameter from the posterior. LLM agent makes an implicit Bayesian inference via In-Context learning, which enables LLM agents to achieve good performance with a few interactions. Since our proposed method is a refinement of previously proposed LLM mechanisms, we use the same large language model for a fair comparison.
>
> **W2** This paper also lacks numerical verification of Assumption 4.1.
>
> **A for W2**
> We conduct two experiments to numerically verify Assumption 4.1. Since the predictions given by posterior sampling have the contraction property and the variance heterogeneity, we examine these two properties of LLM predictions in multiple settings to study if the distribution of LLM predictions matches the one given by posterior sampling.
>
> **Experiment 1.** We test if GPT-3 can make an implicit Bayesian inference for the linear regression problem. We first generate i.i.d. data $[(x_i,y_i)]^N_{ i = 1}$ from $x_i \sim \mathcal{N}(0.8,1), y_i\sim \mathcal N(0.5x_i,0.04)$. Then, we feed the following prompt to GPT-3 API (temperature=1.0, top_p=0.8, these parameter configurations can be found in [7]) to obtain 200 predictions when $x_{\text{pred}} = 1$. Specifically, the prompts are "The following data is for linear regression: $(x_1, y_1), (x_2, y_2), \cdots, (x_N, y_N), (1.000000, $". Then, we plot the histogram of the LLM-generated predictions and the density function of the calculated $y_{\text{pred}}$ by posterior sampling, where we choose the prior of the coefficient as $\mathcal N(0,0.5)$. We report the results in Figure 8 of the updated manuscript. We find that the histogram of LLM predictions approximately matches the theoretical distribution of the prediction given by the posterior sampling, which supports Assumption 4.1. Also, with an increasing number of generated data, the variance of LLM predictions decays, which shows the contraction property. We also examine variance heterogeneity in the two-dimensional Bayesian linear regression setting. The variance heterogeneity of predictions helps LLMs give more diverse answers when the current prompt contains less relevant information. \texttt{RAFA} relies on the diversity of LLM predictions to guarantee the exploration for those states less explored, hence we wish the variance of LLM predictions for the less explored states is higher. Please check more details and results in Appendix D of the updated manuscript.

---

> > ### Author Response · Authors · 2023-11-22
> > **Response for Reviewer n4NZ (Part II)**
> >
> > **Experiment 2.** We also test if the LLM performs an implicit Bayesian inference in the ALFWorld environment. We begin by examining the contraction property in the ALFWorld environment. Specifically, the model LLM is prompted to predict the observation of a certain action "go to countertop 1". The prediction result is provided in Figure 9 of the updated manuscript. We find that as more action-observation paris are added to the GPT-3 prompt when timestep increases, the number of distinct prediction responses decreases, i.e., the model LLM has decreasing uncertainty of what objects a certain place has after observing the locations of objects at other places. This corresponds to a decreasing entropy and contraction of the posterior distribution, which supports Assumption 4.1. Besides, to show the variance heterogeneity of the predictions, we prompt the model LLM to predict the next observations when taking actions that the agent has less relevant information. Specifically, in the trajectory that the agent has never explored any dining table, we prompt the model LLM to predict the observation after taking the action "go to dining table 1" at different timesteps. The result is shown in Figure 10 of the updated manuscript. We can observe that the number of distinct predictions of taking this action is much larger the "go to countertop 1" action where similar locations such as "countertop 2" and "countertop 3" are explored by the agent in this episode. This indicates the variance heterogeneity of the predictions.
> >
> > **W3** Present Tasks involve too small steps and more complex tasks are needed.
> >
> > **A for W3** We evaluate our proposed LLM mechanism on similar benchmarks with previously proposed LLM agents ([4],[5],[6]) for a fair comparison. Even though LLM agents solve these tasks with a few interactions, RL agent cannot easily cope with them. Particularly, we compare RAFA and other baseline LLM agents with an RL agent BUTLER in Table 2 in the paper, where the most basic LLM agent performs better than BUTLER. A similar result can be found in [7].
> >
> > **Q1** Does this value iteration using LLM as the critic converge?
> >
> > **A for Q1** In Algorithm 6, we incorperate LLMs as reward function $r_\theta$ and model $P_\theta$ in the truncated-horizon value iteration, which is proved to converge according to Proposistion B.1 in Appendix B.
> >
> > **Q2** I don't fully understand how the memory buffer is being incorporated into the prompts for the model, critic, and elite even after looking at the example in the appendix. Could you give me a concrete example?
> >
> > **A for Q2** We add all the historical trajectories into the prompt for the model, critic, and elite. If we want let the model LLM predict the next state $s^\prime$ given the current state-action $(s,a)$, we will wrap the historical trajectory and $(s,a)$ as the prompts and feed them to the model LLM.
> >
> > **Q3** Why sometimes do you use MCTS and sometimes you use other tree searches? How should the users choose between them and what are the trade-offs?
> >
> > **A for Q3**
> >
> > Our method is compatible with various search methods and we use different search methods for different tasks to align with other baselines for a fair comparison. For example, ToT [5] uses beam search and RaP [4] uses MCTS. As a general guideline, for relatively simple tasks like ALFWorld, beam search or even simple BFS is sufficient, which is also easy to implement and token efficient. For complex tasks with large action space and long horizons, MCTS is required to reduce the planning complexity.
> >
> > **Q4** Why do you say you search all sequences in the pseudocode? Wouldn't that create
> >  number of trajectories $B^U$ which is not scalable at all?
> >
> > **A for Q4** We use the basic BFS algorithm as an example for the planning module, which is applicable when the search space is small (as used in ALFWorld). We also employ advanced search algorithms like beam search and MCTS, as shown in Algorithm 4 and 5 in Appendix B, which is efficient and scalable.

---

> > > ### Author Response · Authors · 2023-11-22
> > > **Response for Reviewer n4NZ (Part III)**
> > >
> > > **Q5** How many queries are needed for each run? In traditional RL, the number of environmental steps is important because that is the bottleneck. I am not sure if this is the case here so the number of environment steps seems less useful. For a more fair comparison to the baselines, I think it would be fairer to compare the number of queries vs return to rule out the possibility that other methods can get better performance via more queries.
> > >
> > > **A for Q5** As suggested, we compare the token complexity for each method. Our method achieves a better token complexity compared with baseline methods. We compare the token efficiency of our method on the Game of 24 with various baseline methods and the results are shown in Appendix G.1 of the updated manuscript. RAFA is superior in terms of token complexity. Methods that lack planning like Reflexion have a low token demand, however it is not enough to compensate for the drop in performance. Methods that lack in-context learning like Tree of Thoughts would generate unnecessary repeated trials due to lack of reflection and improvement, which makes the method token inefficient.
> > >
> > >
> > >
> > > [1] Lu, Xiuyuan, and Benjamin Van Roy. "Information-theoretic confidence bounds for reinforcement learning." Advances in Neural Information Processing Systems 32 (2019).
> > >
> > > [2] Russo, Daniel, and Benjamin Van Roy. "Learning to optimize via posterior sampling." Mathematics of Operations Research 39.4 (2014): 1221-1243.
> > >
> > > [3] Gruver, Nate, et al. "Large language models are zero-shot time series forecasters." arXiv preprint arXiv:2310.07820 (2023).
> > >
> > > [4] Hao, Shibo, et al. "Reasoning with language model is planning with world model." arXiv preprint arXiv:2305.14992 (2023).
> > >
> > > [5] Yao, Shunyu, et al. "Tree of thoughts: Deliberate problem solving with large language models." arXiv preprint arXiv:2305.10601 (2023).
> > >
> > > [6] Shinn, Noah, Beck Labash, and Ashwin Gopinath. "Reflexion: an autonomous agent with dynamic memory and self-reflection." arXiv preprint arXiv:2303.11366 (2023).
> > >
> > > [7] Shridhar, Mohit, et al. "Alfworld: Aligning text and embodied environments for interactive learning." arXiv preprint arXiv:2010.03768 (2020).

---

> ### Comment · Reviewer_n4NZ · 2023-11-22
> **Thank you for the response and revision**
>
> I sincerely thank the authors for their hard work on the revision and additional experiments. They certainly addressed some of my concerns (e.g., token efficiency).  The new verification (in particular, ALFworld) for Assumption 4.1, while still limited, at least provides some amount of evidence that LLM can do something similar to posterior sampling. I think the paper would be a lot stronger if this is done more thoroughly but I understand it's hard to do some in such short period of time. For example, I would like to see a head-to-head comparison on a simple RL task of LLM and actual posterior sampling (e.g., Figure 9). I would also somewhat disagree that the extension from finite horizon case to infinite horizon case is as signifianct as the authors claim, since one can apply the $\frac{1}{1-\gamma}$ horizon argument for the infinite horizon case (although this is a more subjective opinion). Not to mention that the all of the tested environment's horizons are very short.
>
> More importantly, the formatting of the paper is still unacceptable. Margins are violated in pg 2, 8, and 9. On page 9, the top line of the ICLR template is completely missing and everythinhg is still crampped together. Even to make this little space for a conclusion, **the authors removed the anonymous author region from the first page**. Per ICLR guideline, the final paper *does not* have an additional page for the camera ready so there is no way that this paper will fit within 9 pages in its current form. Page limit exists for a reason and I would encourage the authors to adhere to the guidelines. In this current sate, I cannot in good conscience recommend acceptance, but I would consider raising my score to 6 if the author can at least make the paper follow the format guideline.

---

> > ### Author Response · Authors · 2023-11-22
> > **Thank you for the updated review.**
> >
> > Thank you for your updated response and review of our revision. We have revised the format of our manuscript according to your review and the ICLR instructions. If you have more suggestions, we are pleased to make more revisions.
> >
> > We also want to clarify that the gap between the finite horizon RL and infinite horizon RL is not trivial beyond the linear structure. In the $H$-horizon setting, the agent updates its knowledge (model or Q-function) each episode and uses the knowledge for planning over $H$ steps. It suggests that the agent takes a different policy at each episode $k\in[K]$. If we simply set $H=1/(1-\gamma)$ and use the finite-horizon RL algorithm to solve the infinite horizon setting task, the value inconsistency in Lemma E.1 in the manuscript can only be trivially bounded by $\tilde{O}(K)$ without more assumptions. As the number of total timesteps is $T=KH=K/(1-\gamma)$, the value inconsistency incurs a linear term of $T$ on the regret, which causes sample inefficiency.
> >
> > Hence, we need to incorporate the lazy update method to make the agent update its policy at a lower frequency compared with the finite-horizon RL algorithm. With specific structure assumptions, [1] and [2] show that the agent only needs to update $K=\tilde{O}(\log(T))$ times of policy (the agent only updates its policy if the switching condition is satisfied), hence the value inconsistency can be bounded by $K=\tilde{O}(\log(T))$, which is crucial for the sublinear regret bound. As our scope in this paper is to interpret RAFA from the theoretical perspective, their structure assumptions and the switching conditions (relying on calculating the determinants) are too strong an assumption for LLMs. Hence, we extend the analysis and the switching condition to a general case, leveraging the definition of entropy and information gain. In practice, information gain can be approximated by checking the diversity of LLM responses or detecting the failure of the current task. When RAFA fails the current task, it is likely that the LLM faces new formation (information gain is high) and needs to update its policy.
> >
> > Thank you again for your response.
> >
> > [1] Abbasi-Yadkori, Yasin, and Csaba Szepesvári. "Bayesian Optimal Control of Smoothly Parameterized Systems." UAI. 2015.
> >
> > [2] Zhou, Dongruo, Jiafan He, and Quanquan Gu. "Provably efficient reinforcement learning for discounted mdps with feature mapping." International Conference on Machine Learning. PMLR, 2021.

---

> > > ### Author Response · Authors · 2023-11-23
> > > **A new experiment is added**
> > >
> > > We are writing to kindly remind that we also add a new experiment in a two-arm bandit setting to make a clearer comparison with LLMs, Thompson sampling (posterior sampling), and Bayesian model averaging. Please see the details and plot in Appendix D of the updated manuscript. The result shows that Thompson sampling (posterior sampling) and LLMs perform better than Bayesian model averaging due to a more efficient exploration, which also numerically supports Assumption 4.1. We hope these clarifications solve your concern and are pleased to have more discussions.

---

> ### Comment · Reviewer_n4NZ · 2023-11-23
>
> I thank the authors for the clarification, the manuscript revision, and the new experiments. If the information gain is indeed novel, then I think it would be a good addition to the literature. Since I am not an expert in this area, I will defer this judgment to others.
>
> The last page is still a bit rough but at least there are no egregious formatting violations anymore. The additional experiments (in particular contextual bandits) instill some confidence that LLMs can do posterior sampling in this case although it is evident that there is a visible gap.
>
> As promised, I will raise my score to 6. However, I still strongly recommend that the authors make the last page more readable, and take other reviewers' feedback into consideration. I wish the authors the best of luck.

---

> > ### Author Response · Authors · 2023-11-23
> >
> > We thank Reviewer n4NZ for updating the score. We will try to make the last page more readable in the later revision and have tried to address other reviewers' concerns. Thank you for the discussions.

---

### Official Review · Reviewer_P2KV · 2023-10-31

**Soundness:** 2 fair
**Presentation:** 3 good
**Contribution:** 2 fair
**Rating:** 5
**Confidence:** 3

**Summary:**

The paper presents an approach (RAFA) to use the pretrained LLMs as sequential decision-making agents, by designing a system in which LLM instances are asked to first generate and evaluate a plan, and then to execute it in the environment. The paper includes a theoretical analysis based on Bayesian regret bounds and an empirical investigation in text-based domains.

**Strengths:**

- I find the general perspective of the paper to be promising. Analyzing LLM-based agents using the conceptual and theoretical tools traditionally associated with reinforcement learning research can provide interesting insights and a more rigorous perspective on modern AI agents.
- The discussion of related work is quite complete.

**Weaknesses:**

- I am skeptical of how the theoretical background and analysis are related at all to the algorithmic and empirical setting. I would expect any meaningful theory about LLM-based agents to capture something about LLMs, but Assumption 4.1 basically says that the theory is going to assume access to the true posterior and ignores anything about the underlying models. I think this completely disconnects the theoretical results from the practice presented in the paper. In addition, I am not entirely sure of how the Bayesian perspective, in which the first part of the algorithm is supposed to estimate the posterior, is connected to the Model-Predictive Control-based algorithm that is actually employed in the paper.
- The paper is so compressed and poor in terms of space and content organization that it is very hard to read. As a notable example, page 9 contains many plots too close one to the other and overflowing beyond the regular page limits and margins; moreover, it does not contain any conclusion, making it hard for the reader to further contextualize and understand these results.
- The empirical evaluation does not seem to be rigorous. First, I do not fully understand why different models (GPT3.5, Vicuna, GPT4) have been employed in the different environments. What was the rationale behind the choice? Why one system and not the other for a specific task? Then, I do not see any error bars in the performance plots and table, nor any information about the number of repetitions of experiments.

**Questions:**

- What are the empirical results under a single LLM model?
- How many repetitions did you use for the empirical results? Do the results vary that much if you have more?
- Is there a way to avoid the complete disconnect between theory and practice?
- Can you reformat the paper to avoid the abuse of vspaces and other techniques that make it hardly readable?

---

> ### Author Response · Authors · 2023-11-22
> **Response for Reviewer P2KV**
>
> Thank you for your review. We address your concerns as follows.
>
> **Q1** What are the empirical results under a single LLM model?
>
> **A for Q1** We have compared our proposed method (RAFA) with Reflexion [1] in all four experiments in this paper, where Reflexion only uses a single LLM with closed-loop interactions. Results can be found in Section 5 and Appendix G. We observed that Reflexion improves much slower compared to RAFA and has an inferior final performance. This is because Reflexion is an oversimplified version of RAFA, where the planning subroutine revises the current action to maximize the reward function (“reason for now”) instead of planning multiple future steps to maximize the value function (“reason for future”), which measures the expected cumulative future reward. Besides, in the ALFWorld environment, we compared RAFA and AdaPlanner [2], which is also a baseline that uses a single LLM model. We found that AdaPlanner has a higher initial performance due to a handcrafted set of programs for rejecting suboptimal candidate trajectories, which, however, is challenging to construct without the domain knowledge of a specific task. Besides, its final success rate is also lower than RAFA. In the BlocksWorld and Tic-Tac-Toe environments, we compared RAFA with the Chain-of-Thought and GPT-4 baselines, respectively. The results showed that RAFA outperforms these baselines by a remarkable margin.
>
> **Q2** How many repetitions did you use for the empirical results? Do the results vary that much if you have more?
>
> **A for Q2** We follow a similar reporting procedure as previous LLM work (such as [2], [3], and [4]) and do not repeat the experiments multiple times. The reason behind this choice is that the results do not vary too much, as opposed to experiments for reinforcement learning methods.  How the Bayesian perspective is connected to the Model-Predictive Control-based algorithm.
>
> **Q3 and W1** Is there a way to avoid the complete disconnect between theory and practice? This paper assumes the access to the true posterior and ignores anything about the underlying models. How to connect
>
> **A for Q3 and W1** First, we want to clarify that Assumption 4.1 casts LLM's In-Context Learning as an implicit Bayesian inference, which finds support in various existing literature [3], [4], [5]. In Appendix D in the manuscript, we prove that Assumption D.1. holds when we pose several regularity conditions on the model and data-generating process. Furthermore, we extend our analysis to accommodate cases where LLM approximates the posterior within an error margin $\iota$, resulting in a bounded additional regret of $\iota \cdot T$. Importantly, our novel RAFA algorithm (Algorithm 1) stands distinctly from prevalent LLM mechanisms ([1], [2], [6], and [7]), and our analysis, guided by Algorithm 3 and Assumption 4.1, only elucidates the theoretical performance of RAFA.
>
>
> Besides, we conduct two experiments to numerically verify Assumption 4.1. Since the predictions given by posterior sampling have the contraction property and the variance heterogeneity, we examine these two properties of LLM predictions in multiple settings to study if the distribution of LLM predictions matches the one given by posterior sampling.
>
> **Experiment 1.** We test if GPT-3 can make an implicit Bayesian inference for the linear regression problem. We first generate i.i.d. data $[(x_i,y_i)]^N_{i = 1}$ from $x_i \sim \mathcal{N}(0.8,1),\ y_i\sim \mathcal N(0.5x_i,0.04)$. Then, we feed the following prompt to GPT-3 API (temperature=1.0, top_p=0.8, these parameter configurations can be found in [7]) to obtain 200 predictions when $x_{\text{pred}} = 1$. Specifically, the prompts are "The following data is for linear regression: $(x_1, y_1), (x_2, y_2), \cdots, (x_N, y_N), (1.000000, $". Then, we plot the histogram of the LLM-generated predictions and the density function of the calculated $y_{\text{pred}}$ by posterior sampling, where we choose the prior of the coefficient as $\mathcal N(0,0.5)$. We report the results in Figure 8 of the updated manuscript. We find that the histogram of LLM predictions approximately matches the theoretical distribution of the prediction given by the posterior sampling, which supports Assumption 4.1. Also, with an increasing number of generated data, the variance of LLM predictions decays, which shows the contraction property. We also examine variance heterogeneity in the two-dimensional Bayesian linear regression setting. The variance heterogeneity of predictions helps LLMs give more diverse answers when the current prompt contains less relevant information. \texttt{RAFA} relies on the diversity of LLM predictions to guarantee the exploration for those states less explored, hence we wish the variance of LLM predictions for the less explored states is higher. Please check more details and results in Appendix D of the updated manuscript.

---

> > ### Author Response · Authors · 2023-11-22
> > **Response for Reviewer P2KV (Part II)**
> >
> > **Experiment 2.** We also test if the LLM performs an implicit Bayesian inference in the ALFWorld environment. We begin by examining the contraction property in the ALFWorld environment. Specifically, the model LLM is prompted to predict the observation of a certain action "go to countertop 1". The prediction result is provided in Figure 9 of the updated manuscript. We find that as more action-observation paris are added to the GPT-3 prompt when timestep increases, the number of distinct prediction responses decreases, i.e., the model LLM has decreasing uncertainty of what objects a certain place has after observing the locations of objects at other places. This corresponds to a decreasing entropy and contraction of the posterior distribution, which supports Assumption 4.1. Besides, to show the variance heterogeneity of the predictions, we prompt the model LLM to predict the next observations when taking actions that the agent has less relevant information. Specifically, in the trajectory that the agent has never explored any dining table, we prompt the model LLM to predict the observation after taking the action "go to dining table 1" at different timesteps. The result is shown in Figure 10 of the updated manuscript. We can observe that the number of distinct predictions of taking this action is much larger than the "go to countertop 1" action where similar locations such as "countertop 2" and "countertop 3" are explored by the agent in this episode. This indicates the variance heterogeneity of the predictions.
> >
> > Next, we answer how RAFA connects the Bayesian perspective and the planning subroutine. In RAFA, the learning subroutine infers the transition
> > and reward models (model) or/and the value function (critic) from the Bayesian perspective (ICL). (b) The planning subroutine generates an optimal policy (actor) or trajectory for multiple future steps, which maximizes the value function (up to a certain error). Through the learning subroutine, the collected feedback reduces the posterior uncertainty in models or values, which allows the planning subroutine to obtain an improved policy at each step. In other words, we emulate the actor-model or actor-critic update for Bayesian adaptive MDPs in an in-context manner, where LLMs function as an internal mechanism that improves models, values, and policies iteratively. Compared with MPC, RAFA does not assume the environment to be known but infers the environment via ICL.
> >
> > **Q4 and W2** Can you reformat the paper to avoid the abuse of vspaces and other techniques that make it hardly readable? The paper is so compressed and contains no conclusion.
> >
> > **A for Q4 and W2** We have revised the format of the paper in the updated manuscript. We are pleased to make more revisions if you have additional suggestions.
> >
> > **W3** Why different base LLM models are chosen in the experiments? Why do not repeat multiple times?
> >
> > **A for W3**
> >  We choose the same base LLM models according to the baselines (such as [1], [2], and [7]) and it is not fairly comparable if we choose another base LLM model. When we compare \texttt{RAFA} with these baselines, we want to exclude the effect of choosing a different planner. As for the repeating issue, we follow a similar reporting procedure as previous LLM work (such as [2], [3], and [4]) and do not repeat the experiments multiple times. The reason behind this choice is that the results do not vary too much, as opposed to experiments for reinforcement learning methods.  How the Bayesian perspective is connected to the Model-Predictive Control-based algorithm.
> >
> >
> > [1] Shinn, Noah, Beck Labash, and Ashwin Gopinath. "Reflexion: an autonomous agent with dynamic memory and self-reflection." arXiv preprint arXiv:2303.11366 (2023).
> >
> > [2] Sun, Haotian, et al. "AdaPlanner: Adaptive Planning from Feedback with Language Models." arXiv preprint arXiv:2305.16653 (2023).
> >
> > [3] Zhang, Yufeng, et al. "What and How does In-Context Learning Learn? Bayesian Model Averaging, Parameterization, and Generalization." arXiv preprint arXiv:2305.19420 (2023).
> >
> > [4] Lee, Jonathan N., et al. "Supervised Pretraining Can Learn In-Context Reinforcement Learning." arXiv preprint arXiv:2306.14892 (2023).
> >
> > [5] Hollmann, Noah, et al. "Tabpfn: A transformer that solves small tabular classification problems in a second." arXiv preprint arXiv:2207.01848 (2022).
> >
> > [6] Yao, Shunyu, et al. "Tree of thoughts: Deliberate problem solving with large language models." arXiv preprint arXiv:2305.10601 (2023).
> >
> > [7] Yao, Shunyu, et al. "React: Synergizing reasoning and acting in language models." arXiv preprint arXiv:2210.03629 (2022).
> >
> > [8] Gruver, Nate, et al. "Large language models are zero-shot time series forecasters." arXiv preprint arXiv:2310.07820 (2023).

---

> > > ### Author Response · Authors · 2023-11-23
> > > **A new experiment is added**
> > >
> > > We are writing to kindly remind that we also add a new experiment in a two-arm bandit setting to make a clearer comparison with LLMs, Thompson sampling (posterior sampling), and Bayesian model averaging. Please see the details and plot in Appendix D of the updated manuscript. The result shows that Thompson sampling (posterior sampling) and LLMs perform better than Bayesian model averaging due to a more efficient exploration, which also numerically supports Assumption 4.1. We hope these clarifications solve your concern and are pleased to have more discussions.

---

### Author Response · Authors · 2023-11-22
**General Response (Part I)**

Thank each reviewer for the review. We provide comments on some common questions here.

**1. Verification of Assumption 4.1.**
First, we want to clarify that Assumption 4.1 casts LLM's In-Context Learning as an implicit Bayesian inference, which finds support in various existing literature [1], [2], [8]. To reinforce its theoretical foundation, we have rigorously verified Assumption 4.1 in Appendix D, outlining the requisite regularity conditions and justifications for each. Furthermore, we extend our analysis to accommodate cases where LLM approximates the posterior within an error margin $\iota$, resulting in a bounded additional regret of $\iota \cdot T$. Importantly, our novel RAFA algorithm (Algorithm 1) stands distinct from prevalent LLM mechanisms ([3], [4], [5], [6]), and our analysis, guided by Algorithm 3 and Assumption 4.1, only elucidates the theoretical performance of RAFA.

Besides, we conduct two experiments to numerically verify Assumption 4.1.

**Experiment 1.** We test if GPT-3 can make an implicit Bayesian inference for the linear regression problem. We first generate i.i.d. data $[(x_i,y_i)]\_{\ i = 1}^N$ from $x_i \sim \mathcal{N}(0.8,1), y_i\sim \mathcal{N}(0.5x_i,0.04)$. Then, we feed the following prompt to GPT-3 API (temperature=1.0, top_p=0.8, these parameter configurations can be found in [7]) to obtain 200 predictions when $x_{\text{pred}} = 1$. Specifically, the prompts are "The following data is for linear regression: $(x_1, y_1), (x_2, y_2), \cdots, (x_N, y_N), (1.000000,$". Then, we plot the histogram of the LLM-generated predictions and the density function of the calculated $y_{\text{pred}}$ by posterior sampling, where we choose the prior of the coefficient as $\mathcal N(0,0.5)$. We report the results in Figure 8 of the updated manuscript. We find that the histogram of LLM predictions approximately matches the theoretical distribution of the prediction given by the posterior sampling. Also, with an increasing number of generated data, the variance of LLM predictions decays, which shows the contraction property. We also examine variance heterogeneity in the two-dimensional Bayesian linear regression setting. The variance heterogeneity of predictions helps LLMs give more diverse answers when the current prompt contains less relevant information. \texttt{RAFA} relies on the diversity of LLM predictions to guarantee the exploration for those states less explored, hence we wish the variance of LLM predictions for the less explored states is higher. Please check more details and results in Appendix D of the updated manuscript.

**Experiment 2.** We also test if the LLM performs an implicit Bayesian inference in the ALFWorld environment. We begin by examining the contraction property in the ALFWorld environment. Specifically, the model LLM is prompted to predict the observation of a certain action "go to countertop 1". The prediction result is provided in Figure 9 of the updated manuscript. We find that as more action-observation paris are added to the GPT-3 prompt when timestep increases, the number of distinct prediction responses decreases, i.e., the model LLM has decreasing uncertainty of what objects a certain place has after observing the locations of objects at other places. This corresponds to a decreasing entropy and contraction of the posterior distribution, which supports Assumption 4.1. Besides, to show the variance heterogeneity of the predictions, we prompt the model LLM to predict the next observations when taking actions that the agent has less relevant information. Specifically, in the trajectory that the agent has never explored any dining table, we prompt the model LLM to predict the observation after taking the action "go to dining table 1" at different timesteps. The result is shown in Figure 10 of the updated manuscript. We can observe that the number of distinct predictions of taking this action is much larger the "go to countertop 1" action where similar locations such as "countertop 2" and "countertop 3" are explored by the agent in this episode. This indicates the variance heterogeneity of the predictions.

---

> ### Author Response · Authors · 2023-11-22
> **General Response (Part II)**
>
> **2. Comparison with RL and Posterior Sampling.**
> From the theoretical perspective, our analysis is aimed at discussing the theoretical performance of our proposed LLM mechanisms (RAFA), where we adopt the analysis tools from the Thomspon sampling (PSRL) in RL. Besides, our sample complexity analysis extends the present PSRL to an infinity horizon setting beyond the linear structure assumption, which is still missing in the current literature. Since we want to interpret the theoretical performance of LLM agents, it is important to let the analysis accommodate general function approximation. For the linear setting, we also refine the analysis in the previous literature ([10,11]), which assumes a bounded reward while considering the reward to be Gaussian distributed. On the other hand, we claim that our major contribution is not on PSRL but on orchestrating reasoning (learning and planning) and acting following the principled approach in RL. Please see the Appendix A for more detailed discussions. Empirically, we do not need to update the parameter nor sample a parameter from the posterior. LLM agent makes an implicit Bayesian inference via In-Context learning, which enables LLM agents to achieve good performance with a few interactions. Since our proposed method is a refinement of previously proposed LLM mechanisms, we use the same large language model for a fair comparison.
>
> **3. Format Issue**
> We acknowledge the feedback regarding formatting and are committed to optimizing the paper's readability. In the updated manuscript, we condensed the content to maintain clarity while ensuring the paper's coherence and comprehensiveness. We also added a conclusion section to help readers get the gist quickly.
>
> **4. An additional metric other than sample complexity for evaluation.**
> To evaluate the performance of RAFA in a more profound way, we also consider other metrics like token complexity for evaluation. We compare the token efficiency of our method on the Game of 24 with various baseline methods and the results are shown in Appendix G.1 of the updated manuscript. RAFA is superior in terms of token complexity. Methods that lack planning like Reflexion have a low token demand, however, it is not enough to compensate for the drop in performance. Methods that lack in-context learning like Tree of Thoughts would generate unnecessary repeated trials due to lack of reflection and improvement, which makes the method token inefficient.
>
>
> [1] Zhang, Yufeng, et al. "What and How does In-Context Learning Learn? Bayesian Model Averaging, Parameterization, and Generalization." arXiv preprint arXiv:2305.19420 (2023).
>
> [2] Lee, Jonathan N., et al. "Supervised Pretraining Can Learn In-Context Reinforcement Learning." arXiv preprint arXiv:2306.14892 (2023).
>
> [3] Yao, Shunyu, et al. "Tree of thoughts: Deliberate problem solving with large language models." arXiv preprint arXiv:2305.10601 (2023).
>
> [4] Yao, Shunyu, et al. "React: Synergizing reasoning and acting in language models." arXiv preprint arXiv:2210.03629 (2022).
>
> [5] Shinn, Noah, Beck Labash, and Ashwin Gopinath. "Reflexion: an autonomous agent with dynamic memory and self-reflection." arXiv preprint arXiv:2303.11366 (2023).
>
> [6] Sun, Haotian, et al. "AdaPlanner: Adaptive Planning from Feedback with Language Models." arXiv preprint arXiv:2305.16653 (2023).
>
> [7] Gruver, Nate, et al. "Large language models are zero-shot time series forecasters." arXiv preprint arXiv:2310.07820 (2023).
>
> [8] Hollmann, Noah, et al. "Tabpfn: A transformer that solves small tabular classification problems in a second." arXiv preprint arXiv:2207.01848 (2022).
>
> [9] Han, Chi, et al. "In-Context Learning of Large Language Models Explained as Kernel Regression." arXiv preprint arXiv:2305.12766 (2023).
>
> [10] Lu, Xiuyuan, and Benjamin Van Roy. "Information-theoretic confidence bounds for reinforcement learning." Advances in Neural Information Processing Systems 32 (2019).
>
> [11] Russo, Daniel, and Benjamin Van Roy. "Learning to optimize via posterior sampling." Mathematics of Operations Research 39.4 (2014): 1221-1243.

---

### Meta-Review · Area_Chair_xV4P · 2023-11-28

**Metareview:**

The promise of a definitive way to combine planning and LMs with regret guarantees is very valuable, however it seems that the paper is not quite ready for publication in its current state and would benefit from another round of review.
Reviewers raised concerns about the value and assumptions required for the theoretical analysis, and the novelty of the approach. Concerns raised about structure, clarity and related work were addressed in part by the rebuttal, but were substantial changes for a rebuttal and didn't fully assuage reviewer concerns.

**Justification For Why Not Higher Score:**

Reviewers raised concerns about the value and assumptions required for the theoretical analysis. Concerns raised about structure, clarity and related work were addressed in part by the rebuttal, but were substantial changes that were difficult to re-evaluate within the discussion period.

**Justification For Why Not Lower Score:**

N/A

---

### Decision · Program_Chairs · 2024-01-16

Reject